# FROM EMBEDDING TO CONTROL: REPRESENTATIONS FOR STOCHASTIC MULTI-OBJECT SYSTEMS

**Xiaoyuan Cheng**[a]* **Yiming Yang**[a] **Wei Jiang**[b] **Chenyang Yuan**[c]
**Zhuo Sun**[d] **Yukun Hu**[a]†

[a]Dynamic Systems Lab, University College London, UK
[b]Independent Researcher, Hong Kong, China
[c]Electrical and Electronic Engineering, University of Sheffield, UK
[d]Statistics and Data Science, Shanghai University of Finance and Economics, China

## ABSTRACT

This paper studies how to achieve accurate modeling and effective control in stochastic nonlinear dynamics with multiple interacting objects. However, non-uniform interactions and random topologies make this task challenging. We address these challenges by proposing *Graph Controllable Embeddings* (GCE), a general framework to learn stochastic multi-object dynamics for linear control. Specifically, GCE is built on Hilbert space embeddings, allowing direct embedding of probability distributions of controlled stochastic dynamics into a reproducing kernel Hilbert space (RKHS), which enables linear operations in its RKHS while retaining nonlinear expressiveness. We provide theoretical guarantees on the existence, convergence, and applicability of GCE. Notably, a mean field approximation technique is adopted to efficiently capture inter-object dependencies and achieve provably low sample complexity. By integrating graph neural networks, we construct data-dependent kernel features that are capable of adapting to dynamic interaction patterns and generalizing to even unseen topologies with only limited training instances. GCE scales seamlessly to multi-object systems of varying sizes and topologies. Leveraging the linearity of Hilbert spaces, GCE also supports simple yet effective control algorithms for synthesizing optimal sequences. Experiments on physical systems, robotics, and power grids validate GCE and demonstrate consistent performance improvement over various competitive embedding methods in both in-distribution and few-shot tests.

## 1 INTRODUCTION

Controlling nonlinear dynamics with continuous state and action spaces across multiple interacting objects is challenging in domains such as network systems (Qin et al., 2022), robotics (Yoneda et al., 2021), and autonomous agent systems (Gelada et al., 2019). Model-based control algorithms offer a promising solution to this problem (Jacobson & Mayne, 1970; Todorov & Li, 2005), typically approximating the nonlinear control problem through global or local linearization techniques. While these methods have demonstrated strong performance (Tassa et al., 2007; Levine & Koltun, 2013), they generally assume access to a known system model and a carefully designed, low-dimensional state representation. In practice, however, system dynamics are often unknown and governed by complex interactions among multiple objects, which significantly complicates control design (Bullo et al., 2018). To address these challenges, we propose a framework, termed *graph controllable embedding* (GCE), which learns controllable embeddings [1] of stochastic multi-object dynamics for efficient control. GCE consists of: (1) *Modeling*, where the embeddings capture system dynamics and interactions directly from different stochastic graph representations; and (2) *Control*, where simple linear control algorithms are deployed within the learned embedding space.

---

*Email to first author: `ucesxc4@ucl.ac.uk`

†Email to corresponding author: `yukun.hu@ucl.ac.uk`

[1]A controllable embedding is a representation space where system dynamics are modeled to allow linear or locally linear control synthesis, following (Banijamali et al., 2018).

**Controllable Embedding.** A common approach to handling complex, unknown dynamics is to learn a latent space (e.g., smooth manifold or function space) in which the system evolution becomes easier to model and control (Ha & Schmidhuber, 2018; Watter et al., 2015). The goal of constructing such latent spaces is to simplify the dynamics into an approximately linear or globally/locally linear one, enabling the use of linear control methods (Mauroy & Goncalves, 2016; Banijamali et al., 2018). The mainstream for learning globally linear embeddings is Koopman theory (Brunton et al., 2021; Koopman, 1931; Mezic, 2020). It lifts system states into an infinite-dimensional function space where the dynamics evolve linearly (Korda & Mezić, 2018; Mauroy et al., 2020; Bevanda et al., 2021; Cheng et al., 2023). However, applying Koopman-based methods to stochastic multi-object systems is challenging due to two key limitations. First, conventional Koopman operators are originally formulated for deterministic dynamics, making their extension to stochastic settings non-trivial (Brunton, 2019). Second, most Koopman-based models treat the system as a single entity, thereby neglecting the relational topology among interacting objects (Brunton et al., 2021). As a result, they often exhibit poor generalization for multi-object environments and require parameterizations that scale quadratically with the number of objects, increasing the risk of overfitting. Another stream focuses on learning locally linear dynamics (Levine et al., 2020; Mhammedi et al., 2020; Klushyn et al., 2021), often leveraging variational autoencoders (VAE) for constructing low-dimensional manifolds (Farenga et al., 2024; Mudrik et al., 2024). While effective for reconstructing high-dimensional observations such as images, they also neglect object-level interactions, making them less suitable for multi-object systems and ultimately limiting their control performance.

**Graph Representation.** Graph neural networks (GNNs) offer a natural framework for modeling interactions among multiple objects. Early workstreams (Battaglia et al., 2016; Chang et al., 2016; Poli et al., 2021) laid the foundation for data-driven physics simulators by modeling multi-object dynamics using GNN-based architectures. Building on this, subsequent studies (Li et al., 2018; 2019; Sanchez-Gonzalez et al., 2020; Han et al., 2022; Luo et al., 2023; Poli et al., 2021) adopted message-passing networks to learn object-centric dynamics over graphs. However, these methods primarily target prediction tasks rather than control. Consequently, the learned embeddings do not have a linear or locally linear dynamics, which can be easy for downstream control. Then, either additional local linearization techniques or difficult nonlinear control methods are needed, which complicates the problem studied in this paper. Another workstream focuses on model-based multi-agent reinforcement learning for multi-object systems (Jiang et al., 2018; Liu et al., 2020; Zhang et al., 2022; Haramati et al., 2024). While these approaches leverage learned dynamics to improve sample efficiency and value function approximation, the resulting dynamics are neither linear nor locally linear in the embedding space, making them unsuitable for synthesizing optimal sequences.

A natural idea for the studied problem in this paper is to integrate the controllable embedding and graph representation together. For instance, Li et al. (2020) proposed a compositional Koopman method for *deterministic* multi-object systems. Specifically, GNN is first used to encode each object's state into a latent representation. Then, a shared Koopman operator is constructed to linearly evolve all latent states in a common embedding space. While these kinds of methods are empirically effective, several challenges remain. (1) *Theoretical gap in stochastic setting*: existing methods for stochastic multi-object dynamics lack rigorous theoretical foundations with no formal theories to guide the design of controllable embeddings. (2) *Unrealistic relational assumptions*: they often assume uniform interaction among all neighbors, a simplification that is typically misspecified in a probabilistic sense; (3) *Limited scalability and generalization*: they lack thorough evaluation of scalability and generalization to large-scale or random graphs in stochastic multi-object systems.

Motivated by these findings, GCE in Figure 2 is proposed here to address the above challenges. In detail, we study how to embed stochastic nonlinear dynamics into a general reproducing kernel Hilbert space (RKHS) where the multi-object dynamics become linear and linear control methods can be easily implemented. The choice of Hilbert space embeddings is inspired by their compatibility with two properties underlying controllable embeddings (Song et al., 2009; 2010; Fukumizu et al., 2013; Uehara et al., 2022): (1) Modeling: they provide a flexible and expressive representation to capture complex dynamics with multi-object interactions; (2) Control: their inherent linearity supports simple and analytically tractable control policies. Our key contributions are as follows:

(1) **Generalized framework:** A theoretical framework for learning controllable embeddings in multi-object systems with formal consistency and convergence guarantees. Unlike prior deterministic methods, GCE generalizes representation, prediction, and control synthesis within a probabilistically interpretable setting; (2) **Adaptive interaction modeling:** One specific embedding based on

mean field approximation, supported by theoretical guarantees. This breaks the uniform neighbor influence assumption by adaptively capturing non-uniform interactions and achieves provably low sample complexity. Moreover, we further test multiple forms of potential energy functions, demonstrating the flexibility and effectiveness of our framework. (3) **Scalability and generalization:** GCE naturally scales to large systems with random graphs and remains robust under uncertainty and noise, enabling effective control in challenging environments.

## 2 PRELIMINARIES

**Notation and Setup.** Blackboard bold letters (e.g., $\mathbb{O}$, $\mathbb{A}$, $\mathbb{H}$) denote compact spaces, capital letters (e.g., $O_t$, $A_t$, $H_t$) denote random variables, and lowercase letters (e.g., $o_t$, $a_t$, $h_t$) denote their realizations. At time $t$, $O_t$ and $A_t$ are the observation and action random variables with realizations $o_t \in \mathbb{O}$ and $a_t \in \mathbb{A}$, respectively. The history random variable $H_t$ summarizes past information (here $H_t := O_{t-1}$) with realization $h_t \in \mathbb{H}$. The future observation $O_t$ follows the conditional distribution $P(O_t \mid A_t = a_t, H_t = h_t)$—abbreviated as $P(O_t \mid a_t, h_t)$ hereafter—given the executed action $a_t$ and the history $h_t$. We use Greek letters $\phi$ and $\psi$ to denote feature maps into RKHSs, with $\phi_{\mathbb{O}}$, $\phi_{\mathbb{A}}$, and $\phi_{\mathbb{H}}$ mapping into $\mathcal{H}_{\mathbb{O}}$, $\mathcal{H}_{\mathbb{A}}$, and $\mathcal{H}_{\mathbb{H}}$, respectively; $\psi_t^O := \phi_{\mathbb{O}}(O_t)$ and $\psi_t^o := \phi_{\mathbb{O}}(o_t)$ represent the mapped random variable and its realization, respectively. The symbols $\otimes$, $\oplus$, and $\odot$ represent the tensor product, the direct sum, and the element-wise product, respectively. To clarify indexing, subscripts indicate time steps, while superscripts indicate object indices. For example, $o_t^i$ denotes the observation of the $i$-th object at time step $t$. Finally, we define the shorthand $[N] := \{1, \ldots, N\}$ for the consecutive integer set. please see more notations in Tables 6 and 7 in Appendix A.

**Definition 1** (Hilbert Space Embedding of Conditional Distributions (Sriperumbudur et al., 2010))**.** *For the conditional distribution $P(O_t \mid a_t, h_t)$ of the future observation random variable $O_t$ given the action $a_t$ and history $h_t$, its Hilbert space embedding is defined as the conditional expectation of the feature map in a RKHS $\mathcal{H}_{\mathbb{O}}$:*

$$\mathbb{E}\big[\psi_t^O \mid a_t, h_t\big] = \int_{o_t \in \mathbb{O}} \psi_t^o \, P(o_t \mid a_t, h_t) do_t = \mathcal{C}_{O|AH}\big[\psi_t^h \otimes \psi_t^a\big], \tag{1}$$

*where $\psi_t^h := \phi_{\mathbb{H}}(h_t)$ and $\psi_t^a := \phi_{\mathbb{A}}(a_t)$ are the characteristic kernel features in their respective RKHSs $\mathcal{H}_{\mathbb{H}}$ and $\mathcal{H}_{\mathbb{A}}$; $\mathcal{C}_{O|AH} : \mathcal{H}_{\mathbb{H}} \otimes \mathcal{H}_{\mathbb{A}} \to \mathcal{H}_{\mathbb{O}}$ is the conditional embedding operator.*

This embedding yields a nonparametric representation of the conditional distribution in the RKHS (see more details about RHKS in Appendix C), thereby avoiding density estimation (see formal analysis in Appendix D.1 and D.2). *With characteristic feature maps, the embedding uniquely specifies the conditional distribution (Sriperumbudur et al., 2010).* Intuitively, the tensor product $\psi_t^h \otimes \psi_t^a$ represents the joint features of the conditioning variables $(a_t, h_t)$, allowing the operator $\mathcal{C}_{O|AH}$ to act linearly on the joint input space and represent the conditional expectation in closed form. In practice, $\mathcal{C}_{O|AH}$ enables efficient one-step prediction from history and action features, and can be applied recursively to perform multi-step rollouts and facilitate control planning (shown in Figure 1).

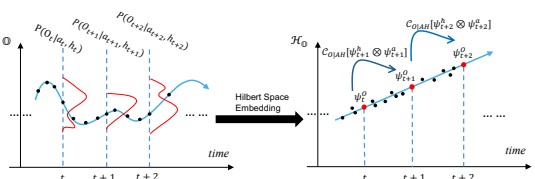

Figure 1: Hilbert space embedding of conditional distributions. *Left:* Stochastic nonlinear dynamics as evolving conditional distributions $P(O_t|a_t, h_t)$ (red curves: probability distributions, black dots: realizations). *Right:* After embedding into RKHS, dynamics become linear under $\mathcal{C}_{O|AH}$ (red dots: expectations).

**Problem Formulation.** We consider a stochastic dynamics composed of $N$ interacting objects, represented as a graph $G_t = (\mathcal{V}_t^o, \mathcal{E})$ at time $t$. Each one $o_t^i \in \mathcal{V}_t^o := \{o_t^i\}_{i=1}^N$ corresponds to the observation of the $i$-th object, and $a_t^i$ denotes the control action applied to it. The full set of actions at time $t$ is denoted by $\mathcal{V}_t^a = \{a_t^i\}_{i=1}^N$. The relational structure is encoded by a fixed binary adjacency matrix $\mathcal{E} = \{e^{i,j}\}_{i,j=1}^N$, where $e^{i,j} = 1$ indicates an edge from object $i$ to $j$, and $e^{i,j} = 0$ otherwise. For each object $i$, we denote its (inclusive) neighborhood by $\mathcal{E}(i) := \{j | e^{i,j} = 1\} \cup \{i\}$. Our goal

Figure 2: An example illustration of the GCE framework. The robot with multiple interconnected objects is initialized on top of a thin pole. At time step $t$, the observation is embedded into an RKHS via characteristic feature maps. A mean field approximation is then applied: the history features are computed via an element-wise product to reduce computation, while the action features preserve their structure for optimization (see math in Equation 9). The predicted features are mapped back to predict the $(t+1)$-step observation, enabling the robot to stabilize on the pole.

is to construct a controllable embedding to model the stochastic multi-object dynamics governed by the unknown transition function $\mathcal{F}$:

$$\mathcal{V}_{t+1}^o = \mathcal{F}(\mathcal{V}_t^o, \mathcal{V}_t^a, \mathcal{E}), \tag{2}$$

where $\mathcal{F}$ captures the time evolution of object observations based on actions and interactions. After system identification, the second goal is to enable effective control in the learned embedding space. Specifically, we seek a representation in which the dynamics are amenable to linear control algorithms. The objective is to find actions $\{\mathcal{V}_t^a\}_{t=1}^M$ that minimizes a cumulative cost over $M$ steps:

$$\min_{\{\mathcal{V}_t^a\}_{t=1}^M} \sum_{t=1}^M \mathcal{J}(\mathcal{V}_t^o, \mathcal{V}_t^a), \qquad \text{s.t. } \mathcal{V}_{t+1}^o = \mathcal{F}(\mathcal{V}_t^o, \mathcal{V}_t^a, \mathcal{E}), \tag{3}$$

where $\mathcal{J} : \mathcal{V}^o \times \mathcal{V}^a \to \mathbb{R}$ is a cost function defined over graph representations. As shown in Figure 1, Hilbert space embeddings offer a general mechanism to represent stochastic dynamics in a form amenable to linear control. This motivates the extension of Hilbert space embeddings to GCE, which captures stochastic multi-object dynamics while enabling efficient linear control. In the next section, we detail how to construct such embeddings to jointly support the modeling objective in Equation 2 and the control objective in Equation 3.

## 3 Framework of Graph Controllable Embeddings

In this section, we first establish the theoretical foundation of GCE. Then, we propose a mean field-based instantiation for capturing non-uniform interaction weights among multiple objects. Finally, we compare different embeddings within the GCE framework, analyzing their properties across various settings and deriving sample complexity guarantees.

### 3.1 Foundation of GCE: Embedding and Consistency

We formalize the Hilbert space embedding of a stochastic multi-object system as an instantiation of GCE. Using the feature maps in Definition 1, let $\psi_t^{h,i}$, $\psi_t^{o,i}$, and $\psi_t^{a,i}$ denote the realization features of the $i$-th object's history, future observation and action, respectively. For each pair of connected objects $i$ and $j$, we define a conditional embedding operator $\mathcal{C}_{O^i|A^jH^j}$ that models the contribution of object $j$'s history and action to the conditional expectation of object $i$'s future observation:

$$\mathbb{E}[\psi_t^{O,i,j} \mid a_t^j, h_t^j] = \mathcal{C}_{O^i|A^jH^j}[\psi_t^{h,j} \otimes \psi_t^{a,j}], \quad \forall i,j \in [N], \tag{4}$$

where $\psi_t^{O,i,j}$ indicates the component of the future observation feature of object $i$ attributable to object $j$'s influence. The overall conditional expectation of object $i$'s observation feature is:

$$\mathbb{E}[\psi_t^{O,i} \mid \{a_t^j, h_t^j\}_{j=1}^N] = \sum_{j \in \mathcal{E}(i)} \mathcal{C}_{O^i|A^jH^j}[\psi_t^{h,j} \otimes \psi_t^{a,j}], \tag{5}$$

where $\mathcal{C}_{O^i|A^jH^j}$ is the zero operator whenever $j \notin \mathcal{E}(i)$. Each $\mathcal{C}_{O^i|A^jH^j}$ implicitly encodes the relative importance of the $j$-th object's influence on the $i$-th object, as these effects are generally not equally distributed. More probabilistic analysis and formal derivation are referred to Appendix D.3.

**Theorem 1** (Consistency and Convergence of GCE). *Let $\widehat{\mathcal{C}}_{O^i|A^jH^j}$ be the empirical estimator of the embedding operator $\mathcal{C}_{O^i|A^jH^j}$, constructed from $T$ i.i.d. samples using characteristic kernel features. For any input set $\{a_t^j, h_t^j\}_{j=1}^N$, define the estimated embedding of object $i$'s future observation as: $\widehat{\psi}_t^{O,i} = \sum_{j \in \mathcal{E}(i)} \widehat{\mathcal{C}}_{O^i|A^jH^j} \left[\psi_t^{h,j} \otimes \psi_t^{a,j}\right]$. Then, as $T \to \infty$, $\left\|\widehat{\psi}_t^{O,i} - \mathbb{E}\left[\psi_t^{O,i} \mid \{\psi_t^{a,j}, \psi_t^{h,j}\}_{j=1}^N\right]\right\|_{\mathcal{H}} \to 0$. Since the kernel feature is characteristic, this convergence implies that the conditional distribution $\widehat{P}(O_t^i \mid \{a_t^j, h_t^j\})$ induced by $\widehat{\mathcal{C}}_{O^i|A^jH^j}$ also converges: $\widehat{P}(O_t^i \mid \{a_t^j, h_t^j\}_{j=1}^N) \xrightarrow{T \to \infty} P(O_t^i \mid \{a_t^j, h_t^j\}_{j=1}^N)$.*

Theorem 1 shows that, given sufficient samples, the conditional embedding operators in GCE exist and converge consistently, ensuring probabilistic consistency of the induced conditional distributions (see proof in Appendix F.1). Although Equation 5 provides a principled formulation for embeddings of stochastic multi-object dynamics, its direct application for control poses two difficulties. First, estimating $\mathcal{C}_{O^i|A^jH^j}$ is computationally intensive due to the high dimensionality introduced by the tensor product $\psi_t^{h,j} \otimes \psi_t^{a,j}$, which in turn increases the required sample size. Second, the entangled representation of actions and histories in the tensor space makes sequential action optimization intractable. Inspired by the decomposition of joint distributions in exponential families (Zhang et al., 2008; Altun et al., 2012), we address these difficulties by disentangling the tensorized feature $\psi_t^{h,j} \otimes \psi_t^{a,j}$ into a simplified form:

$$\mathcal{C}_{O^i|A^jH^j}[\psi_t^{h,j} \otimes \psi_t^{a,j}] \approx [\mathcal{C}_{O^i|H^j}, \mathcal{C}_{O^i|A^j}][\psi_t^{h,j} \| \psi_t^{a,j}] = \mathcal{C}_{O^i|H^j}\psi_t^{h,j} + \mathcal{C}_{O^i|A^j}\psi_t^{a,j}, \quad \forall i,j \in [N], \tag{6}$$

where $[\cdot \| \cdot]$ represents the concatenation operation, $\mathcal{C}_{O^i|H^j}$ and $\mathcal{C}_{O^i|A^j}$ denote a neighbor-specific linear operator that captures the distinct influence of object $j$'s history and action on object $i$, respectively. In this reformulation, the tensor map in Equation 6 is replaced by concatenation $[\psi_t^{h,j} \| \psi_t^{a,j}]$. This concatenation approximates linear relationships in the probability distributions while neglecting higher-order interactions in the original tensor product. Consequently, the computational complexity is significantly reduced, and the action and history representations are disentangled, making sequential action optimization tractable. See more formal analysis in Appendix D.4.

## 3.2 Efficient Embedding: Adaptive Mean Field Approximation

Although valid controllable embeddings have been formally defined in Equation 6, estimating all linear conditional operators $\mathcal{C}_{O^i|A^jH^j}$ remains computationally prohibitive. The number of such operators grows with the number of edges and can scale quadratically with the number of objects in dense graphs, i.e., $\mathcal{O}(N^2)$ in the worst case. However, modeling all pair-wise interactions is often unnecessary. To mitigate this issue, we leverage mean field approximation, which significantly reduces the sample complexity by summarizing the collective influence of neighbors.

Probabilistically, the mean field approximation avoids dealing with the full joint conditional distribution over all pair-wise history-observation interactions. It replaces this with an aggregated form that captures the combined influence of neighbors, while keeping their contributions unequal. A probabilistic derivation of this factorization, including its separation into history and action terms, is provided in Appendix D.5. In such case, the interaction weight of node $i$ interacting with neighbor $j$ at time $t$ is approximated as

$$\alpha_t^{i,j} = \frac{\exp(f(\psi_t^{h,i}, \psi_t^{h,j}))}{\sum_{k \in \mathcal{E}(i)} \exp(f(\psi_t^{h,i}, \psi_t^{h,k}))}, \tag{7}$$

where $\alpha_t^{i,j} \in [0, 1]$ denotes the Boltzmann-Gibbs weight, $f$ is a pair-wise negative potential energy function, and the denominator $\sum_{k \in \mathcal{E}(i)} \exp(f(\psi_t^{h,i}, \psi_t^{h,k}))$ serves as the partition function, normalizing the probability one. $f$ can be either predefined or learned via neural networks. Based on the mean field approximation, the expected feature of object $i$ can be computed as the weighted aggregation from its neighbors: $\sum_{j \in \mathcal{E}(i)} \alpha_t^{i,j} \psi_t^{h,j}$. Accordingly, $\sum_{j \in \mathcal{E}(i)} \mathcal{C}_{O^i|H^j}\psi_t^{h,j}$ can be approximated

by applying a shared conditional operator to this aggregated feature:

$$\sum_{j \in \mathcal{E}(i)} \mathcal{C}_{O^i|H^j} \psi_t^{h,j} \approx \mathcal{C}_{O^i|H} \big( \sum_{j \in \mathcal{E}(i)} \alpha_t^{i,j} \psi_t^{h,j} \big), \tag{8}$$

where $\mathcal{C}_{O^i|H}$ is a shared approximation operator derived under the homogeneity setting [2] to furthermore reduce computational complexity. This formulation captures the essence of the mean field approximation by replacing heterogeneous, neighbor-specific interactions with a unified, averaged effect, to reduce per-object computation to constant time, independent of the neighbor number.

Unlike $\mathcal{C}_{O^i|H}$, we keep $\mathcal{C}_{O^i|A^j}$ unchanged to facilitate the optimization of sequential actions. Finally, the overall conditional expectation of object $i$'s observation feature in Equation 5 changes to:

$$\mathbb{E}[\psi_t^{O,i}|\{a_t^j, h_t^j\}_{j=1}^N] = \mathcal{C}_{O^i|H} \big( \sum_{j \in \mathcal{E}(i)} \alpha_t^{i,j} \psi_t^{h,j} \big) + \sum_{j \in \mathcal{E}(i)} \mathcal{C}_{O^i|A^j} \psi_t^{a,j} \tag{9}$$

Notably, the action-to-observation block is often sparse in real-world scenarios. Thus, the complexity is dominated by the history-to-observation block whose computational complexity can be reduced by making all conditional linear operators $\mathcal{C}_{O^i|H}$ share the same parameters, leveraging the homogeneous nature of the relational structure (since the transition probability of each object is the same). Under this representation, the total computation time in the history part can be reduced to $\mathcal{O}(N)$.

### 3.3 COMPARISON OF DIFFERENT EMBEDDINGS: PROPERTIES AND SAMPLE COMPLEXITY

To generalize our framework, we compare four embedding formulations existing in GCE, each formally defined in the main text and derived in detail from the probabilistic view in the appendices: (1) Tensor form(**Tensor**, Equation 5; Appendix D.3): full tensor-product embedding without controllable structure constraints. This form is probabilistically consistent (see Theorem 1), as it retains all high-order interaction terms; (2) Dense form (**Dense**, Equation 6; Appendix D.4): controllable embedding without homogeneity or mean field approximation. It remains consistent for all pair-wise terms but ignores higher-order interactions in Hilbert space embedding; (3) Homogeneous form (**Hom**; cf. Li et al. (2020)): assumes graph homogeneity and enforces uniform neighbor weights, leading to a misspecified product form from a probabilistic viewpoint; (4) Homogeneous form with mean field approximation (**Hom + Mean**, Equation 9; Appendix D.5): leverages homogeneity and adaptive neighbor weighting via Boltzmann–Gibbs weights. The product form is an approximation to the true joint distribution over the entire graph, retaining unequal influence from each neighbor while reducing complexity. Table 1 summarizes their properties, including satisfaction of controllable embedding, applicable scenarios, sample complexity, computation time, ability to incorporate adaptive weights, and generalizability to random graphs. Overall, **Hom + Mean** strikes a balance between sample efficiency and expressiveness by approximating the probabilistic structure with adaptive weights, while avoiding the misspecification in **Hom** and the combinatorial complexity in **Tensor** and **Dense**. The complete quantitative analysis is given in Appendix E with proved sample complexity and generalization on graphs in Appendix F.2.

Table 1: Comparison of four different embeddings in GCE.

| Property | Tensor | Dense | Hom | Hom + Mean |
|---|---|---|---|---|
| Controllable embedding | ✗ | ✓ | ✓ | ✓ |
| Scenario | heterogeneity | heterogeneity | homogeneity | homogeneity |
| Sample Complexity | very high | high | medium | low |
| Computation Time | very high | high | high | low |
| Adaptive Weight | ✓(implicit) | ✓(implicit) | ✗(misspecified) | ✓(explicit) |
| Generalize to Random Graphs | hard | hard | medium | easy |

From a sample complexity perspective, **Hom + Mean** enjoys two key advantages: the mean field approximation replaces $\mathcal{O}(N^2)$ neighbor-specific operators with a single shared operator, greatly

---

[2]In our setting, "homogeneity" means all nodes share the same operator mapping from history features to observation features, while neighbor influencing weights vary. The shared operator represents one universal law of interaction. The local interaction rule between neighboring objects is the same, e.g., in Rope environment in Figure **??**. Since the embedding feature for a "top mass" and a "regular mass" are different, applying the same operator to them will naturally produce different dynamic evolutions.

reducing the number of parameters to estimate, while adaptive weights preserve variability across neighbors without explicit pair-wise modeling. These properties enable efficient learning and few-shot adaptation to new or random graphs, unlike **Tensor** and **Dense** (poor transfer) or **Hom** (misspecified weights). Using random matrix theory, we show that the required sample size of **Hom** + **Mean** scales with an effective operator dimension rather than the total number of graph edges, yielding much smaller requirements for training data than pair-wise conditional embedding operators on graphs (see Theorems a, b, c in Appendix F.2).

## 4  IMPLEMENTATION

As shown in Figure 2, the implimentation of GCE is as follows: (1) an encoder that maps observations into RKHSs, (2) various ways for estimating adaptive Boltzmann-Gibbs weights, (3) tailored loss functions for controllable embedding, and (4) a linear quadratic regulator (LQR) controller in the RKHS. Pseudo code is given in Appendix H.

**Mapping to RKHSs.** A message passing GNN encoder projects the $i$-th object's history and future observations into RKHSs $\psi_t^{h,i} \in \mathbb{R}^{d_h}, \psi_t^{o,i} \in \mathbb{R}^{d_o}$. The observation feature is $\psi_t^{o,i} = f_{\mathcal{V}}(o_t^i, \oplus_{j \in \mathcal{E}(i)} f_{\mathcal{E}}(o_t^i, o_t^j, e^{i,j}))$, where $f_{\mathcal{V}}$ and $f_{\mathcal{E}}$ are neural networks extracting deep kernel features; and action $a_t^i$ uses a fixed linear projection to its feature $\psi_t^{a,i} \in \mathbb{R}^{d_a}$ for easier optimization (see Appendix H).

**Boltzmann–Gibbs Weights.** To compute adaptive interaction weights, we employ a Nadaraya–Watson kernel estimator, which ensures a stable approximation of Boltzmann–Gibbs weights. For each object $i$, the weight is given by $\alpha_t^{i,j} = G(\psi_t^i, \psi_t^j) / \sum_{k \in \mathcal{E}(i)} G(\psi_t^i, \psi_t^k)$, where $G(\cdot, \cdot)$ is an exponential kernelized energy function of the form $G(\psi_t^i, \psi_t^j) = \exp\left(f(\psi_t^{h,i}, \psi_t^{h,j})\right)$. We evaluate both kernel-based and neural parameterizations of $f$ as shown in Table 2; please refer more details to Appendix D.5).

Table 2: Parameterizations of the potential function $f(\cdot, \cdot)$ in Equation 7. $(\sigma, \lambda, \kappa)$ are hyperparameters.

| Type | Name | Potential $f$ |
|---|---|---|
| Kernel-based | Gaussian | $-\frac{\|\psi_t^{h,i} - \psi_t^{h,j}\|_2^2}{2\sigma^2}$ |
| | Laplace | $-\frac{\|\psi_t^{h,i} - \psi_t^{h,j}\|_1}{\lambda}$ |
| | vMF | $\kappa \left(\frac{\psi_t^{h,i}}{\|\psi_t^{h,i}\|_2}\right)^{\top} \left(\frac{\psi_t^{h,j}}{\|\psi_t^{h,j}\|_2}\right)$ |
| Neural-based | MLP | $\mathrm{MLP}([\psi_t^{h,i} \| \psi_t^{h,j}])$ |

**Loss Functions.** The GNN block is then utilized to encode this feature in RKHS, the forward loss $\mathcal{L}_{\text{fwd}}$ in the embedding feature space can be represented as

$$\mathbb{E}_{\mu}\left[\sum_{t=1}^{M}\left\|\begin{bmatrix}\psi_t^{o,1}\\\vdots\\\psi_t^{o,N}\end{bmatrix} - \begin{bmatrix}\hat{\mathcal{C}}_{O^1|H}\\\vdots\\\hat{\mathcal{C}}_{O^N|H}\end{bmatrix} \odot \begin{bmatrix}\sum_j \alpha_t^{1,j}\psi_t^{h,j}\\\vdots\\\sum_j \alpha_t^{N,j}\psi_t^{h,j}\end{bmatrix} - \begin{bmatrix}\hat{\mathcal{C}}_{O^1|A^1} & \cdots & \hat{\mathcal{C}}_{O^1|A^N}\\\vdots & \ddots & \vdots\\\hat{\mathcal{C}}_{O^N|A^1} & \cdots & \hat{\mathcal{C}}_{O^N|A^N}\end{bmatrix}\begin{bmatrix}\psi_t^{a,1}\\\vdots\\\psi_t^{a,N}\end{bmatrix}\right\|_{HS}\right] \tag{10}$$

where $\mu$ is sampled data distribution of realizations $[\psi_t^{o,1}, \cdots, \psi_t^{o,N}]$ and $\|\cdot\|_{HS}$ denotes the Hilbert-Schmidt norm. All conditional operators $\hat{\mathcal{C}}_{O^i|H}$ share the same learnable parameters. To improve latent control quality, we can achieve $M$-step open-loop control by optimizing $M$-step sequential prediction loss. This is implemented by iteratively generating $M$-step sequences in the feature space through autoregressive prediction. After learning the observation features, we utilize the same message passing block as the decoder. The loss function of reconstruction becomes

$$\mathcal{L}_{\text{rec}} = \mathbb{E}_{\mu}\left[\sum_{t=1}^{M}\left\|\psi^{\dagger}\left([\hat{\psi}_t^{O,1}, \ldots, \hat{\psi}_t^{O,N}]^{\top}\right) - [o_t^1, \ldots, o_t^N]^{\top}\right\|_2\right], \tag{11}$$

where $\psi^{\dagger}$ indicates the decoder function, which pulls estimated features $\hat{\psi}_t^{O,i}$ back to the observation space. Our total loss function has two parts, the forward and reconstruction loss, and we train the encoder-decoder GNN blocks and linear conditional operators based on Equations 10 and 11.

**LQR Control.** As presented in Equation 3, we directly solve a quadratic cost function in feature space over $M$ steps:

$$\min_{\{\mathcal{V}_t^a\}_{t=1}^{M}} \mathbb{E}[\sum_{t=1}^{M} \|[\hat{\psi}_t^{O,1},\ldots,\hat{\psi}_t^{O,N}]^{\top} - [\psi_*^{o,1},\cdots,\psi_*^{o,N}]^{\top}\|_{Q_1}^2 + [\psi_t^{a,1},\ldots,\psi_t^{a,N}]^{\top}\|_{Q_2}^2] \quad (12)$$

where $Q_1$ and $Q_2$ are two pre-defined positive definite matrices to measure quadratic costs, and $[\psi_*^{o,1},\cdots,\psi_*^{o,N}]^{\top}$ is a pre-defined target observation feature which should be achieved in $M$ steps. We minimize the control cost function Equation 12 by using convex optimization.

## 5 NUMERICAL EXPERIMENTS

We evaluate our method on four control environments: (1) Rope – a chain of masses with the top mass controlled horizontally; (2) Soft – a soft robot made of interconnected objects; (3) Swim – the soft robot swims freely in fluid (Li et al., 2020); and (4) Power-Grid – a high-dimensional system with random topologies aiming to stabilize node voltages. Robustness is tested with additive white noise at $2\%$, $5\%$, $10\%$, and $20\%$ of observation variance (see more details in Appendix I).

**Metrics and Baselines.** We use open-loop control with a 40-step LQR in Rope, while Soft and Swim generate 64-step control with feedback from step 32. Power-Grid uses a 100-step horizon with feedback at step 50. Performance is evaluated via *control cost* (Equation 12) and *control error* $\|\mathcal{V}_M^o - \mathcal{V}_*^o\|/\|\mathcal{V}_*^o\|$, averaged over 200 runs. Few-shot validation on unseen graphs tests the generalization of the algorithm. The baselines are controllable embedding methods without relational structures, including VAE (Banijamali et al., 2018) and Prediction, Consistency, Curvature (PCC) (Levine et al., 2020), and graph representation methods, including Koopman Polynomial Model (KPM) (Li et al., 2020), Compositional Koopman Operator (CKO) (Li et al., 2020), and GraphODE (Luo et al., 2023), where CKO represents the current state-of-the-art in graph embedding control tasks (see control animations in Appendix I.7).

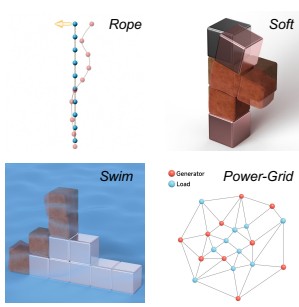

Figure 3: The 3D demonstration of control tasks.

Table 3: Performance comparison for in-distribution and few-shot validation. Control cost and error are shown as mean ± standard deviation, with the best and second-best results highlighted in green and blue, respectively. Due to page limit, for additional Rope and Soft comparison, see Table 9.

| Methods | Environments | In-Distribution Validation | | Few-Shot Validation | |
|---|---|---|---|---|---|
| | | Control cost | Control error | Control cost | Control error |
| VAE | | 573.1 ± 108.7 | 0.73 ± 0.19 | 835.4 ± 113.2 | 0.92 ± 0.15 |
| PCC | | 513.3 ± 92.5 | 0.68 ± 0.15 | 732.8 ± 94.5 | 0.80 ± 0.12 |
| GraphODE | | 417.8 ± 87.9 | 0.52 ± 0.17 | 693.5 ± 58.2 | 0.58 ± 0.09 |
| KPM | | 385.5 ± 75.2 | 0.44 ± 0.06 | 523.4 ± 22.8 | 0.61 ± 0.11 |
| CKO | Swim | 389.1 ± 76.9 | 0.42 ± 0.13 | 421.0 ± 70.0 | 0.44 ± 0.08 |
| Ours (vMF) | | 392.7 ± 73.1 | 0.45 ± 0.09 | 452.3 ± 62.9 | 0.43 ± 0.15 |
| Ours (Laplace) | | 403.1 ± 68.3 | 0.46 ± 0.13 | 435.7 ± 74.4 | 0.45 ± 0.10 |
| Ours (Gaussian) | | **383.7 ± 77.8** | **0.41 ± 0.08** | **404.3 ± 74.2** | **0.41 ± 0.09** |

Table 4: Evaluation under varying noise levels in Power-Grid on random graphs with 100-150 objects. "NaN" means the unstable control. For additional control cost comparison, see Table 10.

| | Method | Noiseless | 2% | 5% | 10% | 20% |
|---|---|---|---|---|---|---|
| | GraphODE | 0.58 ± 0.043 | 0.62 ± 0.073 | NaN | NaN | NaN |
| *Control Error* | KPM | 0.42 ± 0.028 | 0.50 ± 0.022 | NaN | NaN | NaN |
| | CKO | 0.47 ± 0.031 | 0.48 ± 0.034 | 0.51 ± 0.027 | 0.65 ± 0.051 | 0.85 ± 0.055 |
| | Ours (Gaussian) | **0.21 ± 0.005** | **0.27 ± 0.018** | **0.39 ± 0.024** | **0.63 ± 0.037** | **0.83 ± 0.041** |

### 5.1 RESULT ANALYSIS

(1) *Is it necessary to design specific controllable embeddings for multi-object dynamics? Yes.* In Table 3, general-purpose controllable embeddings such as VAE and PCC, while capable of capturing

individual trajectories, fail to provide structured features for control. Although GraphODE incorporates relational structures, it lacks an explicit controllable design and depends on local linearization via auto-differentiation, resulting in suboptimal performance. This highlights that merely encoding relational dependencies is not enough; the embedding must also be tailored to control objectives.

(2) *How well do the empirical findings validate the theoretical predictions of our framework?* We divide the answer into four points. (a) **Analysis of baseline methods.** CKO is an instance of the **Hom** subclass in our framework, with misspecfied uniform weighting. As shown in Figure 9 (Appendix I.5), the empirical prediction error accumulates faster than our methods in both deterministic and stochastic environments. As a result, CKO performance is degraded across all tasks as shown in Table 3. (b) **Failure cases and theoretical justification.** KPM fails under high-noise conditions (see Table 4). This is primarily because Hilbert space embeddings require characteristic features to uniquely represent probability distributions. The polynomial features in KPM are not characteristic, and therefore cannot faithfully embed the underlying distributions (see Theorem 1). (c) **Analysis sample complexity of different embeddings.** To validate the properties summarized in Table 1, we compared: **Dense**, **Hom**, **Hom** + **Mean** and excluded **Tensor** due to its lack of both local and global linearity. In few-shot settings, we varied the number of training trajectories from 1 to 32 to assess sample efficiency and expressiveness. As shown in Table 5, **Hom** + **Mean** consistently outperforms **Dense** and **Hom** with limited samples, owing to the introduction of mean field approximation. As sample size increases, all methods improve due to reduced operator error, aligning with Theorems a–c. While **Dense** eventually has similar control error with **Hom** + **Mean**, **Hom** + **Mean** always maintains lower control costs. (d) **Gaussian Kernel leads to a stable mean field approximation.** Among the three variants for estimating the energy function $f$, Gaussian consistently outperforms vMF and Laplace as shown in Table 3. Its advantage lies in providing a smoother and more stable mean field approximation, which better captures both central tendencies and uncertainty in RKHSs. In contrast, vMF emphasizes directional alignment, and Laplace encourages more uniform energy with slow decay rates, making them less robust across diverse environments. We also experimented with a neural network, but it was unstable for the energy function $f$ in RKHSs, further underscoring the benefit of analytically tractable kernels such as Gaussian.

(3) *To what extent does the non-uniform weighting from the mean field approximation enhance control performance?* As shown in Table 3, our method consistently outperforms baselines across all three tasks in terms of both control cost error. While CKO achieves comparable performance to ours on in-distribution validation for Soft and Swim tasks, it shows a larger performance gap during few-shot generalization. This discrepancy stems from the limitations of misspecified representation (see Figure 12), which reduces model expressiveness. To further validate this observation, we compare the 100-step prediction errors and sample complexity of CKO and our method in Figure 9 and 10 (Appendix I.5). The results validate that our method yields lower error and sample complexity.

(4) *Is the proposed framework capable of generalizing to large-scale multi-object systems with random topologies in noisy environments? Yes.* As shown in Table 4, Power-Grid presents a more challenging setting due to its scale, graph complexity, and random topology. GraphODE and KPM fail consistently above 2% noise, producing unstable (NaN) results. At low noise levels, our method significantly outperforms CKO in both control error and cost. While the performance gap narrows with increasing noise, our approach remains more robust and generalizes better across conditions.

Table 5: Control error under different numbers of training trajectories used for few-shot adaptation in the Rope environment. The table reports performance using varying numbers of demonstration trajectories, referred to as "fitting number". For additional control cost comparison, see Table 11.

|  | Method | 1 | 4 | 8 | 16 | 32 |
|---|---|---|---|---|---|---|
|  | **Dense** | $0.79 \pm 0.25$ | $0.41 \pm 0.11$ | $0.36 \pm 0.12$ | $0.28 \pm 0.08$ | $0.26 \pm 0.10$ |
| *Control Error* | **Hom** | $0.74 \pm 0.25$ | $0.32 \pm 0.08$ | $0.30 \pm 0.06$ | $0.30 \pm 0.06$ | $0.30 \pm 0.06$ |
|  | **Hom** + **Mean** | $\mathbf{0.51 \pm 0.12}$ | $\mathbf{0.29 \pm 0.09}$ | $\mathbf{0.26 \pm 0.08}$ | $\mathbf{0.25 \pm 0.08}$ | $\mathbf{0.23 \pm 0.09}$ |

**Ablation study on Gaussian Kernel.** The Gaussian kernel leads to the best performance across all tasks. The Gaussian kernel bandwidth $\sigma$ in Table 2 can be interpreted as a temperature factor that determines how local features are mixed with those of their neighbors. To investigate its impact, we evaluated $\sigma = \{0.1, 0.5, 1, 2, 3, 4, 5, 10\}$ during few-shot validation in Rope and Soft environments. Due to the quadratic effect of the bandwidth, when $\sigma \geq 10$, the attention weights become nearly uniform across neighboring nodes, which is the reason we do not evaluate $\sigma$ with a value larger than

10. We found in Figure 4 that $\sigma = 2$ achieves the best performance in both control error and control cost. This can be interpreted as physical temperatures: when $\sigma$ is small (Temp $\rightarrow 0$), node dynamics are nearly self-determined due to strong penalties for deviation. When $\sigma$ is large (Temp $\rightarrow \infty$), all neighbors exert nearly equal influence, reflecting maximal thermal mixing. An ablation on feature dimension is provided in Appendix I.6, with dimension 32 achieving the best performance.

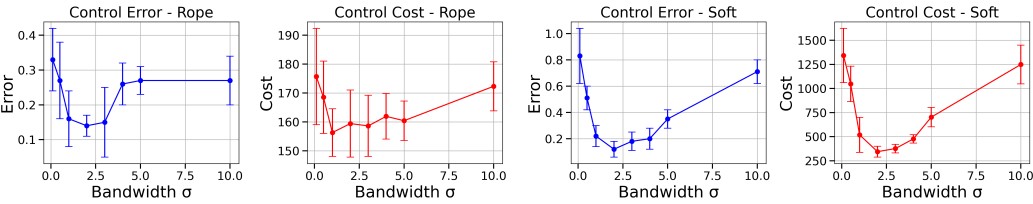

Figure 4: Control error and control cost for varying bandwidth $\sigma$ values in Rope and Soft environments. Results are reported as mean and standard deviation over multiple runs.

## 6 CONCLUSION

We proposed Graph Controllable Embeddings (GCE), a general framework for modeling stochastic multi-object dynamics and synthesizing effective control sequences. By introducing the mean field approximation, GCE efficiently captures inter-object dependencies and achieves provably low sample complexity. Leveraging graph neural networks, our method adapts to dynamic interaction patterns and generalizes to unseen topologies with limited training data. A limitation of our framework is that it currently focuses on pair-wise relations; extending it to richer relational structures, such as hypergraphs and incorporating attention-based techniques for modeling interaction weights remains unexplored.

## ACKNOWLEDGEMENT

The corresponding author would like to thank the financial support provided by the Royal Academy of Engineering through the Industrial Fellowship scheme (IF-2425-19-AI165).

## ETHICS STATEMENT AND REPRODUCIBILITY STATEMENT

This work raises no specific ethical concerns beyond standard practices in machine learning research. All methods, datasets, and hyperparameters are described in detail, and the core code is released in the supplementary materials.

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

## THE USE OF LARGE LANGUAGE MODELS (LLMS)

During the preparation of this manuscript, the authors used ChatGPT to polish the writing (e.g., improving grammar, readability, and clarity). The content, technical contributions, and conclusions of the paper were developed entirely by the authors, who take full responsibility for all ideas and results presented.

## A  NOTATIONS

Table 6: Basic Notations of Hilbert Space Embedding

|  | observation | action | history |
|---|---|---|---|
| compact spaces | $\mathbb{O}$ | $\mathbb{A}$ | $\mathbb{H}$ |
| random variables | $O_t$ | $A_t$ | $H_t$ |
| realizations | $o_t$ | $a_t$ | $h_t$ |
| kernels | $k_O(\cdot, o_t)$ | $k_A(\cdot, a_t)$ | $k_H(\cdot, h_t)$ |
| covariance operators | $\mathcal{C}_{OO}$ | $\mathcal{C}_{AA}$ | $\mathcal{C}_{HH}$ |
| feature maps of realizations | $\psi_t^o := \phi_{\mathbb{O}}(o_t)$ | $\psi_t^a := \phi_{\mathbb{A}}(a_t)$ | $\psi_t^h := \phi_{\mathbb{H}}(h_t)$ |
| feature maps of random variables | $\psi_t^O := \phi_{\mathbb{O}}(O_t)$ | $\psi_t^a := \phi_{\mathbb{A}}(o_t)$ | $\psi_t^h := \phi_{\mathbb{H}}(h_t)$ |
| RKHS | $\mathcal{H}_{\mathbb{O}}$ | $\mathcal{H}_{\mathbb{A}}$ | $\mathcal{H}_{\mathbb{H}}$ |

Table 7: Notations in the Main Text

| Notations | Meaning |
|---|---|
| $\mathcal{C}_{O\|AH}$ | general embedding operator |
| $\mathcal{C}_{O^i\|A^j H^j}$ | embedding operator that models object $j$'s effect to $i$'s observation |
| $\mathcal{C}_{O^i\|H^j}, \mathcal{C}_{O^i\|A^j}$ | the disentangled embedding operators |
| $\mathcal{C}_{O^i\|H}$ | shared embedding operator |
| $\mathcal{E}$ | binary adjacency matrix |
| $\mathcal{E}(i)$ | (inclusive) neighborhood |
| $\mathcal{F}$ | unknown transition function for multi-object dynamics |
| $f$ | pair-wise negative potential energy |
| $f_{\mathcal{V}}, f_{\mathcal{E}}$ | neural networks in message passing GNN |
| $G$ | kernelized energy function |
| $\mathcal{J}$ | cost function defined over graph representations |
| $P(O_t\|A_t = a_t, H_t = h_t)$ | conditional distribution given action $a_t$ and history $h_t$ |
| $Q_1, Q_2$ | positive-definite matrices |
| $\mathcal{V}_t^a$ | action of graph at time step $t$ |
| $\mathcal{V}_t^o$ | observation of graph at time step $t$ |
| $\alpha$ | Gibbs-Boltzmann weight |
| $(\sigma, \lambda, \kappa)$ | hyperparameters of kernel potentials |
| $\|\cdot\|_1$ | $1-$norm |
| $\|\cdot\|_2$ | $2-$norm |
| $\|\cdot\|_{\mathcal{H}}$ | norm over Hilbert space |
| $\|\cdot\|_{\mathrm{HS}}$ | Hilbert-Schmidt norm |

# B    NAVIGATION FOR READING

To help the reader navigate the Appendix, we briefly summarize its structure and content as follows:

- **Preliminaries: Basics of RKHS in C.** We begin with a short introduction of kernel methods and reproducing kernel Hilbert spaces (RKHS), which provides the necessary background for our embedding framework.

- **Theoretical Foundations of GCE in D.** We then construct the Graph Controllable Embedding (GCE) under the Hilbert space embedding framework, presenting the **Tensor**, **Dense**, and **Hom+Mean** forms. Each embedding is also interpreted from a probabilistic perspective.

- **Discussion of Various Embeddings in E.** Next, we compare the four forms: **Tensor**, **Dense**, **Hom**, and **Hom+Mean** in terms of their properties, sample complexity, and computational trade-offs. This section also provides explicit derivations and calculations.

- **Theoretical Analysis in F.** After that, we present formal proofs, including probabilistic consistency, convergence results, and sample complexity analysis for different embedding forms.

- **Experimental Details in I.** Finally, we provide additional experimental details, implementation setups, and extra results that complement the main text.

This organization mirrors the logical flow from foundations, to construction, to comparison, to theoretical analysis, and finally to empirical support. Readers may choose to follow the entire sequence or jump to the section most relevant to their interests.

## C PRELIMINARIES: BASICS OF RKHS

In this section, by largely following the research (Fukumizu et al., 2013), the Hilbert spaces are assumed to be separable and dense in $L^2$ space.

### C.1 POSITIVE DEFINITE KERNELS

Let $\mathbb{X}$ be a set, and consider a symmetric function $k : \mathbb{X} \times \mathbb{X} \to \mathbb{R}$, known as a positive definite kernel, satisfying the condition:

$$\sum_{i,j=1}^{n} c_i c_j k(x_i, x_j) \geq 0,$$

for any finite points $x_1, \ldots, x_n \in \mathbb{X}$ and real coefficients $c_1, \ldots, c_n$. The matrix $\left(k(x_i, x_j)\right)_{i,j=1}^{n}$ is called the covariance operator.

By the Moore-Aronszajn theorem (Aronszajn, 1950), every positive definite kernel on $\mathbb{X}$ defines a unique reproducing kernel Hilbert space (RKHS) $\mathcal{H}$ with an inner product structure $\langle \cdot, \cdot \rangle$, comprising functions on $\mathbb{X}$ such that:

1. For any $x \in \mathbb{X}$, $k(\cdot, x) \in \mathcal{H}$.
2. The span $\text{Span}\{k(\cdot, x) \mid x \in \mathbb{X}\}$ is dense in $\mathcal{H}$.
3. For all $x \in \mathbb{X}$ and $f \in \mathcal{H}$, $\langle f, k(\cdot, x) \rangle = f(x)$.

The RKHS $\mathcal{H}$ is thus characterized by the kernel $k$, with $k(\cdot, x)$ serving as the reproducing kernel. A kernel $k$ is termed *bounded* if there exists $M > 0$ such that $k(x, x) \leq M$ for all $x \in \mathbb{X}$.

### C.2 KERNEL MEANS AND COVARIANCE OPERATORS

Let $(\mathbb{X}, \mathcal{B}_{\mathbb{X}})$ be a measurable space and $X$ a random variable taking values in $\mathbb{X}$ with distribution $P_X$. Suppose $k$ is a measurable positive definite kernel on $\mathbb{X}$ such that $\mathbb{E}[k(X, X)] < \infty$. The associated RKHS is denoted by $\mathcal{H}$.

The kernel mean embedding $m_X$ of $X$ in $\mathcal{H}$ is defined by:

$$m_X = \mathbb{E}[k(\cdot, X)] = \int k(\cdot, x) \, dP_X(x).$$

Using the reproducing property, the kernel mean satisfies:

$$\langle f, m_X \rangle = \mathbb{E}[f(X)], \quad \text{for all } f \in \mathcal{H}.$$

Now consider two measurable spaces $(\mathbb{X}, \mathcal{B}_{\mathbb{X}})$ and $(\mathbb{Y}, \mathcal{B}_{\mathbb{Y}})$, and let $(X, Y)$ be a pair of random variables taking values in $\mathbb{X}$ and $\mathbb{Y}$, respectively, with joint distribution $P_{XY}$ over $\mathbb{X} \times \mathbb{Y}$. Let $k_X$ and $k_Y$ be measurable positive definite kernels with associated RKHSs $\mathcal{H}_{\mathbb{X}}$ and $\mathcal{H}_{\mathbb{Y}}$, satisfying $\mathbb{E}[k_X(X, X)] < \infty$ and $\mathbb{E}[k_Y(Y, Y)] < \infty$.

The (uncentered) covariance operator $\mathcal{C}_{YX} : \mathcal{H}_{\mathbb{X}} \to \mathcal{H}_{\mathbb{Y}}$ is a linear operator defined as:

$$\langle g, \mathcal{C}_{YX} f \rangle_{\mathcal{H}_Y} = \mathbb{E}[f(X)g(Y)],$$

for all $f \in \mathcal{H}_{\mathbb{X}}$ and $g \in \mathcal{H}_{\mathbb{Y}}$. The integral representations are given by:

$$(\mathcal{C}_{YX} f)(y) = \int k_Y(y, \tilde{y}) f(\tilde{x}) \, dP_{XY}(\tilde{x}, \tilde{y}), \quad (\mathcal{C}_{XX} f)(x) = \int k_X(x, \tilde{x}) f(\tilde{x}) \, dP_{XY}(\tilde{x}).$$

Under the bounded assumption, the kernel mean and covariance operators are well-defined for a given probability distribution. The (cross-)covariance operators can be written as

$$\mathcal{C}_{YX} := \mathbb{E}[k_Y(\cdot, Y) \otimes k_X(\cdot, X)], \quad \mathcal{C}_{XX} := \mathbb{E}[k_X(\cdot, X) \otimes k_X(\cdot, X)].$$

The conditional mean embedding of $P_{XY}(Y | X = x)$ in $\mathcal{H}_{\mathbb{Y}}$ is then

$$m_{Y|x} := \mathbb{E}[k_Y(\cdot, Y) | X = x] = \mathcal{C}_{YX} \mathcal{C}_{XX}^{-1} k_X(\cdot, x) = \mathcal{C}_{Y|X} k_X(\cdot, x), \tag{13}$$

where $\mathcal{C}_{Y|X} : \mathcal{H}_{\mathbb{X}} \to \mathcal{H}_{\mathbb{Y}}$ is the conditional embedding operator and $\mathcal{C}_{XX}^{-1}$ is the inverse (or regularized inverse) of the covariance operator of $\mathcal{C}_{XX}$.

# D    THEORETICAL FOUNDATION OF GCE

Following the definitions from Appendix C, we first introduce the essential theoretical foundation of Hilbert space embedding, followed by the construction of GCE.

## D.1    INTRODUCTION TO KERNEL BAYES' RULE AND HILBERT SPACE EMBEDDING

Before presenting the main theoretical analysis, we recall the kernel mean map and its application to Bayes' rule, known as the **Kernel Bayes' Rule (KBR)** (Fukumizu et al., 2011).

Let $(X, Y, Z)$ be random variables taking values in measurable spaces $(\mathbb{X}, \mathcal{B}_\mathbb{X})$, $(\mathbb{Y}, \mathcal{B}_\mathbb{Y})$, and $(\mathbb{Z}, \mathcal{B}_\mathbb{Z})$, respectively, with joint distribution $P$. We seek a kernel-based analogue of the conditional distribution

$$P(Z \mid Y = y, X = x) = \frac{P(Z, Y = y \mid X = x)}{P(Y = y \mid X = x)},$$

where $x \in \mathbb{X}$, $y \in \mathbb{Y}$, and $z \in \mathbb{Z}$ are realizations of random variables $X$, $Y$, and $Z$, respectively. To obtain this kernel representation, we require the kernel mean embedding of the *conditional joint distribution* $P(Z, Y \mid X = x)$.

Let $k_X, k_Y, k_Z$ be measurable positive definite kernels on $\mathbb{X}$, $\mathbb{Y}$, and $\mathbb{Z}$, with associated RKHSs $\mathcal{H}_\mathbb{X}, \mathcal{H}_\mathbb{Y}$, and $\mathcal{H}_\mathbb{Z}$. The canonical feature maps for realizations and random variables as

$$\phi_\mathbb{X}(x) = k_X(\cdot, x), \quad \phi_\mathbb{Y}(y) = k_Y(\cdot, y), \quad \phi_\mathbb{Z}(z) = k_Z(\cdot, z),$$

and

$$\phi_\mathbb{X}(X) = k_X(\cdot, X), \quad \phi_\mathbb{Y}(Y) = k_Y(\cdot, Y), \quad \phi_\mathbb{Z}(Z) = k_Z(\cdot, Z).$$

We assume each kernel is *characteristic*, ensuring that the kernel mean embedding defines an injective mapping from probability measures over the domain to elements in the corresponding RKHS (Sriperumbudur et al., 2011).

**Step 1.** The conditional joint distribution $P(Z, Y \mid X = x)$ admits the kernel mean embedding

$$m_{ZY|x} := \mathbb{E}\left[\phi_\mathbb{Z}(Z) \otimes \phi_\mathbb{Y}(Y) \mid X = x\right] \in \mathcal{H}_\mathbb{Z} \otimes \mathcal{H}_\mathbb{Y}.$$

Equivalently, it can be expressed via the conditional embedding operator $\mathcal{C}_{ZY|X} : \mathcal{H}_\mathbb{X} \to \mathcal{H}_\mathbb{Z} \otimes \mathcal{H}_\mathbb{Y}$:

$$m_{ZY|x} = \mathcal{C}_{ZY|X}\, \phi_\mathbb{X}(x) = \mathcal{C}_{ZYX}\, \mathcal{C}_{XX}^{-1}\, \phi_\mathbb{X}(x), \tag{14}$$

where

$$\mathcal{C}_{ZYX} := \mathbb{E}\left[\phi_\mathbb{Z}(Z) \otimes \phi_\mathbb{Y}(Y) \otimes \phi_\mathbb{X}(X)\right]$$

is the (uncentered) cross-covariance operator between $(Z, Y)$ and $X$, and $\mathcal{C}_{XX}$ is the covariance operator of $X$.

**Step 2.** Once we have the conditional joint embedding $m_{ZY|x}$, KBR gives the conditional mean embedding of $Z$ give $Y = y$ and $X = x$ as

$$m_{Z|y,x} = \mathbb{E}[\phi_\mathbb{Z}(Z) \mid Y = y, X = x] = \mathcal{C}_{ZY|x}\mathcal{C}_{YY|x}^{-1}\phi_\mathbb{Y}(y) \tag{15}$$

where

$$\mathcal{C}_{ZY|x} := \mathbb{E}\left[\phi_\mathbb{Z}(Z) \otimes \phi_\mathbb{Y}(Y) \mid X = x\right], \quad \mathcal{C}_{YY|x} := \mathbb{E}\left[\phi_\mathbb{Y}(Y) \otimes \phi_\mathbb{Y}(Y) \mid X = x\right].$$

Expressing these conditional operators in terms of unconditional covariance operators yields

$$\mathcal{C}_{ZY|x} = \mathcal{C}_{ZYX}\mathcal{C}_{XX}^{-1}\phi_\mathbb{X}(x), \quad \mathcal{C}_{YY|x} = \mathcal{C}_{YYX}\mathcal{C}_{XX}^{-1}\phi_\mathbb{X}(x),$$

such that

$$\mathbb{E}[\phi_\mathbb{Z}(Z) \mid Y = y, X = x] = \left(\mathcal{C}_{ZYX}\mathcal{C}_{XX}^{-1}\phi_\mathbb{X}(x)\right)\left(\mathcal{C}_{YYX}\mathcal{C}_{XX}^{-1}\phi_\mathbb{X}(x)\right)^{-1}\phi_\mathbb{Y}(y). \tag{16}$$

**Two Steps in One Formula.** Observing the properties of Equations 14 and 15, the conditional embedding of $Z$ given joint distribution $X$ and $Y$ can be written as a single formula in the tensor space:

$$\mathbb{E}[\phi_\mathbb{Z}(Z) \mid Y = y, X = x] = \mathcal{C}_{Z|YX}[\phi_\mathbb{X}(x) \otimes \phi_\mathbb{Y}(y)] = \mathcal{C}_{ZYX}\mathcal{C}_{(YX)(YX)}^{-1}[\phi_\mathbb{X}(x) \otimes \phi_\mathbb{Y}(y)] \tag{17}$$

where tensor covariance operator $\mathcal{C}_{(YX)(YX)}$ and conditional embedding operator $\mathcal{C}_{Z|YX}$ are

$$\mathcal{C}_{(YX)(YX)} := \mathbb{E}[(\phi_{\mathbb{Y}}(Y) \otimes \phi_{\mathbb{X}}(X)) \otimes (\phi_{\mathbb{Y}}(Y) \otimes \phi_{\mathbb{X}}(X))]$$

and

$$\mathcal{C}_{Z|YX} = \mathcal{C}_{ZYX} \mathcal{C}_{(YX)(YX)}^{-1}.$$

**Remark 1.** *The tensor formulation in Equation 17 is equivalent to apply KBR in two steps ($X \rightarrow (Z, Y)$, then $Y \rightarrow Z$), but it avoids intermediate conditioning and directly represents $P(Z \mid Y, X)$ in one operator by $\mathcal{C}_{Z|YX}$.*

## D.2 Hilbert Space Embedding of Stochastic Dynamics

In this paper, our goal is to embed the conditional probability

$$P(O_t | A_t = a_t, H_t = h_t) \tag{18}$$

of the future observation $O_t$ given history $h_t \in \mathbb{H}$ and $a_t \in \mathbb{A}$. Following Definition 1, we denote the kernel feature mappings of their realizations and random variables in their respective RKHSs $\mathcal{H}_{\mathbb{H}}$, $\mathcal{H}_{\mathbb{A}}$ and $\mathcal{H}_{\mathbb{O}}$ as

$$\psi_t^h := \phi_{\mathbb{H}}(h_t), \quad \psi_t^a := \phi_{\mathbb{A}}(a_t), \quad \psi_t^O := \phi_{\mathbb{O}}(o_t), \tag{19}$$

and

$$\psi_t^H := \phi_{\mathbb{H}}(H_t), \quad \psi_t^A := \phi_{\mathbb{A}}(A_t), \quad \psi_t^O := \phi_{\mathbb{O}}(O_t). \tag{20}$$

According the properties of KBR (see Equation 17 in Appendix D.1), the conditional mean embedding of $O_t$ given $(h_t, a_t)$ is expressed directly in tensor form as

$$\mathbb{E}[\psi_t^O \mid A_t = a_t, H_t = h_t] = \mathcal{C}_{OAH} \mathcal{C}_{(AH)(AH)}^{-1} [\psi_t^h \otimes \psi_t^a] = \mathcal{C}_{O|AH}[\psi_t^h \otimes \psi_t^a], \tag{21}$$

where $\mathcal{C}_{OAH}$ and $\mathcal{C}_{(AH)(AH)}$ are denoted as

$$\mathcal{C}_{OAH} = \mathbb{E}[\psi_t^O \otimes \psi_t^A \otimes \psi_t^H],$$
$$\mathcal{C}_{(AH)(AH)} = \mathbb{E}[(\psi_t^A \otimes \psi_t^H) \otimes (\psi_t^A \otimes \psi_t^H)].$$

The operator $\mathcal{C}_{O|AH} : \mathcal{H}_{\mathbb{H}} \otimes \mathcal{H}_{\mathbb{A}} \rightarrow \mathcal{H}_{\mathbb{O}}$ is the conditional embedding operator, mapping joint history-action features to observation features in the RKHS. Figure 1 illustrates the core idea of Hilbert space embedding of stochastic dynamics. **Left:** The nonlinear stochastic dynamics are represented by the evolving conditional distributions $P(O_t | A_t = a_t, H_t = h_t)$, shown as red curves. Red dots indicate possible samples from these distributions, and black dots denote realized observations. **Right:** After embedding into an RKHS, the dynamics become linear under the conditional embedding operator $\mathcal{C}_{O|AH}$, with red dots indicating the corresponding conditional mean estimates in the RKHS.

## D.3 Derivation of GCE Based on Hilbert Space Embedding

The Hilbert space embedding of stochastic dynamics has been established in Appendix D.2. We now show how the GCE can be constructed via Hilbert space embedding.

Consider a stochastic system of $N$ interacting objects, where each object $i$ is associated with an observation $\mathbb{O}^i$, action space $\mathbb{A}^i$, and history space $\mathbb{H}^i$. At time $t$, object $i$ has history $h_t^i \in \mathbb{H}^i$, action $a_t^i \in \mathbb{A}^i$, and future observation $o_t^i \in \mathbb{O}^i$. Also, we denote the $H_t^i$, $A_t^i$ and $O_t^i$ as the corresponding random variables. Applying Definition 1, the corresponding RKHS feature maps are

$$\psi_t^{h,i} := \phi_{\mathbb{H}^i}(h_t^i), \quad \psi_t^{a,i} := \phi_{\mathbb{A}^i}(a_t^i), \quad \psi_t^{o,i} := \phi_{\mathbb{O}^i}(o_t^i),$$

in the corresponding RKHSs $\mathcal{H}_{\mathbb{H}^i}$, $\mathcal{H}_{\mathbb{A}^i}$ and $\mathcal{H}_{\mathbb{O}^i}$, respectively.

Recalling the principles of KBR in Appendix D.1, the joint conditional distribution of all $N$ objects can be written as

$$P(\{O_t^i\}_{i=1}^N \mid \{A_t^j = a_t^j, H_t^j = h_t^j\}_{j=1}^N)$$

$$\overset{\text{(Bayes' rule)}}{=} \frac{P\big(\{O_t^i, A_t^j = a_t^j, H_t^j = h_t^j\}_{i,j=1}^N\big)}{P\big(\{A_t^j = a_t^j, H_t^j = h_t^j\}_{j=1}^N\big)}$$

$$\overset{\text{(object-wise factorization)}}{=} \prod_{i=1}^N \frac{P\big(O_t^i, \{A_t^j = a_t^j, H_t^j = h_t^j\}_{j=1}^N\big)}{P\big(\{A_t^j = a_t^j, H_t^j = h_t^j\}_{j=1}^N\big)}.$$

(22)

In the RKHS setting, this corresponds to a *global* conditional embedding operator $\mathcal{C}_{O|AH}$ acting on the pair-wise joint history–action feature $[\psi_t^{h,1} \otimes \psi_t^{a,1}, \ldots, \psi_t^{h,N} \otimes \psi_t^{a,N}]$. While this global form is valid, it does not exploit the relational structure of the interaction graph, where each object interacts only with a subset of other objects (e.g., neighbors).

**Decomposition into pair-wise contributions.** From the tensorized formulation in Equation 21, the global conditional embedding of all objects can be written as

$$\mathbb{E}\left[[\psi_t^{O,1}, \cdots \psi_t^{O,N}] \mid \{A_t^j = a_t^j, H_t^j = h_t^j\}_{j=1}^N\right] = \mathcal{C}_{O|AH} \begin{bmatrix} \psi_t^{h,1} \otimes \psi_t^{a,1} \\ \vdots \\ \psi_t^{h,N} \otimes \psi_t^{a,N} \end{bmatrix},$$

where $[\psi_t^{O,1}, \cdots \psi_t^{O,N}]$ stacks the observation features of all $N$ objects and $\mathcal{C}_{O|AH}$ is a *global* operator from the joint history–action RKHS to the joint observation RKHS.

By the linearity of $\mathcal{C}_{O|AH}$, this global operator can be represented in block form:

$$\mathcal{C}_{O|AH} = \begin{bmatrix} \mathcal{C}_{O^1|A^1 H^1} & \cdots & \mathcal{C}_{O^1|A^N H^N} \\ \vdots & \ddots & \vdots \\ \mathcal{C}_{O^N|A^1 H^1} & \cdots & \mathcal{C}_{O^N|A^N H^N} \end{bmatrix},$$

where each block $\mathcal{C}_{O^i|A^j H^j}$ maps the history–action feature of object $j$ to its contribution in the conditional expectation of object $i$'s observation.

For the $i$-th object, the $i$-th row of this block matrix yields:

$$\mathbb{E}[\psi_t^{O,i} \mid \{a_t^j, h_t^j\}_{j=1}^N] = [\mathcal{C}_{O^i|A^1 H^1}, \ldots, \mathcal{C}_{O^i|A^N H^N}] \begin{bmatrix} \psi_t^{h,1} \otimes \psi_t^{a,1} \\ \vdots \\ \psi_t^{h,N} \otimes \psi_t^{a,N} \end{bmatrix}$$

$$= \sum_{j \in \mathcal{E}(i)} \mathcal{C}_{O^i|A^j H^j}[\psi_t^{h,j} \otimes \psi_t^{a,j}].$$

(23)

Interactions occur only with neighbors $j \in \mathcal{E}(i)$; for non-neighbors ($j \notin \mathcal{E}(i)$), we set $\mathcal{C}_{O^i|A^j H^j} = 0$. This induces a localized form of the global conditional embedding, fully aligned with the entire topology.

**Interpretation.** In this view, Equation 23 is a *factorization* of the global conditional embedding operator induced by KBR, where only the operators $\mathcal{C}_{O^i|A^j H^j}$ corresponding to edges in the interaction graph are active. This factorization preserves the theoretical grounding of the Hilbert space embedding while enabling efficient modeling and control through localized graph structure.

**Limitations.** While the embedding offers a powerful representation of distributions, the tensorized formulation in Equation 23 still suffers from two limitations:

- **Sample complexity.** Based on the random matrix theory (Tao, 2012), the convergence of the covariance operator grows significantly with the modes of the tensor.

- **Optimization.** The intertwining of history and action features within the tensor product complicates the optimization of sequential actions. Since control algorithms like LQR/iLQR assume additive disentanglement between history and action features, the tensorized embedding in Equation 23 violates this assumption.

To concretize the controllable embedding within the Hilbert space embedding framework, the tensorized features in Equation 23 must be disentangled to meet the requirements of controllable embedding.

### D.4 DISENTANGLEMENT OF HISTORY AND ACTION FEATURES FOR OPTIMIZATION

Building on previous studies (Altun et al., 2012; Zhang et al., 2008), an efficient approach to address the aforementioned problems involves simplifying the tensorized formulation into a concatenated feature representation, $[\psi_t^{h,j} \| \psi_t^{a,j}]$. This decomposition simplifies the problem by factorizing the action and history random variables into two product conditional probability distributions, as exemplified in (Song et al., 2009; Altun et al., 2012). By adopting the concatenated feature representation, Equation 23 can be simplified as:

$$
\begin{aligned}
\mathbb{E}\big[\psi_t^{O,i} \,\big|\, \{a_t^j, h_t^j\}_{j=1}^N\big] &= \big[\mathcal{C}_{O^i|A^1 H^1}, \cdots, \mathcal{C}_{O^i|A^N H^N}\big] \times \big[\psi_t^{h,1}\|\psi_t^{a,1}, \ldots, \psi_t^{h,N}\|\psi_t^{a,N}\big]^\top \\
&= \sum_{j\in\mathcal{E}(i)} \mathcal{C}_{O^i|A^j H^j}\big[\psi_t^{h,j}\|\psi_t^{a,j}\big]. \\
&\approx \sum_{j\in\mathcal{E}(i)} \mathcal{C}_{O^i|H^j}\psi_t^{h,j} + \mathcal{C}_{O^i|A^j}\psi_t^{a,j},
\end{aligned}
\tag{24}
$$

where the linear conditional operator is simplified to the form $[\mathcal{C}_{O^i|H^j}, \mathcal{C}_{O^i|A^j}]$. Here, this formulation in Equation 24 can be regarded as an approximation of Equation 23 by neglecting the high-order interaction of history and action features. The formulation directly brings two advantages:

- **Reduced Sample Complexity.** Based on the concentration inequalities in random matrices (e.g., matrix Hoeffding and Bernstein inequalities (Van Handel, 2014)), the sample complexity can be significantly reduced with a lower dimension. Then $3-$mode tensor operator $\mathcal{C}_{O|AH}$ in Equation 21 can lead to a higher sample complexity.
- **Simple Optimization.** When optimizing the quadratic cost function constrained by Equation 3, the only undetermined variable is the fully decomposed action feature $\psi_t^a$ instead of the entangled feature $\psi_t^h \otimes \psi_t^a$.

### D.5 EFFICIENT EMBEDDING BY MEAN FIELD APPROXIMATION

In systems with multiple interacting objects, each object directly influences its neighboring objects through their relational structure. For multi-object systems with many objects and dense edges, modeling all these pair-wise interactions explicitly becomes computationally expensive, scaling with the number of edges in the graph. The core idea of using mean field approximation is to simplify the modeling of complex interactions in systems with many interacting components by approximating the influence of all other components on a given component as an average or "mean" effect, rather than modeling every pair-wise interaction explicitly. Thanks to the disentangled structure in Equation 24, we can separately model how history influences the future observation probability and how actions affect the observation, combining them multiplicatively in the joint conditional distribution.

To disentangle history and action effects, we assume conditional independence between actions and histories given the observation ($A \perp\!\!\!\perp H \mid O$). This allows us to factorize the conditional distribution into pair-wise history potentials and an action-dependent likelihood term as

$$
\begin{aligned}
&P(O_t^i \mid \{H_t^j = h_t^j\}_{j\in\mathcal{E}(i)}, \{A_t^j = a_t^j\}_{j\in\mathcal{E}(i)}) \\
&\overset{\text{(pair-wise factorization on history effect)}}{\propto} \left[\prod_{j\in\mathcal{E}(i)} \Phi_h^{i,j}\Big(O_t^i, H_t^j = h_t^j\Big)\right] \cdot P\Big(\{A_t^j = a_t^j\}_{j\in\mathcal{E}(i)}|O_t^i\Big),
\end{aligned}
\tag{25}
$$

where

- $\Phi_h^{i,j} \geq 0$ is a *pair-wise history potential* encoding the interaction between object $i$'s observation and object $j$'s history.

- $P\left(\{A_t^j = a_t^j\}_{j \in \mathcal{E}(i)} | O_t^i\right)$ is a *disentangled action-dependent likelihood* term, reflecting the statistical dependency between the neighbors' actions and object $i$'s observation.

Based on the disentangled probability in Equation 25, the naive assumption of equally weighted potentials $\Phi^{i,j}$ for all $j$ is generally invalid, rendering the representation in (Li et al., 2020) *misspecified*.

**Mean field normalization for history.**   Based on Equation 25, only the history part requires mean field normalization. We convert $\Phi_h^{i,j}$ into normalized influence coefficients by taking its expectation under the current mean field approximation $q(O_t^i)$:

$$\alpha_h^{i,j} \;=\; \frac{\exp\left(\mathbb{E}_{q(O_t^i)}[\log \Phi_h^{i,j}(O_t^i, h_t^j)]\right)}{\sum_{k \in \mathcal{E}(i)} \exp\left(\mathbb{E}_{q(O_t^i)}[\log \Phi_h^{i,k}(O_t^i, h_t^k)]\right)}, \quad \text{with} \quad \sum_{j \in \mathcal{E}(i)} \alpha_h^{i,j} = 1. \qquad (26)$$

Here the expectation $\mathbb{E}_{q(O_t^i)}[\cdot]$ (under variational distribution $q(O_t^i)$) ensures that the influence weights do not directly depend on the unknown variable $O_t^i$, but rather on its current mean field approximation.

**GCE with mean field approximation.**   Substituting these weights into the GCE formulation in Equation 24 and applying the principles of KBR yields:

$$\begin{aligned}
&\sum_{j \in \mathcal{E}(i)} \mathcal{C}_{O^i|H^j} \, \psi_t^{h,j} \;+\; \mathcal{C}_{O^i|A^j} \, \psi_t^{a,j} \\
=&\mathcal{C}_{O^i|H}\left(\sum_{j \in \mathcal{E}(i)} \alpha^{i,j} \, \psi_t^{h,j}\right) + \sum_{j \in \mathcal{E}(i)} \mathcal{C}_{O^i|A^j} \, \psi_t^{a,j}.
\end{aligned} \qquad (27)$$

Here, $\mathcal{C}_{O^i|H}$ is a shared conditional embedding operator used in the homogeneous case, while the non-uniformity of $\alpha^{i,j}$ directly reflects the unequal pair-wise potentials $\Phi^{i,j}$.

**Practical Computation.** Directly estimating the distribution $q(O_t^i)$ is infeasible, as the multi-object dynamics evolves in a continuous observation space. Instead, we approximate the influence weight $\alpha_t^{i,j}$ using the history $h_t^i = o_{t-1}^i$, which is temporally close to the future observation $o_t^i$. Since the forward dynamics are modeled in an RKHS, we compute the weights from the history features via a Gibbs measure:

$$\alpha_t^{i,j} \;=\; \frac{\exp\left(f(\psi_t^{h,i}, \psi_t^{h,j})\right)}{\sum\limits_{k \in \mathcal{E}(i)} \exp\left(f(\psi_t^{h,i}, \psi_t^{h,k})\right)}, \qquad (28)$$

where $f$ [3] is a negative pair-wise potential function between history features. The normalization in Equation 7 ensures $\sum_{j \in \mathcal{E}(i)} \alpha_t^{i,j} = 1$.

We consider the following four choices for $f$ to instantiate the pair-wise potential:

1. **Gaussian potential.** Let $\psi_t^{h,i}, \psi_t^{h,j} \in \mathbb{R}^{d_h}$. Define

$$f_{\text{RBF}}(\psi_t^{h,i}, \psi_t^{h,j}) \;=\; -\frac{\|\psi_t^{h,i} - \psi_t^{h,j}\|_2^2}{2\sigma^2},$$

   with bandwidth $\sigma > 0$. This corresponds to a positive definite Gaussian kernel and is robust to small perturbations.

---

[3]Here, $\exp\left(f(\cdot, \cdot)\right)$ can be regarded as practical proxy to measure $\exp\left(\mathbb{E}_{q(O_t^i)}[\log \Phi_h^{i,j}(\cdot, \cdot)]\right)$, this holds since there exist some functions that pull features to the observation space.

2. **Laplace potential.**

$$f_{\text{Lap}}(\psi_t^{h,i}, \psi_t^{h,j}) \;=\; -\frac{\|\psi_t^{h,i} - \psi_t^{h,j}\|_1}{\lambda},$$

with scale $\lambda > 0$. Compared to the quadratic potential energy in Gaussian kernel, the Laplace form induces heavier tails due to the nature of $L^1$ norm.

3. **von Mises–Fisher (vMF) potential.**

$$f_{\text{vMF}}\Big(\frac{\psi_t^{h,i}}{\|\psi_t^{h,i}\|_2}, \frac{\psi_t^{h,j}}{\|\psi_t^{h,j}\|_2}\Big) \;=\; \kappa \, \Big(\frac{\psi_t^{h,i}}{\|\psi_t^{h,i}\|_2}\Big)^\top \frac{\psi_t^{h,j}}{\|\psi_t^{h,j}\|_2},$$

with concentration $\kappa \geq 0$. This favors alignment on the unit sphere and is natural for directional embeddings.

4. **Neural potential.** Let $f_\theta : \mathbb{R}^{d_h} \times \mathbb{R}^{d_h} \to \mathbb{R}$ be a neural scorer:

$$f_\theta(\psi_t^{h,i}, \psi_t^{h,j}) \;=\; \text{MLP}_\theta\big([\psi_t^{h,i} \| \psi_t^{h,j}]\big),$$

where $[\cdot \| \cdot]$ denotes concatenation. This yields a flexible, learnable potential while preserving nonnegativity via the exponential link.

In all cases, the Gibbs normalization in Equation 7 ensures $\sum_{j \in \mathcal{E}(i)} \alpha_t^{i,j} = 1$. Hyperparameters $(\sigma, \lambda, \kappa)$ or the neural parameters $\theta$ can be learned by the joint training.

# E   DISCUSSION OF VARIOUS EMBEDDINGS IN GCE

This section introduces four embedding formulations used in our framework and evaluates their properties for controllable embedding design. We denote the dimensions of the history, action, and observation features by $d_h$, $d_a$, and $d_o$, respectively.

## E.1   FORMULATION OF VARIOUS EMBEDDINGS IN GCE

**Tensor Representation.**   Originating from the inherent structure of Kernel Bayes' Rule (KBR), the tensor representation directly follows Equation 21 and can be expressed as:

$$
\begin{bmatrix} \mathbb{E}\big[\psi_t^{O,1} \mid \{a_t^j, h_t^j\}_{j=1}^N\big] \\ \vdots \\ \mathbb{E}\big[\psi_t^{O,N} \mid \{a_t^j, h_t^j\}_{j=1}^N\big] \end{bmatrix} = \begin{bmatrix} \mathcal{C}_{O^1|A^1H^1} & \cdots & \mathcal{C}_{O^1|A^NH^N} \\ \vdots & \ddots & \vdots \\ \mathcal{C}_{O^N|A^1H^1} & \cdots & \mathcal{C}_{O^N|A^NH^N} \end{bmatrix} \begin{bmatrix} \psi_t^{h,1} \otimes \psi_t^{a,1} \\ \vdots \\ \psi_t^{h,N} \otimes \psi_t^{a,N} \end{bmatrix}.
\tag{29}
$$

Each sub-operator $\mathcal{C}_{O^i|A^jH^j} \in \mathbb{R}^{d_o \times (d_h \times d_a)}$ acts on the tensorized feature $\psi_t^{h,j} \otimes \psi_t^{a,j} \in \mathbb{R}^{(d_h \times d_a) \times 1}$, entangling history and action information. This entanglement makes it not suitable for controllable embedding, as the representation is neither locally nor globally linear. The computational complexity is $\mathcal{O}(N^2 d_o d_h d_a)$ per forward pass, making it computationally expensive for $N$ with a large value. For simplicity, we refer to this form as **Tensor**.

**Dense Representation.**   The dense representation removes the tensor product entanglement by decomposing the history and action features, as in Equation 24:

$$
\begin{bmatrix} \mathbb{E}\big[\psi_t^{O,1} \mid \{a_t^j, h_t^j\}_{j=1}^N\big] \\ \vdots \\ \mathbb{E}\big[\psi_t^{O,N} \mid \{a_t^j, h_t^j\}_{j=1}^N\big] \end{bmatrix}
$$
$$
= \begin{bmatrix} \mathcal{C}_{O^1|H^1} & \cdots & \mathcal{C}_{O^1|H^N} \\ \vdots & \ddots & \vdots \\ \mathcal{C}_{O^N|H^1} & \cdots & \mathcal{C}_{O^N|H^N} \end{bmatrix} \begin{bmatrix} \psi_t^{h,1} \\ \vdots \\ \psi_t^{h,N} \end{bmatrix} + \begin{bmatrix} \mathcal{C}_{O^1|A^1} & \cdots & \mathcal{C}_{O^1|A^N} \\ \vdots & \ddots & \vdots \\ \mathcal{C}_{O^N|A^1} & \cdots & \mathcal{C}_{O^N|A^N} \end{bmatrix} \begin{bmatrix} \psi_t^{a,1} \\ \vdots \\ \psi_t^{a,N} \end{bmatrix}.
\tag{30}
$$

Here, $\mathcal{C}_{O^i|H^j} \in \mathbb{R}^{d_o \times d_h}$ and $\mathcal{C}_{O^i|A^j} \in \mathbb{R}^{d_o \times d_a}$ making the representation locally linear in each feature type. The complexity is $\mathcal{O}(N^2(d_o d_h + d_o d_a))$ per forward pass, lower than the tensor form but is still quadratic in $N$ if with dense connections. For simplicity, we name this form as **Dense**.

**Homogeneous Form with Uniform Weight.**   This form generalizes the compositional Koopman operator (Li et al., 2020), assuming equal-weight influence from all neighbors:

$$
\begin{bmatrix} \mathbb{E}\big[\psi_t^{O,1} \mid \{a_t^j, h_t^j\}_{j=1}^N\big] \\ \vdots \\ \mathbb{E}\big[\psi_t^{O,N} \mid \{a_t^j, h_t^j\}_{j=1}^N\big] \end{bmatrix}
$$
$$
= \begin{bmatrix} \mathcal{C}_{O^1|H^1} & \cdots & \mathcal{C}_{O^1|H^N} \\ \vdots & \ddots & \vdots \\ \mathcal{C}_{O^N|H^1} & \cdots & \mathcal{C}_{O^N|H^N} \end{bmatrix} \begin{bmatrix} \psi_t^{h,1} \\ \vdots \\ \psi_t^{h,N} \end{bmatrix} + \begin{bmatrix} \mathcal{C}_{O^1|A^1} & \cdots & \mathcal{C}_{O^1|A^N} \\ \vdots & \ddots & \vdots \\ \mathcal{C}_{O^N|A^1} & \cdots & \mathcal{C}_{O^N|A^N} \end{bmatrix} \begin{bmatrix} \psi_t^{a,1} \\ \vdots \\ \psi_t^{a,N} \end{bmatrix}.
\tag{31}
$$

The sub-operators have the same dimensions as in the dense form, and the complexity remains $\mathcal{O}(N^2(d_o d_h + d_o d_a))$. In the setting of (Li et al., 2020), $\mathcal{C}_{O^i|H^j}$ are shared for the entire graph. However, the uniform-weight assumption is generally inaccurate in probabilistic settings (see Equation 25). We call this form **Hom**.

**Homogeneous Form with Mean Field Approximation.** To address the limitation of uniform weighting, we propose to use a mean field approximation with adaptive Boltzmann–Gibbs weights:

$$
\begin{bmatrix}
\mathbb{E}\big[\psi_t^{O,1} \,\big|\, \{a_t^j, h_t^j\}_{j=1}^N\big] \\
\vdots \\
\mathbb{E}\big[\psi_t^{O,N} \,\big|\, \{a_t^j, h_t^j\}_{j=1}^N\big]
\end{bmatrix}
$$
$$
=
\begin{bmatrix}
\mathcal{C}_{O^1|H} \\
\vdots \\
\mathcal{C}_{O^N|H}
\end{bmatrix}
\odot
\begin{bmatrix}
\sum_{j\in\mathcal{E}(1)} \alpha_t^{1,j}\psi_t^{h,j} \\
\vdots \\
\sum_{j\in\mathcal{E}(N)} \alpha_t^{N,j}\psi_t^{h,j}
\end{bmatrix}
+
\begin{bmatrix}
\mathcal{C}_{O^1|A^1} & \cdots & \mathcal{C}_{O^1|A^N} \\
\vdots & \ddots & \vdots \\
\mathcal{C}_{O^N|A^1} & \cdots & \mathcal{C}_{O^N|A^N}
\end{bmatrix}
\times
\begin{bmatrix}
\psi_t^{a,1} \\
\vdots \\
\psi_t^{a,N}
\end{bmatrix},
\tag{32}
$$

where $\odot$ denotes the element-wise product with shared operator $\mathcal{C}_{O^i|H}$. We refer to this form as **Hom + Mean**. This form offers:

1. **Lower complexity**: The history term costs $\mathcal{O}(Nd_od_h)$ and the action term costs $\mathcal{O}(N^2 d_o d_a)$, reducing cost compared to dense and tensor forms.
2. **Fewer parameters**: Requires fewer learnable parameters than dense or tensor forms.
3. **Adaptive weighting**: Captures non-uniform neighbor influence without quadratic parameter growth.

**Comparison of Embeddings.** Table 8 summarizes the properties of the four embeddings.

Table 8: Comparison of embedding formulations in GCE.

| Embedding | Complexity | Parameters | Adaptive weights |
|---|---|---|---|
| **Tensor** | $\mathcal{O}(N^2 d_o d_h d_a)$ | Quadratic | Yes (implicit) |
| **Dense** | $\mathcal{O}(N^2(d_o d_h + d_o d_a))$ | Quadratic | Yes (implicit) |
| **Hom** | $\mathcal{O}(N^2(d_o d_h + d_o d_a))$ | Constant | No (uniform) |
| **Hom + Mean** | $\mathcal{O}(Nd_o d_h + N^2 d_o d_a)$ | Constant | Yes (explicit) |

### E.2 EMPIRICAL ESTIMATION OF CONDITIONAL EMBEDDING OPERATORS

The empirical computation of the four forms in GCE is listed as follows. Given an i.i.d. sample $\big\{(o_t^i, a_t^1, \ldots, a_t^N, h_t^1, \ldots, h_t^N)\big\}_{t=1}^T$. The empirical covariance and cross-covariance operators are estimated as:

$$
\hat{\mathcal{C}}_{O^i H^j} = \frac{1}{T}\sum_{t=1}^T \psi_t^{o,i} \otimes \psi_t^{h,j}, \quad j \in [N],
$$

$$
\hat{\mathcal{C}}_{O^i A^j} = \frac{1}{T}\sum_{t=1}^T \psi_t^{o,i} \otimes \psi_t^{a,j}, \quad j \in [N],
$$

$$
\hat{\mathcal{C}}_{H^j H^j} = \frac{1}{T}\sum_{t=1}^T \psi_t^{h,j} \otimes \psi_t^{h,j}, \quad j \in [N],
$$

$$
\hat{\mathcal{C}}_{A^j A^j} = \frac{1}{T}\sum_{t=1}^T \psi_t^{a,j} \otimes \psi_t^{a,j}, \quad j \in [N].
$$

**1. Tensor Form (Tensor).** From Equation 29, the empirical block-operator is

$$
\hat{\mathcal{C}}_{O^i|A^j H^j} = \frac{1}{T}\sum_{t=1}^T \psi_t^{o,i} \otimes \big(\psi_t^{h,j} \otimes \psi_t^{a,j}\big), \quad j \in [N],
$$

and the conditional embedding operator is computed as

$$
\mathcal{C}_{O^i|A^j H^j} = \hat{\mathcal{C}}_{O^i|A^j H^j}\big(\hat{\mathcal{C}}_{(H^j A^j)(H^j A^j)} + \lambda I\big)^{-1},
\tag{33}
$$

where $\lambda I$ is the Tikhonov regularization and

$$\hat{\mathcal{C}}_{(A^j H^j)(A^j H^j)} = \frac{1}{T} \sum_{t=1}^{T} \left( \psi_t^{a,j} \otimes \psi_t^{h,j} \right) \otimes \left( \psi_t^{a,j} \otimes \psi_t^{h,j} \right).$$

**2. Dense Form (Dense).**  From Equation 30, the history and action parts are separated:

$$\hat{\mathcal{C}}_{O^i|H^j} = \hat{\mathcal{C}}_{O^i H^j} \left( \hat{\mathcal{C}}_{H^j H^j} + \lambda I \right)^{-1}, \quad j \in [N], \tag{34}$$

$$\hat{\mathcal{C}}_{O^i|A^j} = \hat{\mathcal{C}}_{O^i A^j} \left( \hat{\mathcal{C}}_{A^j A^j} + \lambda I \right)^{-1}, \quad j \in [N]. \tag{35}$$

**3. Homogeneous Form (Hom).**  In Equation 31, operators $\mathcal{C}_{O^i|H^j}$ for all $i$ are shared:

$$\hat{\mathcal{C}}_{O^i|H^j} = \frac{1}{N} \sum_{i=1}^{N} \hat{\mathcal{C}}_{O^i H^j} \left( \hat{\mathcal{C}}_{H^j H^j} + \lambda I \right)^{-1}, \quad j \in [N], \tag{36}$$

while $\mathcal{C}_{O^i|A^j}$ follows the empirical estimation of the *Dense* form.

**4. Homogeneous with Mean Field Approximation (Hom + Mean).**  From Equation 32, the history part uses adaptive weights $\alpha_t^{i,j}$:

$$
\begin{aligned}
\hat{\mathcal{C}}_{O^i|H} &= \mathcal{C}_{O^i H} (\hat{\mathcal{C}}_{HH} + \lambda I)^{-1} \\
&= \frac{1}{NT} \sum_{i=1}^{N} \sum_{t=1}^{T} \left( \psi_t^{o,i} \otimes \left( \sum_{j \in \mathcal{E}(i)} \alpha_t^{i,j} \psi_t^{h,j} \right) \right) \left( \left( \sum_{j \in \mathcal{E}(i)} \alpha_t^{i,j} \psi_t^{h,j} \right) \otimes \left( \sum_{j \in \mathcal{E}(i)} \alpha_t^{i,j} \psi_t^{h,j} \right) + \lambda I \right)^{-1},
\end{aligned}
\tag{37}
$$

with

$$\hat{\mathcal{C}}_{O^i H} = \frac{1}{NT} \sum_{i=1}^{N} \sum_{t=1}^{T} \psi_t^{o,i} \otimes \left( \sum_{j \in \mathcal{E}(i)} \alpha_t^{i,j} \psi_t^{h,j} \right), \tag{38}$$

$$\hat{\mathcal{C}}_{HH} = \frac{1}{NT} \sum_{i=1}^{N} \sum_{t=1}^{T} \left( \sum_{j \in \mathcal{E}(i)} \alpha_t^{i,j} \psi_t^{h,j} \right) \otimes \left( \sum_{j \in \mathcal{E}(i)} \alpha_t^{i,j} \psi_t^{h,j} \right). \tag{39}$$

Here, $\mathcal{C}_{O^i|H}$ is shared and computed as in Equation 37, while $\mathcal{C}_{O^i|A^j}$ follows the empirical estimation of the *Dense* form.

# F    THEORETICAL ANALYSIS

## F.1    PROOF OF THEOREM 1

**Proof 1.** *Step 1: Convergence of empirical operators. Based on the practical computation in Equation 33 in Appendix E.2, for all $j \in [N]$,*

$$\widehat{\mathcal{C}}_{O^i|A^jH^j} = \hat{\mathcal{C}}_{O^i|A^jH^j} \left( \hat{\mathcal{C}}_{(H^jA^j)(H^jA^j)} + \lambda I \right)^{-1},$$

*where*

$$\hat{\mathcal{C}}_{O^iA^jH^j} = \frac{1}{T} \sum_{t=1}^{T} \psi_t^{o,i} \otimes \left( \psi_t^{h,j} \otimes \psi_t^{a,j} \right),$$

$$\hat{\mathcal{C}}_{(A^jH^j)(A^jH^j)} = \frac{1}{T} \sum_{t=1}^{T} \left( \psi_t^{a,j} \otimes \psi_t^{h,j} \right) \otimes \left( \psi_t^{a,j} \otimes \psi_t^{h,j} \right).$$

*Since the feature maps are bounded, continuous, and the samples are i.i.d., the Hilbert space strong law of large numbers (Muandet et al., 2017) implies*

$$\hat{\mathcal{C}}_{O^iA^jH^j} \xrightarrow[HS]{a.s.} \mathcal{C}_{O^iA^jH^j}, \quad \hat{\mathcal{C}}_{(A^jH^j)(A^jH^j)} \xrightarrow[HS]{a.s.} \mathcal{C}_{(A^jH^j)(A^jH^j)}$$

*in Hilbert-Schmidt norm. With $\lambda = \lambda_T \downarrow 0$ and $T\lambda_T \to \infty$, operator perturbation results yield (Kato, 2013)*

$$\left\| \widehat{\mathcal{C}}_{O^i|A^jH^j} - \mathcal{C}_{O^i|A^jH^j} \right\|_{\text{op}} \xrightarrow{a.s.} 0.$$

*Step 2: Pointwise convergence on fixed inputs. For fixed $\{a_t^j, h_t^j\}_{j=1}^N$, define*

$$\widehat{\psi}_t^{O,i} = \sum_{j \in \mathcal{E}(i)} \widehat{\mathcal{C}}_{O^i|A^jH^j} \left[ \psi_t^{h,j} \otimes \psi_t^{a,j} \right],$$

$$\psi_t^{O,i\star} = \sum_{j \in \mathcal{E}(i)} \mathcal{C}_{O^i|A^jH^j} \left[ \psi_t^{h,j} \otimes \psi_t^{a,j} \right] = \mathbb{E} \left[ \psi_t^{O,i} \mid \{a_t^j, h_t^j\}_{j=1}^N \right].$$

*Then*

$$\left\| \widehat{\psi}_t^{O,i} - \psi_t^{O,i\star} \right\|_{\mathcal{H}} \leq \sum_{j \in \mathcal{E}(i)} \left\| \widehat{\mathcal{C}}_{O^i|A^jH^j} - \mathcal{C}_{O^i|A^jH^j} \right\|_{\text{op}} \cdot \left\| \psi_t^{h,j} \otimes \psi_t^{a,j} \right\|_{\mathcal{H}}$$

$$\leq C|\mathcal{E}(i)| \cdot \max_{j \in \mathcal{E}(i)} \left\| \widehat{\mathcal{C}}_{O^i|A^jH^j} - \mathcal{C}_{O^i|A^jH^j} \right\|_{\text{op}},$$

*where $C < \infty$ is a constant due to the boundedness of features in the compact space $\mathbb{O}^i$. The right-hand side converges to $0$ in probability, hence*

$$\left\| \widehat{\psi}_t^{O,i} - \psi_t^{O,i\star} \right\|_{\mathcal{H}} \xrightarrow{T \to \infty} 0.$$

*Step 3: From embedding convergence to weak convergence. If the kernel feature is characteristic (Sriperumbudur et al., 2011), then*

$$\text{MMD}\left( \widehat{P}(O_t^i \mid \{a_t^j, h_t^j\}), \, P(O_t^i \mid \{a_t^j, h_t^j\}) \right) = \left\| \widehat{\psi}_t^{O,i} - \psi_t^{O,i\star} \right\|_{\mathcal{H}}$$

*where $\text{MMD}$ is the maximum mean discrepancy [a], and thus the above convergence in $\mathcal{H}$ implies*

$$\widehat{P}(O_t^i \mid \{a_t^j, h_t^j\}_{j=1}^N) \xrightarrow{T \to \infty} P(O_t^i \mid \{a_t^j, h_t^j\}_{j=1}^N).$$

---

[a]The maximum mean discrepancy (MMD) between two distributions $P$ and $Q$ is defined as $\text{MMD}(P, Q) := \|m_P - m_Q\|_{\mathcal{H}}$, where $m_P$ and $m_Q$ are their kernel mean embeddings in the RKHS $\mathcal{H}$(Hofmann et al., 2005).

### F.2 Proof of Error Bounds of Various Embeddings in GCE

The main steps to prove the error bounds for Equation 10 is outlined as follows:

- **Procedure 1:** Combine the well-established matrix concentration inequalities in Appendix G to determine the sample complexity of the sub-operators $\mathcal{C}_{O^i|H}$ and $\mathcal{C}_{O^i|A^j}$;
- **Procedure 2:** Establish the union error bound for the Hilbert–Schmidt norm in the forward loss function shown in Equation 10.
- **Procedure 3:** Prove the sample complexity for other embeddings: **Hom** and **Dense**.

### F.3 Procedure 1: Sample Complexity of Sub-Operators

The regression error in Equations 35 and 37 arises from two primary sources: (1) the estimation of cross-variance operators $\hat{\mathcal{C}}_{O^i A^j}, \hat{\mathcal{C}}_{O^i H}$, as well as covariance operators $\hat{\mathcal{C}}_{A^j A^j}, \hat{\mathcal{C}}_{H^j H^j}$ based on finite i.i.d. $T$ samples; (2) the influence of the perturbed term $\lambda I$. We proceed to derive the non-asymptotic convergence rate step-by-step using concentration results for random matrices.

Here, we first give an instance proof for the $\mathcal{C}_a^{i,j}$, and this proof can be plugged into the other cases.

**Proposition 1.** *For the $i$-th object in graph, let $\hat{\mathcal{C}}_{O^i|A^j} = \hat{\mathcal{C}}_{O^i A^j}(\hat{\mathcal{C}}_{A^j A^j} + \lambda I)^{-1}$, where $\hat{\mathcal{C}}_{O^i A^j} \in \mathbb{R}^{d_o \times d_a}$ and $\hat{\mathcal{C}}_{A^j A^j} \in \mathbb{R}^{d_a \times d_a}$ are two estimated adjoint and self-adjoint operators, respectively. Assuming both two operators are from a finite i.i.d. $T$ samples in Equation 35, we have probability at least $1 - 3\delta$ with $\delta \in (0,1)$ and $T > \frac{c \log(2d_a/\delta)}{\lambda_{min}(A^j A^j)}$*

$$\|\mathcal{C}_{O^i|A^j} - \hat{\mathcal{C}}_{O^i|A^j}\| \leq E_{a,1}^{i,j} + E_{a,2}^{i,j},$$

*where*

$$E_{a,1}^{i,j} \leq \sqrt{\lambda_{max}(\mathcal{C}_{O^i O^i})} \cdot \frac{1}{\sqrt{\lambda_{min}(\mathcal{C}_{A^j A^j})}} \cdot \left( \frac{c \log(2d_a/\delta)}{T \lambda_{min}(\mathcal{C}_{A^j A^j})} + \lambda \right) \cdot \frac{1}{\lambda_{min}(\hat{\mathcal{C}}_{A^j A^j} + \lambda)}$$

*and*

$$E_{a,2}^{i,j} \leq \frac{\sqrt{\frac{2 \log((d_o + d_a)/\delta)v}{T}} + \frac{2 \log((d_o + d_a)/\delta)L}{3T}}{\left( 1 - \frac{c \log(2d/\delta)}{T \lambda_{min}(\mathcal{C}_{A^j A^j})} \right) \lambda_{min}(\mathcal{C}_{A^j A^j}) + \lambda}.$$

*Proof.* According to Lemmas 2 and 4, we can decompose empirical operators as two terms: target operators and errors as

$$\begin{aligned} \hat{\mathcal{C}}_{O^i A^j} &= \mathcal{C}_{O^i A^j} + \Lambda_{O^i A^j}, \\ \hat{\mathcal{C}}_{A^j A^j} &= \mathcal{C}_{A^j A^j} + \Lambda_{A^j A^j}, \end{aligned} \tag{40}$$

where $\Lambda_{O^i A^j}$ and $\Lambda_{A^j A^j}$ can be regarded as the residual error terms. Our target is to derive how the error bounds of $\Lambda_{O^i A^j}$ and $\Lambda_{A^j A^j}$ change with the sample number.

By differencing two operators, we obtain

$$\begin{aligned} &\mathcal{C}_{O^i|A^j} - \hat{\mathcal{C}}_{O^i|A^j} \\ =&\mathcal{C}_{O^i A^j} \mathcal{C}_{A^i A^j}^{-1} - \hat{\mathcal{C}}_{O^i A^j}(\hat{\mathcal{C}}_{A^j A^j} + \lambda I)^{-1} \\ =&\mathcal{C}_{O^i A^j} \mathcal{C}_{A^i A^j}^{-1} - (\mathcal{C}_{O^i A^j} + \Lambda_{O^i A^j})(\mathcal{C}_{A^j A^j} + \Lambda_{A^j A^j} + \lambda I)^{-1} \\ =&\underbrace{\mathcal{C}_{O^i A^j} \left( \mathcal{C}_{A^j A^j}^{-1} - (\mathcal{C}_{A^j A^j} + \Lambda_{A^j A^j} + \lambda I)^{-1} \right)}_{\text{part 1}} - \underbrace{\Lambda_{O^i A^j} \left( \mathcal{C}_{A^j A^j} + \Lambda_{A^j A^j} + \lambda I \right)^{-1}}_{\text{part 2}}. \end{aligned} \tag{41}$$

We divide the derivation of the error bounds into two steps, corresponding to parts 1 and 2, denoted by $E_{a,1}^{i,j}$ and $E_{a,2}^{i,j}$, respectively..

**Step 1. Deriving the Error bound $E_{a,1}^{i,j}$**

Based on the Woodbury identity [4] (a.k.a. matrix inversion lemma) (Deng, 2011), the first error term becomes

$$
\mathcal{C}_{O^i A^j} \left( \mathcal{C}_{A^j A^j}^{-1} - (\mathcal{C}_{A^j A^j} + \Lambda_{A^j A^j} + \lambda I)^{-1} \right)
$$
$$
= \mathcal{C}_{O^i A^j} \mathcal{C}_{A^j A^j}^{-1} \left( -\mathcal{C}_{A^j A^j} + \mathcal{C}_{A^j A^j} + \Lambda_{A^j A^j} + \lambda I \right) (\mathcal{C}_{A^j A^j} + \Lambda_{A^j A^j} + \lambda I)^{-1} \tag{42}
$$
$$
= \mathcal{C}_{O^i A^j} \mathcal{C}_{A^j A^j}^{-1} (\Lambda_{A^j A^j} + \lambda I)(\mathcal{C}_{A^j A^j} + \Lambda_{A^j A^j} + \lambda I)^{-1}.
$$

Plugging the Lemmas 2 and 4 into Equation 42, we get

$$
\left\| \mathcal{C}_{O^i A^j} \mathcal{C}_{A^j A^j}^{-1} (\Lambda_{A^j A^j} + \lambda I)(\mathcal{C}_{A^j A^j} + \Lambda_{A^j A^j} + \lambda I)^{-1} \right\|
$$
$$
= \left\| \mathcal{C}_{O^i O^i}^{1/2} \mathcal{C}_{A^j A^j}^{1/2} \mathcal{C}_{A^j A^j}^{-1/2} \mathcal{C}_{A^j A^j}^{-1/2} (\Lambda_{A^j A^j} + \lambda I)(\mathcal{C}_{A^j A^j} + \Lambda_{A^j A^j} + \lambda I)^{-1} \right\| \tag{43}
$$
$$
\leq \sqrt{\lambda_{max}(\mathcal{C}_{O^i O^i})} \cdot \frac{1}{\sqrt{\lambda_{min}(\mathcal{C}_{A^j A^j})}} \cdot \left( \frac{c \log(2 d_a / \delta)}{T \lambda_{min}(\mathcal{C}_{A^j A^j})} + \lambda \right) \cdot \frac{1}{\lambda_{min}(\hat{\mathcal{C}}_{A^j A^j} + \lambda)},
$$

where the first and second terms are derived using singular value decomposition techniques, the third term is based on the error bound for self-adjoint matrices with probability at least $1 - \delta$ when $T > \frac{c \log(2 d_a / \delta)}{\lambda_{min}(A^j A^j)}$ (a direct result from matrix Hoeffding inequality in Lemma 4), and the last term leverages fundamental properties of matrix theory. The primary contributor to the error bound is the statistical deviation term $\Lambda_{A^j A^j}$, while the regularization term $\lambda I$ is typically small in practice.

**Step 2. Deriving the Error bound $E_{a,2}^{i,j}$**

To derive the second error term $E_{a,2}^{i,j}$, we consider the bound

$$
\left\| \Lambda_{O^i A^j} (\mathcal{C}_{A^j A^j} + \Lambda_{A^j A^j} + \lambda I)^{-1} \right\| \leq \frac{\| \Lambda_{O^i A^j} \|}{\lambda_{\min}(\mathcal{C}_{A^j A^j} + \Lambda_{A^j A^j} + \lambda I)}
$$
$$
\leq \frac{\| \Lambda_{O^i A^j} \|}{\left( 1 - \frac{c \log(2 d_a / \delta)}{T \lambda_{\min}(\mathcal{C}_{A^j A^j})} \right) \lambda_{\min}(\mathcal{C}_{A^j A^j}) + \lambda} \tag{44}
$$
$$
\leq \frac{\sqrt{\frac{2 \log(2(d_o + d_a)/\delta) \cdot v}{T}} + \frac{2 \log(2(d_o + d_a)/\delta) \cdot L}{3T}}{\left( 1 - \frac{c \log(2 d_a / \delta)}{T \lambda_{\min}(\mathcal{C}_{A^j A^j})} \right) \lambda_{\min}(\mathcal{C}_{A^j A^j}) + \lambda},
$$

which holds with probability at least $1 - 2\delta$. The first inequality follows from the standard sub-multiplicative property of the operator norm. The second inequality is a consequence of Lemma 4, which applies matrix Hoeffding inequality to the self-adjoint operator $\mathcal{C}_{A^j A^j} + \Lambda_{A^j A^j}$. The final inequality uses Lemma 2, which applies the matrix Bernstein inequality to bound the spectral norm of the empirical deviation $\Lambda_{O^i A^j}$ with high probability.

Applying a union bound over the two error terms $E_{a,1}^{i,j}$ and $E_{a,2}^{i,j}$, the total error is bounded with probability at least $1 - 3\delta$.

**Proposition 2.** *For the $i$-th node in a homogeneous graph with $N$ nodes, let $\hat{\mathcal{C}}_{O^i | H} = \hat{\mathcal{C}}_{O^i H}(\hat{\mathcal{C}}_{HH} + \lambda I)^{-1}$, where $\hat{\mathcal{C}}_{HH} \in \mathbb{R}^{d_h \times d_h}$ and $\hat{\mathcal{C}}_{O^i H} \in \mathbb{R}^{d_o \times d_h}$ are two estimated adjoint and self-adjoint operators, respectively. Assuming both two operators are from a finite i.i.d. $T$ samples in Equation 37, we have probability at least $1 - 3\delta$ with $\delta \in (0, 1)$ and $T > \frac{c \log(2 d_h / \delta)}{N \lambda_{min}(HH)}$*

$$
\| \mathcal{C}_{O^i | H} - \hat{\mathcal{C}}_{O^i | H} \| \leq E_{h,1}^i + E_{h,2}^i,
$$

*where*

$$
E_{h,1}^i \leq \sqrt{\lambda_{max}(\mathcal{C}_{O^i O^i})} \cdot \frac{1}{\sqrt{\lambda_{min}(\mathcal{C}_{HH})}} \cdot \left( \frac{c \log(2 d_h / \delta)}{NT \lambda_{min}(\mathcal{C}_{HH})} + \lambda \right) \cdot \frac{1}{\lambda_{min}(\hat{\mathcal{C}}_{HH} + \lambda)}
$$

---
[4]It is shorten as $A^{-1} - (A + B)^{-1} = A^{-1} B (A + B)^{-1}$.

*and*

$$E_{h,2}^i \leq \frac{\sqrt{\frac{2\log((d_o+d_h)/\delta)v}{NT}} + \frac{2\log((d_o+d_h)/\delta)L}{3NT}}{\left(1 - \frac{c\log(2d_h/\delta)}{NT\lambda_{min}(\mathcal{C}_{HH})}\right)\lambda_{min}(\mathcal{C}_{HH}) + \lambda}.$$

**Remark 2.** *Due to the homogeneity of the graph, each node contributes $T$ i.i.d. samples, resulting in a total of $NT$ samples. This significantly improves the convergence rate of the estimated operator $\hat{\mathcal{C}}_h^i$ compared to a single-node setting.*

### F.4 Procedure 2: Sample Complexity of **Hom** + **Mean** in Loss Function 10

**Theorem a** (Sample Complexity of **Hom** + **Mean**). *Considering a homogeneous graph with $N$ nodes, the graph dynamics satisfy the formulation Equation 2. For Hilbert space embedding, history, action, and observation feature dimensions are $d_h$, $d_a$, and $d_o$, respectively. For finite i.i.d. sample number $T > \max_j \left(\frac{c\log(2d_a/\delta)}{\lambda_{min}(A^jA^j)} \vee \frac{c\log(2d_h/\delta)}{N\lambda_{min}(HH)}\right)$, we have a probability at least $1 - 3\delta$ with $\delta \in (0,1)$ satisfying*

$$\left\| \begin{bmatrix} \psi_t^{o,1} \\ \vdots \\ \psi_t^{o,N} \end{bmatrix} - \begin{bmatrix} \hat{\mathcal{C}}_{O^1|H} \\ \vdots \\ \hat{\mathcal{C}}_{O^N|H} \end{bmatrix} \odot \begin{bmatrix} \sum_j \alpha_t^{1,j}\psi_t^{h,j} \\ \vdots \\ \sum_j \alpha_t^{N,j}\psi_t^{h,j} \end{bmatrix} - \begin{bmatrix} \hat{\mathcal{C}}_{O^1|A^1} & \cdots & \hat{\mathcal{C}}_{O^1|A^N} \\ \vdots & \ddots & \vdots \\ \hat{\mathcal{C}}_{O^N|A^1} & \cdots & \hat{\mathcal{C}}_{O^N|A^N} \end{bmatrix} \begin{bmatrix} \psi_t^{a,1} \\ \vdots \\ \psi_t^{a,N} \end{bmatrix} \right\|_{HS}$$
$$\leq \mathcal{O}(N)\max_i(E_{h,1}^i + E_{h,2}^i) + \mathcal{O}(N^2)\max_{i,j}(E_{a,1}^{i,j} + E_{a,2}^{i,j}),$$

*where $E_{a,1}^{i,j}$, $E_{a,2}^{i,j}$, $E_{h,1}^i$ and $E_{h,2}^i$ are defined in Propositions 1 and 2.*

*Proof.* By applying the properties of Hilbert–Schmidt norm and triangle inequality, we have when $M = 1$

$$\begin{aligned} 10 &\leq \sum_{i=1}^N \left\| \psi_t^{o,i} - \hat{\mathcal{C}}_{O^i|H}\left(\sum_{j\in\mathcal{E}(i)} \alpha_t^{i,j}\psi_t^{h,j}\right) - \sum_{j=1}^N \hat{\mathcal{C}}_{O^i|A^j}\psi_t^{a,j} \right\| \\ &\leq \sum_{i=1}^N \left\| \mathcal{C}_{O^i|H}\left(\sum_{j\in\mathcal{E}(i)} \alpha_t^{i,j}\psi_t^{h,j}\right) - \hat{\mathcal{C}}_{O^i|H}\left(\sum_{j\in\mathcal{E}(i)} \alpha_t^{i,j}\psi_t^{h,j}\right) \right\| \\ &\quad + \sum_{i,j=1}^N \left\| \mathcal{C}_{O^i|A^j}\psi_t^{a,j} - \hat{\mathcal{C}}_{O^i|A^j}\psi_t^{a,j} \right\| \\ &\leq \sum_{i=1}^N \left\| \mathcal{C}_{O^i|H} - \hat{\mathcal{C}}_{O^i|H} \right\| \left\| \sum_{j\in\mathcal{E}(i)} \alpha_t^{i,j}\psi_t^{h,j} \right\| + \sum_{i,j=1}^N \left\| \mathcal{C}_{O^i|A^j} - \hat{\mathcal{C}}_{O^i|A^j} \right\| \left\| \psi_t^{a,j} \right\|. \end{aligned} \tag{45}$$

All features are well-defined in the Hilbert space, and thus both $\left\| \sum_{j\in\mathcal{E}(i)} \alpha_t^{i,j}\psi_t^{h,j} \right\|$ and $\left\| \psi_t^{a,j} \right\|$ are uniformly bounded by a constant (see the properties of kernel functions in Appendix C). By combining with Proposition 1, the error bound in the left-hand side scales with the number of nodes as $\mathcal{O}(N)(E_{h,1}^i + E_{h,2}^i)$.

Combining Proposition 2, the error bound of right-hand-side scales as $\mathcal{O}(N^2)\max_{i,j}(E_{a,1}^{i,j} + E_{a,2}^{i,j})$. Here $\max_{i,j}(E_{a,1}^{i,j} + E_{a,2}^{i,j})$ controls the upper error bound of sub-operators in actuation components.

Since the two parts hold under $T > \max_j \left(\frac{c\log(2d_a/\delta)}{\lambda_{min}(A^jA^j)}\right)$ and $T > \frac{c\log(2d_h/\delta)}{N\lambda_{min}(HH)}$, we rewrite the condition as $T > \max_j \left(\frac{c\log(2d_a/\delta)}{\lambda_{min}(A^jA^j)} \vee \frac{c\log(2d_h/\delta)}{N\lambda_{min}(HH)}\right)$. Combining the two parts together, we obtain the union error bound for the forward loss function as $\mathcal{O}(N)\max_i(E_{h,1}^i+E_{h,2}^i)+\mathcal{O}(N^2)\max_{i,j}(E_{a,1}^{i,j}+E_{a,2}^{i,j})$.

## F.5 PROCEDURE 3: SAMPLE COMPLEXITY OF OTHER EMBEDDINGS

**Theorem b** (Sample Complexity of **Hom**). *Considering a homogeneous graph with $N$ nodes, the graph dynamics satisfy the formulation Equation 2. For Hilbert space embedding, history, action, and observation feature dimensions are $d_h$, $d_a$, and $d_o$, respectively. For finite i.i.d sample number $T > \max_j \left( \frac{c \log(2d_a/\delta)}{\lambda_{min}(A^j A^j)} \vee \frac{c \log(2d_h/\delta)}{N \lambda_{min}(HH)} \right)$, we have a probability at least $1 - 3\delta$ with $\delta \in (0,1)$ satisfying*

$$\left\| \begin{bmatrix} \psi_t^{o,1} \\ \vdots \\ \psi_t^{o,N} \end{bmatrix} - \left( \begin{bmatrix} \mathcal{C}_{O^1|H^1} & \cdots & \mathcal{C}_{O^1|H^N} \\ \vdots & \ddots & \vdots \\ \mathcal{C}_{O^N|H^1} & \cdots & \mathcal{C}_{O^N|H^N} \end{bmatrix} \begin{bmatrix} \psi_t^{h,1} \\ \vdots \\ \psi_t^{h,N} \end{bmatrix} + \begin{bmatrix} \mathcal{C}_{O^1|A^1} & \cdots & \mathcal{C}_{O^1|A^N} \\ \vdots & \ddots & \vdots \\ \mathcal{C}_{O^N|A^1} & \cdots & \mathcal{C}_{O^N|A^N} \end{bmatrix} \begin{bmatrix} \psi_t^{a,1} \\ \vdots \\ \psi_t^{a,N} \end{bmatrix} \right) \right\|_{HS}$$
$$\leq \mathcal{O}(N^2) \max_i (E_{h,1}^i + E_{h,2}^i) + \mathcal{O}(N^2) \max_{i,j} (E_{a,1}^{i,j} + E_{a,2}^{i,j})$$

*where $E_{a,1}^{i,j}$, $E_{a,2}^{i,j}$, $E_{h,1}^i$ and $E_{h,2}^i$ are defined in Propositions 1 and 2. .*

*Proof.* The proof of this theorem follows the same idea with Theorem a, the main difference is that here we need to estimate each $\mathcal{C}_{O^i|H^j}$, since the implicit weight influenced by the $j$-th node is unknown. Thus, the upper error bound is scaling quadratically with the number of nodes, since there are $N^2$ operators in the first block.

**Theorem c** (Sample Complexity of **Dense**). *Considering a graph with $N$ nodes, the graph dynamics satisfy the formulation Equation 2. For Hilbert space embedding, history, action, and observation feature dimensions are $d_h$, $d_a$, and $d_o$, respectively. For finite i.i.d. sample number $T > \max_{i,j} \left( \frac{c \log(2d_a/\delta)}{\lambda_{min}(A^i A^i)} \vee \frac{c \log(2d_h/\delta)}{\lambda_{min}(H^j H^j)} \right)$, we have a probability at least $1 - 3\delta$ with $\delta \in (0,1)$ satisfying*

$$\left\| \begin{bmatrix} \psi_t^{o,1} \\ \vdots \\ \psi_t^{o,N} \end{bmatrix} - \left( \begin{bmatrix} \mathcal{C}_{O^1|H^1} & \cdots & \mathcal{C}_{O^1|H^N} \\ \vdots & \ddots & \vdots \\ \mathcal{C}_{O^N|H^1} & \cdots & \mathcal{C}_{O^N|H^N} \end{bmatrix} \begin{bmatrix} \psi_t^{h,1} \\ \vdots \\ \psi_t^{h,N} \end{bmatrix} + \begin{bmatrix} \mathcal{C}_{O^1|A^1} & \cdots & \mathcal{C}_{O^1|A^N} \\ \vdots & \ddots & \vdots \\ \mathcal{C}_{O^N|A^1} & \cdots & \mathcal{C}_{O^N|A^N} \end{bmatrix} \begin{bmatrix} \psi_t^{a,1} \\ \vdots \\ \psi_t^{a,N} \end{bmatrix} \right) \right\|_{HS}$$
$$\leq \mathcal{O}(N^2) \max_{i,j} (F_{h,1}^{i,j} + F_{h,2}^{i,j}) + \mathcal{O}(N^2) \max_{i,j} (E_{a,1}^{i,j} + E_{a,2}^{i,j}),$$

*where $E_{a,1}^{i,j}$ and $E_{a,2}^{i,j}$ are defined in Proposition 1 and $E_{h,1}^{i,j}$ and $E_{h,2}^{i,j}$ are defined as*

$$F_{h,1}^{i,j} \leq \sqrt{\lambda_{max}(\mathcal{C}_{O^i O^i})} \cdot \frac{1}{\sqrt{\lambda_{min}(\mathcal{C}_{H^j H^j})}} \cdot \left( \frac{c \log(2d_h/\delta)}{T \lambda_{min}(\mathcal{C}_{H^j H^j})} + \lambda \right) \cdot \frac{1}{\lambda_{min}(\hat{\mathcal{C}}_{H^j H^j} + \lambda)}$$

*and*

$$F_{h,2}^{i,j} \leq \frac{\sqrt{\frac{2 \log((d_o+d_h)/\delta)v}{T}} + \frac{2 \log((d_o+d_h)/\delta)L}{3T}}{\left( 1 - \frac{c \log(2d_h/\delta)}{T \lambda_{min}(\mathcal{C}_{H^j H^j})} \right) \lambda_{min}(\mathcal{C}_{H^j H^j}) + \lambda}.$$

*Proof.* The error bound difference between the homogeneous graph and inhomogeneous graph is that each operator $\mathcal{C}_{O^i|H^j}$ should be estimated independently (see the explanation in Equations 30 and 31). Thus, each error bound of $F_{h,1}^{i,j}$ and $F_{h,2}^{i,j}$ can be proved by following Proposition 1. Following the same idea shown in Theorems a and 2, the union upper bound can be derived as $\mathcal{O}(N^2) \max_{i,j} (F_{h,1}^{i,j} + F_{h,2}^{i,j}) + \mathcal{O}(N^2) \max_{i,j} (E_{a,1}^{i,j} + E_{a,2}^{i,j})$.

# G   AUXILIARY LEMMAS

The auxiliary lemmas in this section are primarily based on the monographs (Vershynin, 2018; Tropp et al., 2015). These four lemmas are focused on handling error bounds for the approximated adjoint and self-adjoint operators that arise in regression. The derivation of these error bounds relies on three fundamental matrix inequalities: Matrix Bernstein, Chernoff, and Hoeffding, which serve as key tools for controlling the embedding errors.

**Lemma 1** (Matrix Bernstein Inequality (Vershynin, 2018; Tropp et al., 2015)). *Consider a finite i.i.d. sequence $\{X_k\}_{k=1}^n$ of independent, random matrices with finite dimension $d_1 \times d_2$. Assume that $X_k$ is centered at $0$ and bounded by a constant $L$ for any index $k$ as*

$$\mathbb{E}[X_k] = 0 \quad and \quad \|X_k\| \leq L.$$

*Let $Z = \sum_{k=1}^n X_k$. Define the matrix variance statistic:*

$$v(Z) = \max\{\|\sum_{k=1}^n \mathbb{E}[X_k X_k^\top]\|, \|\sum_{k=1}^n \mathbb{E}[X_k^\top X_k]\|\}. \tag{46}$$

*Then we have*

$$P\left(\|Z\| \geq t\right) \leq (d_1 + d_2) \exp(\frac{-t^2/2}{v(Z) + Lt/3}).$$

The introduction of the term $v(Z)$ arises from the symmetrization technique used in the proof, where the operator norm of the sum is related to the maximum of the operator norms of the variance terms in both the original and transposed spaces.

**Lemma 2** (Error Bound of Cross-Covariance Operator). *When Lemma 1 holds, we have a probability at least $1 - \delta$*

$$\|\hat{\mathcal{C}}_{YX} - \mathcal{C}_{YX}\| \leq \sqrt{\frac{2\log((d_1 + d_2)/\delta)v}{N}} + \frac{2\log((d_1 + d_2)/\delta)L}{3N},$$

*where*

$$\mathcal{C}_{YX} \in \mathbb{R}^{d_1 \times d_2},$$

$$c_1 = \left\|(\hat{\mathcal{C}}_{YX} - \mathcal{C}_{YX})(\hat{\mathcal{C}}_{YX} - \mathcal{C}_{YX})^\top\right\|,$$

$$c_2 = \left\|(\hat{\mathcal{C}}_{YX} - \mathcal{C}_{YX})^\top(\hat{\mathcal{C}}_{YX} - \mathcal{C}_{YX})\right\|,$$

$$v = \max(c_1, c_2).$$

**Remark 3.** *Lemma 2 is a direct result based on Lemma 1. Following from finite i.i.d. samples, the convergence rate with the number of samples $N$ is scaling as $\mathcal{O}(\sqrt{\frac{\log(d_1 + d_2)}{N}})$. The error bound established in Lemma 2 is particularly valuable for conducting non-asymptotic analyses of cross-variance operators.*

**Lemma 3** (Matrix Chernoff Bound (Vershynin, 2018)). *Consider a finite i.i.d. sequence $\{X_k\}$ of random and symmetric matrices with a finite dimension $d_o$. Assuming that the eigenvalue of $X_k$ is bounded by a constant $L$ for all index $k$ as*

$$0 \leq \lambda_{min}(X_k) \quad and \quad \lambda_{max}(X_k) \leq L.$$

*Introduce the random matrix*

$$Z = \sum_k X_k.$$

*Define*

$$\mu_{min} = \lambda_{min}(\mathbb{E}[Z]).$$

*Then, for any $\epsilon \in [0, 1)$, we have*

$$P\left(\lambda_{min}(Z) \leq (1 - \epsilon)\mu_{min}\right) \leq 2d_o \exp(-\epsilon\mu_{min}/L). \tag{47}$$

**Lemma 4** (Error Bound of Covariance Operator (Tropp et al., 2015)). *Consider the random variable* $\hat{\mathcal{C}}_{XX} \in \mathbb{R}^{d \times d}$, *we have a probability at least* $1 - \delta$ *with* $\delta \in (0, 1)$ *when* $N > \frac{c \log(2d/\delta)}{\lambda_{min}}$

$$\lambda_{min}(\hat{\mathcal{C}}_{XX}) \geq \left(1 - \frac{c \log(2d/\delta)}{N \lambda_{min}(\mathcal{C}_{XX})}\right) \lambda_{min}(\mathcal{C}_{XX}),$$

*where*

$$\lambda_{max}(\hat{\mathcal{C}}_{XX}) \leq L \leq \frac{c}{N}, \quad with \quad c > 0.$$

**Remark 4.** *Lemma 4 can be derived from the matrix Chernoff bound through the Laplace transform technique, which transforms the multiplicative bound into an additive form of matrix concentration inequality. The resulting error bound exhibits a convergence rate of $\mathcal{O}(\frac{\log d}{N})$ with respect to the sample size $N$. This concentration inequality is particularly effective for analyzing covariance operators, especially in the context of empirical covariance operators in Hilbert space embedding. The primary source of estimation error arises from the finite sampling, which is quantified by this probabilistic bound.*

# H   PSEUDO CODE

---

**Algorithm 1** Graph Controllable Embedding (*Hom+Mean*)

**Modeling**

---

**Require:** Data $\mathcal{D} = \left\{ \{\mathcal{T}_t^i\}_{t=0}^M \right\}_{i=0}^{N_{\text{sample}}}$: $t-$step transition tuple $\mathcal{T}_t^i = (o_t^{1:N}, a_t^{1:N}, h_t^{1:N})$ with $N$ objects; learning rate $\alpha$; number of training epoches $K$; four different pair-wise potential energy functions $f(\,\cdot\,,\,\cdot\,)$ in Table 2

1: Initialize GNN encoder and decoder $\psi$ and $\psi^\dagger$
2: **for** Training epoch $k = 0, ..., K$ **do**
3:     {Encode inputs: observations and history via neural embeddings; actions via a fixed linear projection (ensuring direct recovery back to the original action space)}
4:     Embedding graph history $h_t^{1:N}$ and future observations $o_t^{1:N}$: $\psi_t^{o,1:N} = \phi_\mathbb{O}(o_t^{1:N})$ and $\psi_t^{h,1:N} = \phi_\mathbb{H}(h_t^{1:N})$, action features are obtained via a fixed linear projection: $\psi_t^{a,1:N} = \phi_\mathbb{A}(a_t^{1:N})$
5:     Estimate Boltzmann-Gibbs weights $\alpha_t^{i,j} = \dfrac{\exp\left(f(\psi_t^{h,i}, \psi_t^{h,j})\right)}{\sum\limits_{k \in \mathcal{E}(i)} \exp\left(f(\psi_t^{h,i}, \psi_t^{h,k})\right)}$
6:     Compute the ensemble features $\sum_{j \in \mathcal{E}(i)} \alpha_t^{i,j} \psi_t^{h,j}$
7:     Estimate $\{\hat{\mathcal{C}}_{O^i|H}\}_{i=1:N}$ and $\{\hat{\mathcal{C}}_{O^i|A^j}\}_{i,j=1:N}$ by solving Equation 10 by minimizing Hilbert-Schimit norm
8:     Compute the loss $\mathcal{L}_{\text{rec}}$ defined by Equation 11
9:     Update encoder and decoder $\psi$ and $\psi^\dagger$ with stochastic gradient descent
10: **end for**

**Control**

---

**Require:** target $\mathcal{V}_*^o$; initial observation as history $\mathcal{V}_0^o$; control horizon $M$; GNN encoder and decoder $\psi$ and $\psi^\dagger$

11: Construct the control objective Equation 12
12: Obtain the optimal control sequences $(\mathcal{V}_1^a, ..., \mathcal{V}_M^a)$ from minimizing Equation 12
13: Apply $(\mathcal{V}_1^a, ..., \mathcal{V}_M^a)$

---

# I    ADDITIONAL EXPERIMENT DETAILS

## I.1    CONTROL ENVIRONMENT DETAILS

In the numerical experiments, we adopt the control environments from (Li et al., 2020), where interactions between different objects are considered differently depending on the types of connections and the physical properties of the objects involved. The environments are designed to capture diverse interaction dynamics, as detailed below:

- **Rope.** In the Rope environment, the top mass has a fixed height and is considered different from the other masses. Thus, there are 2 kinds of self-interactions for the top mass and the non-top masses, respectively. Additionally, we have 8 kinds of interactions between different objects. The objects in a relation could be either top mass or non-top mass. It is a combination of 4. The interaction may happen between two adjacent masses or masses that are two-hop away. In total, the number of interactions between different objects is $4 \times 2 = 8$. *The in-distribution test includes $5 - 9$ objects, and few-shot test is from $10 - 14$ objects.* Examples of rope systems are demonstrated in Figure 5.

- **Soft.** In the Soft environments, there are four types of quadrilaterals: rigid, soft, actuated, and fixed. There are four types of self-interactions, respectively. For the interactions between objects, there is an edge between two quadrilaterals only if they are connected by a point or edge. Connections from different directions are considered as different relations. There are 8 different directions: up, down, left, right, up-left, down-left, up-right, down-right. The relation types also encode the type of receiver object. Thus, in total, there are $(8 + 1) \times 4 = 36$ types of relations between different objects. *The in-distribution test includes $5 - 9$ quadrilaterals, and few-shot test is from $10 - 14$ quadrilaterals.* Examples of soft robotics are demonstrated in Figure 6.

- **Swim.** In the Swim environment, there are three types of quadrilaterals: rigid, soft, and actuated. Similarly to the Soft environment, different edge types are specified for different connection directions. The number of edge types is $(8+1) \times 3 = 27$. *The in-distribution test includes $5 - 9$ quadrilaterals, and few-shot test is from $10 - 14$ quadrilaterals.* Examples of swim robotics are demonstrated in Figure 7.

- **Power grid.** The primary control objective in this environment is to stabilize the voltage magnitudes at all nodes around a predefined reference voltage value, $V_{\text{ref}} = 1$ p.u.. Each node has a state vector $[V_i^t, \dot{V}_i^t]$, where $V_i^t$ is the voltage magnitude at node $i$ at time step $t$, and $\dot{V}_i^t$ is its rate of change. This stabilization is achieved by adjusting the reactive power generation $Q_{\text{gen}}$ at generator nodes in response to both internal system dynamics and external disturbances (i.e., noise) generated by the loads.

  *The underlying grid topology is modeled as an undirected graph, which may be fixed (e.g., ring or grid structures) or randomly generated using an Erdős–Rényi model* (Examples are illustrated in Figure 8). The graph is assumed to be connected, and nodes are randomly designated as either generators or loads, maintaining a generator ratio between 20% and 50%. Generator nodes are controllable via reactive power inputs, while load nodes are subject to time-varying noise disturbances. To learn the system dynamics, we generate 1000 randomized instances with topologies ranging from 50 to 100 nodes and evaluate the control performance on larger graphs with over 100 nodes. The control problem is formulated as the following cost minimization over a time horizon $M$:

$$\sum_{t=1}^{M} \left\| V^t - V_{\text{ref}}^t \right\|_{Q_1}^2 + \left\| u^t \right\|_{Q_2}^2,$$

  where $Q_1$ and $Q_2$ are positive definite weighting matrices, and $u^t$ is the control input at time step $t$.

## I.2    DATA GENERATION

We generate 10,000 episodes for the Rope environment and 50,000 episodes for the Soft and Swim environments, with 90% of the episodes used for training and the remaining 10% for testing. Each

episode consists of 100 time steps. The dataset includes physical systems with varying numbers of objects ranging from 5 to 9. In the Rope environment, ropes consist of 5 to 9 masses, where each mass is treated as an object. The observation for each mass includes its position and velocity in a 2D space, resulting in a 4-dimensional observation per mass. In general, a rope with $N$ masses has a $4N$-dimensional observation space. In the Soft and Swim environments, soft robots consist of 5 to 9 quadrilaterals, each treated as an object. For each quadrilateral, we observe the positions and velocities of its four corners, leading to a 16-dimensional observation (4 positions $\times$ 4 velocities). Thus, for a soft robot with $N$ quadrilaterals, the observation has a dimensionality of $16N$. Additionally, we generate a few-shot set with 200 episodes for each environment to evaluate the generalization of our graph controllable embedding. The Rope few-shot set includes ropes with 10–14 masses, while the Soft and Swim sets include robots with 10–14 quadrilaterals, more than the number of objects in the training sets. The graph topology is generated with a random structure and a random number of nodes, ranging from 50 to 150. To ensure the controllability of the random graph, the graph must be connected. This is guaranteed if the edge connection probability $p$ satisfies

$$p > \frac{(1+\delta)\ln n}{n},$$

where $n$ is the number of nodes and $\delta > 0$ is a small constant. In our setup, we set the connection probability to $p = 0.15$.

## I.3    TRAINING DETAILS

Our methods and baseline models are trained using the Adam optimizer with a learning rate of $1e^{-4}$ and a batch size of 8.The learning rate scheduler follows a linear schedule with a decay rate of 50% per 100K gradient-descent steps until reaching a minimum learning rate of $1e^{-6}$. For all models, we apply 400K iterations of gradient-descent steps in the Rope environment and 580K iterations in the Soft and Swim environments, respectively. The number of future steps in the loss function Equation 10 is set to $M = 16$. All experiments were run on a single RTX 4090 GPU.

## I.4    BASELINE ALGORITHMS

The baseline algorithms are trained on the same dataset and device. We compare the control performance of our model with two categories of baselines:

- **Controllable embedding methods without relational structures:**
  - VAE-based controllable embeddings: `https://github.com/ericjang/e2c`,
  - Prediction, Consistency, Curvature (PCC) `https://github.com/VinAIResearch/PCC-pytorch`,
- **Graph representation methods:**
  - Koopman Polynomial Model (KPM): `https://github.com/YunzhuLi/CompositionalKoopmanOperators/blob/master/models/CompositionalKoopmanOperators.py`,
  - Compositional Koopman Operator (CKO): `https://github.com/YunzhuLi/CompositionalKoopmanOperators/blob/master/models/CompositionalKoopmanOperators.py`.
  - GraphODE: `https://github.com/Zymrael/gde`.

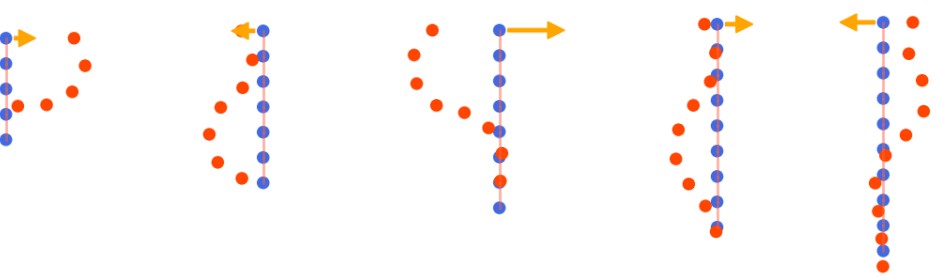

Figure 5: Examples of the random parametrized shape of the rope system. The red dots in each sub-figure represent the control targets.

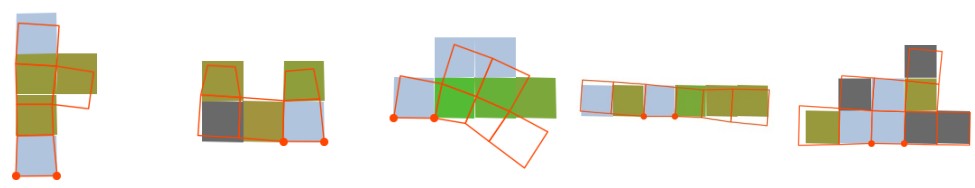

Figure 6: Examples of the random parametrized shape of soft robotics. The red boxes are represented as the control target of each soft robot.

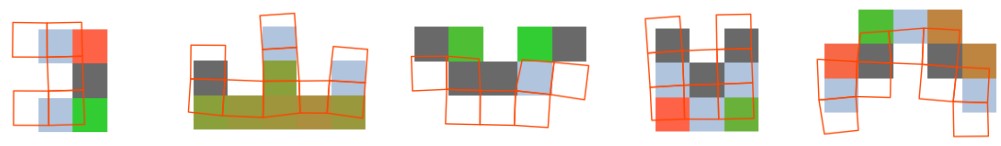

Figure 7: Examples of the random parametrized shape of swim robotics. The red boxes are represented as the control target of each swim robot.

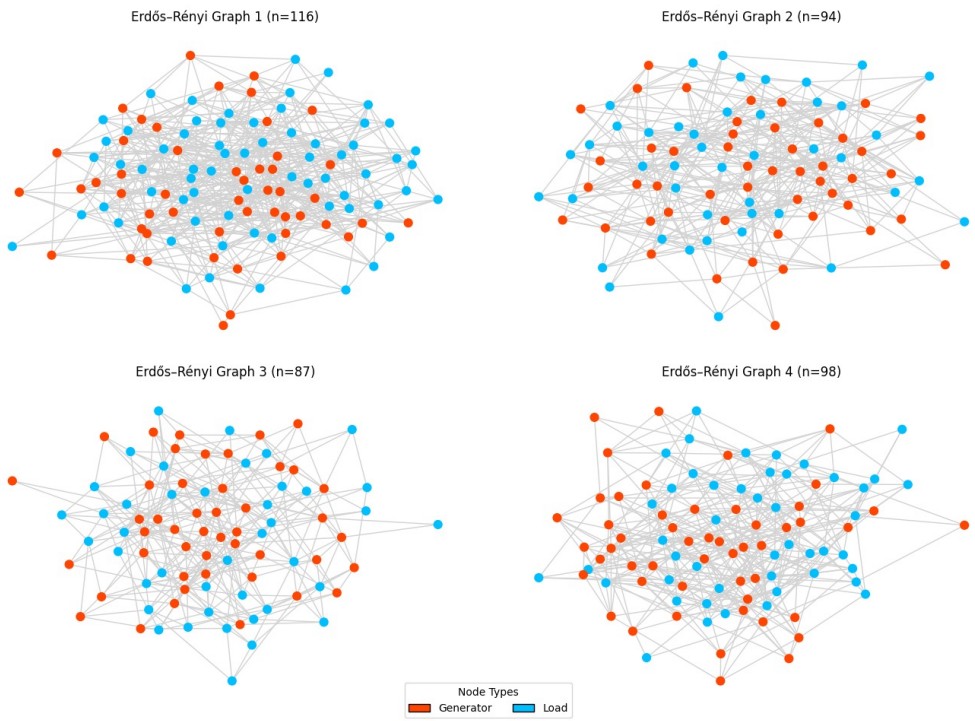

Figure 8: Examples of the random graph of a power grid. The red node represents the generator, and the blue node represents the load.

## I.5 Additional Experimental Results

**Noisy vs. Noiseless Conditions.** We evaluate the predictive performance of our models by forecasting 100 steps into the future in the Rope and Soft environments, with and without observation noise, and compare with the state-of-the-art CKO method. Figure 9 presents quantitative results averaged over 20 trajectories from different initial conditions; shaded regions indicate one standard deviation. Across both environments and noise settings, our method consistently achieves lower NRMSE and standard deviations than CKO, demonstrating improved robustness and accuracy in long-horizon dynamics prediction.

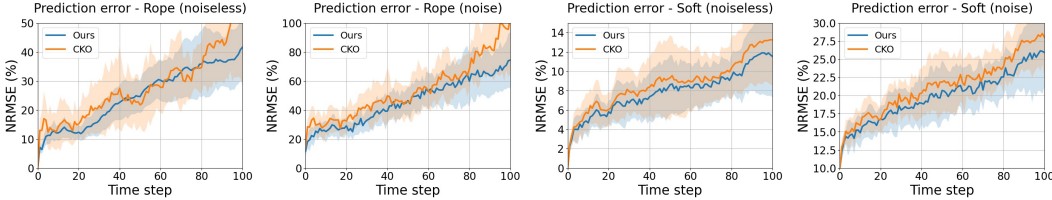

Figure 9: Comparison of normalized root-mean-square error (NRMSE) over 100 rollout steps. The left two figures show results for the Rope environment with and without noise; the right two figures show results for the Soft environment. Additive noise is zero-mean Gaussian with standard deviation equal to 10% of the standard deviation of the observation data.

**Sample Efficiency.** To understand the sample efficiency of our model, we vary the number of data samples used for system identification from 1 to 32, and compare with CKO as shown in Figure 10. This empirical trend aligns with our theoretical analysis: according to Theorems a and b, the error bound scales proportionally to $\sqrt{1/T}$, where $T$ is the number of samples. Given that constants such as $d_o$, $d_h$, and $N$ remain fixed for a chosen feature dimension, the sample size $T$ becomes the primary factor influencing the bound. As the fitting number increases (i.e., $T$), the prediction error consistently decreases, confirming the theoretical rate. Moreover, as shown in Theorems a and b, our method (**Hom + Mean**) achieves an error bound that scales as $\mathcal{O}(N)$ for history-to-observation part, in contrast to CKO (the **Hom** framework) with a bound scaling as $\mathcal{O}(N^2)$. Consequently, when both methods are evaluated with the same fitting number, our method consistently yields a smaller error bound. These results demonstrate that our theoretical predictions are well supported by empirical observations, validating the improved sample efficiency of our approach.

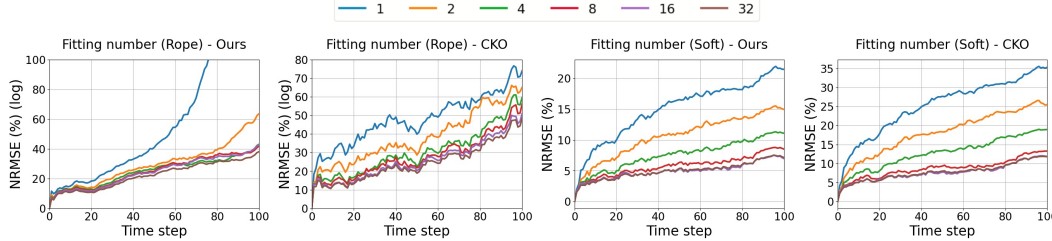

Figure 10: Prediction error (NRMSE) over 100 rollout steps under varying fitting number used for system identification, ranging from 1 to 32, as indicated in the legend. Results are shown for both the Rope and Soft environments. The comparison includes our method and the CKO baseline. As the number of samples increases, our method achieves lower prediction error with faster convergence, demonstrating improved sample efficiency consistent with the theoretical rate predicted in Theorems a and b.

**Evaluation Time of Different Embeddings.** We measure the computation cost of different embeddings by recording the time required to predict 100 steps. As shown in Figure 11, Hom+Mean achieves the fastest runtime, outperforming both Dense and Hom. This observation is consistent with the theoretical computation complexity analysis summarized in Table 8.

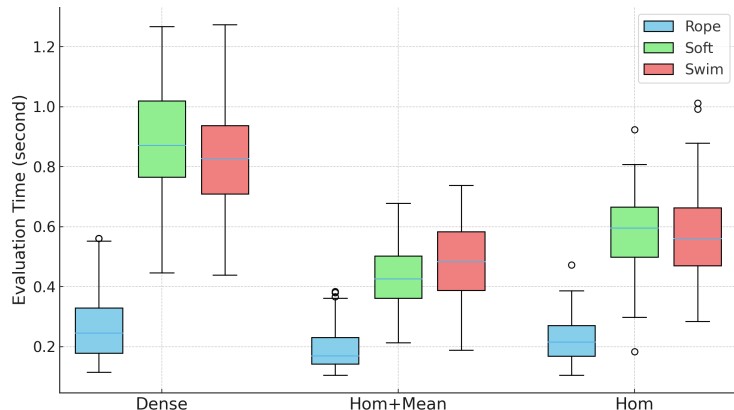

Figure 11: Evaluation time (seconds) for predicting 100 steps with different embeddings. Hom+Mean achieves the lowest runtime, consistent with the theoretical complexity in Table 8.

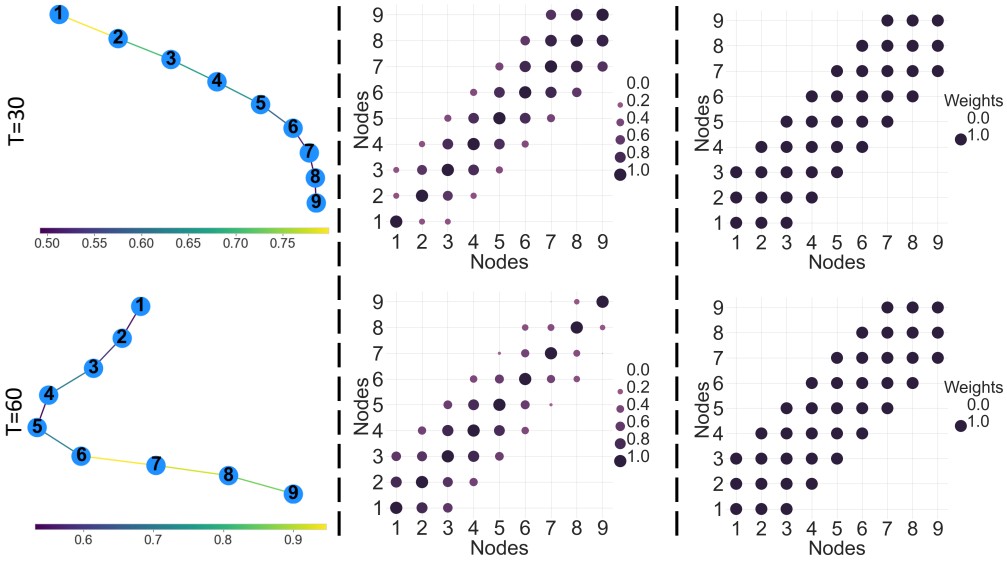

Figure 12: Visualization of a 9-object Rope. The left column shows observations at $t = 30$ and 60, with shading color representing pair-wise $L^2$ distances. The middle column depicts non-normalized adaptive Boltzmann–Gibbs weights, highlighting non-uniform neighbor influence (ours). The right column illustrates the misspecified uniform weights, where all neighbors exert equal, non-normalized influence.

Table 9: Full performance comparison. Results are reported for in-distribution and few-shot validation. Control cost and control error are shown as mean ± standard deviation, with the best and second-best results highlighted in green and blue, respectively.

| Methods | Environments | In-Distribution Validation | | Few-Shot Validation | |
|---|---|---|---|---|---|
| | | Control cost | Control error | Control cost | Control error |
| VAE | | 259.7 ± 80.9 | 0.65 ± 0.14 | 282.4 ± 82.3 | 0.69 ± 0.25 |
| PCC | | 212.6 ± 64.2 | 0.58 ± 0.12 | 246.7 ± 71.5 | 0.62 ± 0.13 |
| GraphODE | | 187.2 ± 51.7 | 0.49 ± 0.13 | 198.0 ± 53.8 | 0.49 ± 0.09 |
| KPM | | 148.5 ± 39.3 | 0.36 ± 0.09 | 206.1 ± 43.5 | 0.21 ± 0.04 |
| CKO | Rope | 135.9 ± 28.0 | 0.30 ± 0.06 | 168.5 ± 45.4 | 0.22 ± 0.04 |
| Ours (vMF) | | 137.5 ± 30.2 | 0.27 ± 0.09 | **146.3 ± 49.7** | **0.13 ± 0.05** |
| Ours (Laplace) | | 130.6 ± 32.7 | 0.32 ± 0.06 | 162.8 ± 46.2 | 0.17 ± 0.05 |
| Ours (Gaussian) | | **122.6 ± 19.2** | **0.26 ± 0.08** | 159.4 ± 43.1 | 0.14 ± 0.03 |
| VAE | | 314.7 ± 57.0 | 0.54 ± 0.25 | 768.1 ± 137.0 | 0.49 ± 0.24 |
| PCC | | 289.4 ± 49.7 | 0.51 ± 0.22 | 698.3 ± 102.6 | 0.45 ± 0.21 |
| GraphODE | | 194.9 ± 35.6 | 0.33 ± 0.15 | 472.9 ± 84.1 | 0.29 ± 0.13 |
| KPM | | 164.6 ± 22.6 | 0.19 ± 0.09 | 388.5 ± 65.8 | 0.20 ± 0.08 |
| CKO | Soft | 160.8 ± 26.4 | 0.17 ± 0.06 | 357.6 ± 63.7 | 0.20 ± 0.07 |
| Ours (vMF) | | **155.2 ± 28.3** | 0.16 ± 0.08 | 357.2 ± 61.4 | 0.22 ± 0.10 |
| Ours (Laplace) | | 169.5 ± 19.1 | 0.18 ± 0.09 | 382.5 ± 70.9 | 0.19 ± 0.08 |
| Ours (Gaussian) | | 158.6 ± 21.7 | **0.13 ± 0.05** | **344.2 ± 56.3** | **0.12 ± 0.06** |
| VAE | | 573.1 ± 108.7 | 0.73 ± 0.19 | 835.4 ± 113.2 | 0.92 ± 0.15 |
| PCC | | 513.3 ± 92.5 | 0.68 ± 0.15 | 732.8 ± 94.5 | 0.80 ± 0.12 |
| GraphODE | | 417.8 ± 87.9 | 0.52 ± 0.17 | 693.5 ± 58.2 | 0.58 ± 0.09 |
| KPM | | 385.5 ± 75.2 | 0.44 ± 0.06 | 523.4 ± 22.8 | 0.61 ± 0.11 |
| CKO | Swim | 389.1 ± 76.9 | 0.42 ± 0.13 | 421.0 ± 70.0 | 0.44 ± 0.08 |
| Ours (vMF) | | 392.7 ± 73.1 | 0.45 ± 0.09 | 452.3 ± 62.9 | 0.43 ± 0.15 |
| Ours (Laplace) | | 403.1 ± 68.3 | 0.46 ± 0.13 | 435.7 ± 74.4 | 0.45 ± 0.10 |
| Ours (Gaussian) | | **383.7 ± 77.8** | **0.41 ± 0.08** | **404.3 ± 74.2** | **0.41 ± 0.09** |

Table 10: Full evaluation under varying noise levels in Power-Grid. With the test on random graphs with 100-150 objects. Additive noise is introduced with standard deviations equal to $2\%, 5\%, 10\%, 20\%$ of the standard deviation of observations. "NaN" means the unstable control optimization.

| | Method | Noiseless | 2% | 5% | 10% | 20% |
|---|---|---|---|---|---|---|
| | GraphODE | 0.58 ± 0.043 | 0.62 ± 0.073 | NaN | NaN | NaN |
| *Control Error* | KPM | 0.42 ± 0.028 | 0.50 ± 0.022 | NaN | NaN | NaN |
| | CKO | 0.47 ± 0.031 | 0.48 ± 0.034 | 0.51 ± 0.027 | 0.65 ± 0.051 | 0.85 ± 0.055 |
| | Ours (Gaussian) | **0.21 ± 0.005** | **0.27 ± 0.018** | **0.39 ± 0.024** | **0.63 ± 0.037** | **0.83 ± 0.041** |
| | GraphODE | 13.29 ± 0.95 | 14.93 ± 1.06 | NaN | NaN | NaN |
| *Control Cost* | KPM | 9.19 ± 1.66 | 11.70 ± 1.68 | NaN | NaN | NaN |
| | CKO | 11.04 ± 0.82 | 11.22 ± 0.84 | 11.69 ± 1.67 | 12.42 ± 2.27 | 15.15 ± 4.55 |
| | Ours (Gaussian) | **4.77 ± 0.61** | **4.99 ± 0.69** | **5.72 ± 0.91** | **7.81 ± 1.13** | **11.43 ± 1.14** |

Table 11: Control error and control cost under different numbers of training trajectories used for few-shot adaptation in the Rope environment. The table reports performance using varying numbers of demonstration trajectories, referred to as "fitting number".

| | Method | 1 | 4 | 8 | 16 | 32 |
|---|---|---|---|---|---|---|
| | **Dense** | 0.79 ± 0.25 | 0.41 ± 0.11 | 0.36 ± 0.12 | 0.28 ± 0.08 | 0.26 ± 0.10 |
| *Control Error* | **Hom** | 0.74 ± 0.25 | 0.32 ± 0.08 | 0.30 ± 0.06 | 0.30 ± 0.06 | 0.30 ± 0.06 |
| | **Hom + Mean** | **0.51 ± 0.12** | **0.29 ± 0.09** | **0.26 ± 0.08** | **0.25 ± 0.08** | **0.23 ± 0.09** |
| | **Dense** | 183.8 ± 38.6 | 163.9 ± 42.9 | 161.6 ± 42.9 | 160.9 ± 41.5 | 159.6 ± 39.6 |
| *Control Cost* | **Hom** | 147.3 ± 22.9 | 149.2 ± 34.5 | 135.9 ± 27.9 | 131.6 ± 32.8 | 130.2 ± 35.6 |
| | **Hom + Mean** | **135.3 ± 31.2** | **119.8 ± 17.5** | **122.7 ± 19.2** | **118.7 ± 20.8** | **111.7 ± 24.9** |

## I.6 ABLATION STUDY ABOUT FEATURE DIMENSION

We conduct an ablation study to investigate the impact of feature dimension on control performance in Power-Grid. As shown in Table 12, increasing the feature dimension from 8 to 32 consistently improves both control accuracy and control cost in both noiseless and noisy settings. The best performance is achieved at **Dim = 32**, which yields the lowest control error and cost, indicating an optimal trade-off between representation capacity and learning complexity. Interestingly, while increasing the dimension to 64 slightly improves performance in the noiseless case, it leads to *notable degradation under noise*, particularly in terms of control cost. This suggests diminishing returns from higher dimensions and potential overfitting. Moreover, the *curse of dimensionality* in optimization may play a role, making convergence more difficult and requiring more iterations for the optimizer to find an effective solution.

Table 12: Comparison of control error and cost across different feature dimensions without and with 2% standard deviation noise in Power-Grid environment.

|  | Noiseless | 2% |
|---|---|---|
| *Control Error* |  |  |
| Ours (Dim = 8) | $0.24 \pm 0.008$ | $0.29 \pm 0.011$ |
| Ours (Dim = 16) | $0.23 \pm 0.007$ | $0.29 \pm 0.009$ |
| Ours (Dim = 32) | $\mathbf{0.21 \pm 0.005}$ | $\mathbf{0.27 \pm 0.018}$ |
| Ours (Dim = 64) | $0.20 \pm 0.008$ | $0.28 \pm 0.013$ |
| *Control Cost* |  |  |
| Ours (Dim = 8) | $8.74 \pm 2.78$ | $9.80 \pm 2.84$ |
| Ours (Dim = 16) | $6.74 \pm 1.64$ | $7.44 \pm 0.44$ |
| Ours (Dim = 32) | $\mathbf{4.77 \pm 0.61}$ | $\mathbf{4.99 \pm 0.69}$ |
| Ours (Dim = 64) | $10.45 \pm 2.44$ | $13.56 \pm 1.56$ |

## I.7 ILLUSTRATION OF CONTROL RESULTS

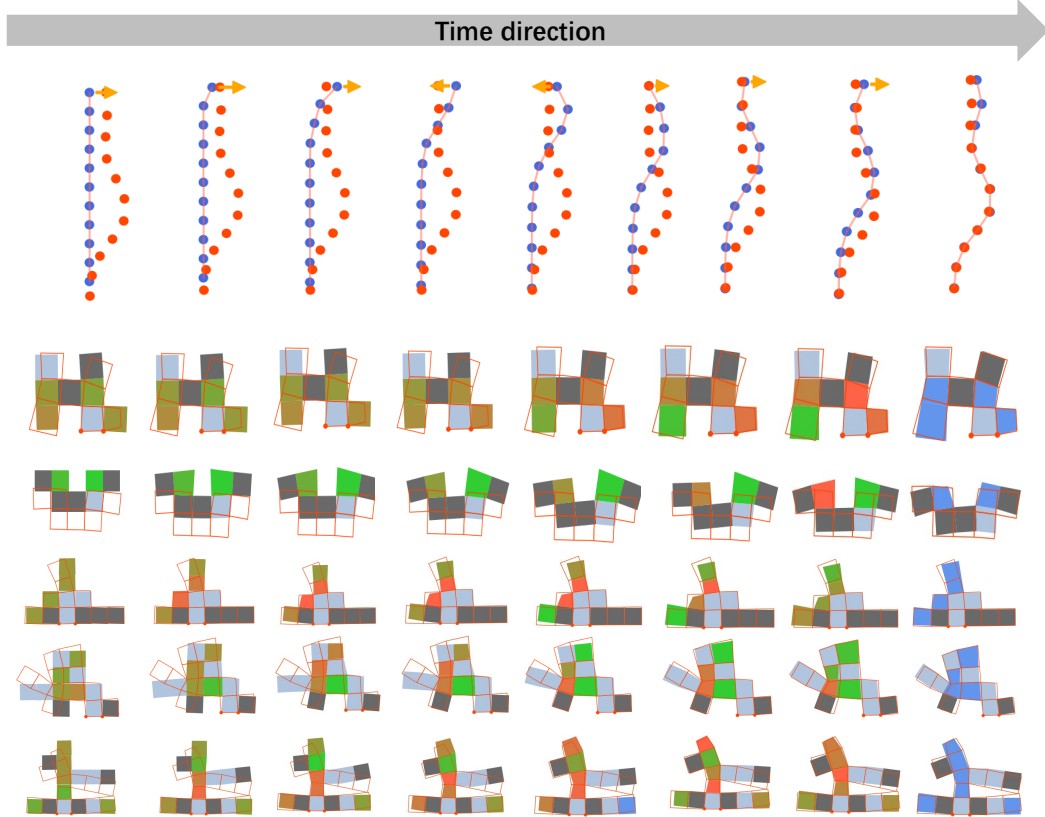

Figure 13: Examples of control performance visualization for Rope, Soft, and Swim. From left to right: the first snapshot shows the initial state, followed by the progression towards the target within a fixed time horizon (indicated by the orange circle in Rope, and the orange box in Soft and Swim).

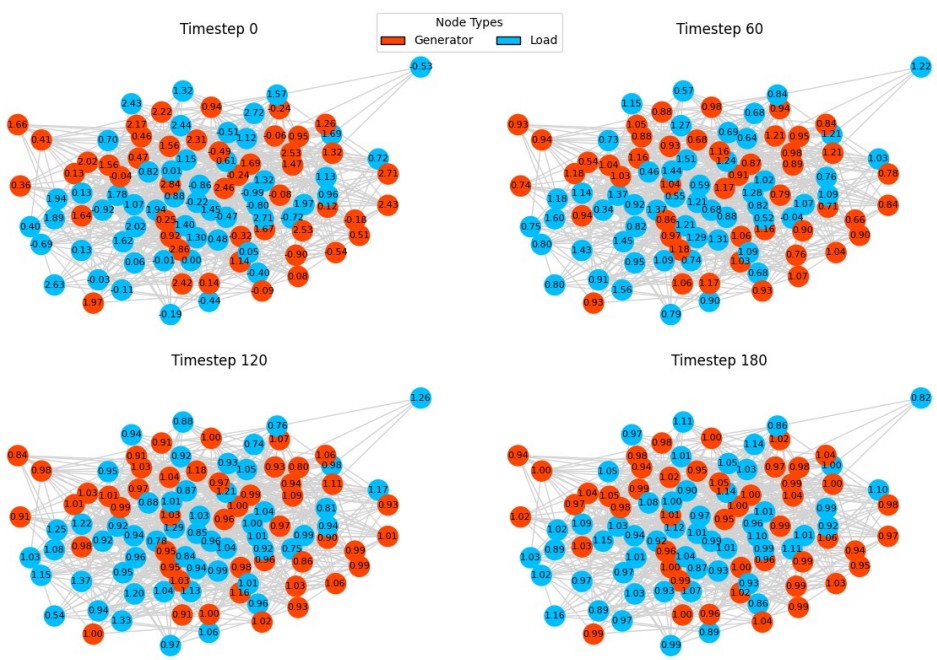

Figure 14: Control visualization of a 102-node power grid over 4 temporal slices (0.05s per step). Blue and red nodes represent loads and generators, respectively, with values indicating voltage magnitude (p.u.). Under the control strategy, voltage magnitudes gradually stabilize towards the reference value of 1 p.u.

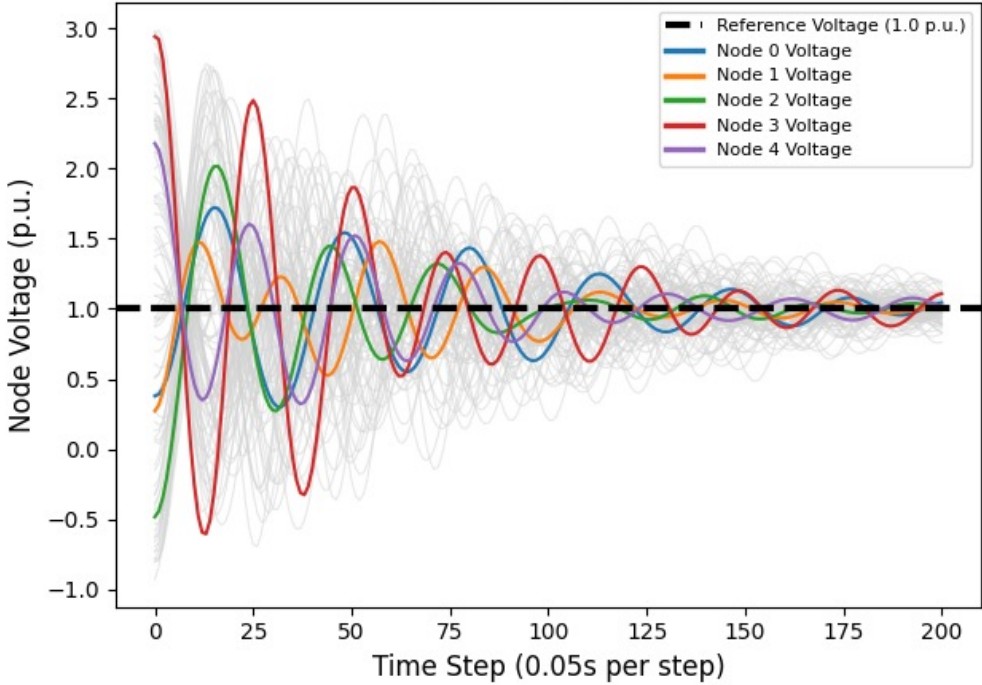

Figure 15: Node-wise power grid voltage trajectories over time for the visualized graph topology, showing the stabilization of voltage magnitudes towards the reference value of 1.0 p.u.

