# OpenReview forum: "From Embedding to Control: Representations for Stochastic Multi-Object Systems"
_ICLR.cc/2026/Conference — ICLR 2026 Poster_

### Official Review · Reviewer_5oTz · 2025-10-28

**Soundness:** 3
**Presentation:** 2
**Contribution:** 3
**Rating:** 4
**Confidence:** 4

**Summary:**

The paper considers the problem of controlling stochastic nonlinear dynamics with interacting objects, and proposes a graph controllable embedding framework to map the dynamics distribution to RKHS. The authors prove asymptotic convergence given sufficient samples, and show improved control cost/error over existing methods on a set of numerical examples.

**Strengths:**

1. The idea of extending RKHS for nonlinear control to stochastic and multi-object settings is novel. However, such contribution is not very clear given the presentation of the paper and lack of discussion and comparison with closely related work (see weakness 2)

2. The paper is in general comprehensive, with theoretical analysis of the convergence of the proposed method, and empirical studies compared to several baselines.

**Weaknesses:**

1. The claim on provably reduced sample complexity is not rigorous. It seems that the reduced complexity is through mean field approximation in the algorithm design. There is no formal proof that such an approximation will not degrade performance.

2. RKHS for nonlinear control has been studied in the community, and the paper seems to miss discussion and comparison with those closely related work [1-4].

3. The explanation of the benchmark problems is missing key details. The paper considers stochastic control, but only one example has noise (power grid), and how noise comes into the dynamics is not given. For benchmark problems in [5], it seems no noises are in the dynamics, unless the authors adapted the setting, though not mentioned in the paper.

[1] Thorpe, Adam J., and Meeko MK Oishi. "Stochastic optimal control via Hilbert space embeddings of distributions." 2021 60th IEEE Conference on Decision and Control (CDC). IEEE, 2021.

[2] Romao, Licio, Ashish R. Hota, and Alessandro Abate. "Distributionally robust optimal and safe control of stochastic systems via kernel conditional mean embedding." 2023 62nd IEEE Conference on Decision and Control (CDC). IEEE, 2023.

[3] Rawlik, Konrad, Marc Toussaint, and Sethu Vijayakumar. "Path integral control by reproducing kernel Hilbert space embedding." arXiv preprint arXiv:1208.2523 (2012).

[4] Bevanda, Petar, et al. "Nonparametric control Koopman operators." arXiv preprint arXiv:2405.07312 (2024).

[5] Li, Yunzhu, et al. "Learning compositional koopman operators for model-based control." arXiv preprint arXiv:1910.08264 (2019).

**Questions:**

1. Why do you choose a simple LQR for control in the embedding space? How would MPC perform in this space?

2. How does the proposed method compare with [1-4] in weakness 2?

---

> ### Author Response · Authors · 2025-11-20
> **Response 1**
>
> We thank reviewer 5oTz for the thoughtful review, constructive feedback, and recognition of our work’s strengths.
>
> **weakness 1** `The claim on provably reduced sample complexity is not rigorous. It seems that the reduced complexity is through mean field approximation in the algorithm design. There is no formal proof that such an approximation will not degrade performance.`
>
> Thanks for the question. We would like to clarify what is formally guaranteed by our approximation.
>
> Our claim is: "Hom + Mean strikes a balance between sample efficiency and expressiveness by approximating the probabilistic structure with adaptive weights, while avoiding the misspecification in Hom and the combinatorial complexity in Tensor and Dense" in Line 296-299.
>
> **The mean-field approximation has formal theoretical gaurantee.** The adaptive Gibbs–Boltzmann weights in mean-field approximation directly arise from the probabilistic factorization in Eq. 25, and Eq. 27 shows their exact RKHS representation (please see details in Appendix D.5). Because the approximation is obtained through this explicit derivation, it formally preserves the non-uniform pairwise interactions encoded in the original model.
>
> **The reduced sample complexity is formally proven.** Appendices F.3 and F.4 provide a non-asymptotic error bound for the approximated model. By reducing the operator from $\mathcal{O}(N^2)$ parameters (Dense/Tensor) to $\mathcal{O}(1)$, the estimation error term is reduced significantly.
>
> **The empirical performance verifies our claims.** “Hom+Mean’’ consistently outperforms both “Dense’’ and “Tensor’’ models in long-term rollout (see Table a). Meanwhile, "Hom+Mean" needs less inference time.
>
> _Table a: Prediction Accuracy and Inference Time Comparison_
>
> | Parametrization | 10-step NRMSE | 20-step NRMSE | 40-step NRMSE | 50-step NRMSE | Inference Time |
> |-----------------|---------------|---------------|---------------|---------------|----------------|
> | **Hom+Mean**    | 5.1 ± 0.4     | **6.5 ± 0.7** | **7.4 ± 1.5** | **8.7 ± 1.9** | **0.42 ± 0.07** |
> | **Dense**       | 4.8 ± 0.6     | 6.8 ± 1.1     | 8.3 ± 1.6     | 9.8 ± 2.3     | 0.86 ± 0.11     |
> | **Tensor**      | **4.7 ± 0.4** | 6.9 ± 0.9     | 8.7 ± 1.8     | 10.5 ± 2.6    | 1.25 ± 0.18     |

---

> ### Author Response · Authors · 2025-11-20
> **Response 2**
>
> **weakness 2** `RKHS for nonlinear control has been studied in the community, and the paper seems to miss discussion and comparison with those closely related work [1-4].`
>
>
> Thank you for providing these references. We have added a discussion of them to our related work. To clarify the difference, we provide a structured comparison with [1–4] below.
>
>
> _Summary Table of Related Works [1–4] and Key Differences_
>
>
> | Paper | Scope | Method | Algorithm | Key Differences from Our Work |
> |-------|--------|----------|------------|-------------------------------|
> | **[1] Thorpe & Oishi (2021)** | Single-object stochastic control | RKHS embeddings | Linear/dynamic programming in RKHS | Single-object only, no interactions; No GNN/mean-field modeling; Not scalable to large or varying graphs; No feature-space linear control for multi-object systems |
> | **[2] Romao et al. (2023)** | Single-object robust and safe stochastic systems | dynamics via conditional kernel mean embedding (CKME); maximum mean discrepancy (MMD) ambiguity sets | Min–max dynamic programming | Robust safety focus, single-object setting; No multi-object or pairwise interactions; No embedding-based linear control; No graph/topology generalization |
> | **[3] Rawlik et al. (2012)** | Single-agent continuous-time | RKHS covariance operators | Path-integral techniques | Continuous-time single-agent vs. discrete multi-object; No GNN relational encoding; No mean-field approximation techniques |
> | **[4] Bevanda et al. (2024)** | Single-object control-affine | Nonparametric Koopman operator, Kernel ridge regression | Koopman MPC | Koopman prediction for single object only; No modeling of interacting multi-object dynamics; No GNN/mean-field approximation |
>
>
> **Condensed Difference Summary**
>
> **1. Scope: Multi-object vs. Single-object**
> All prior works [1–4] address **single-object** stochastic systems.
> Our framework targets **stochastic multi-object dynamics** with explicit interactions in graphs.
>
> **2. Method: RKHS + GNN + Mean-field vs. Pure Kernel Methods**
> Prior works rely purely on **kernel statistical learning**.
> Our method integrates RKHS embeddings with **GNN encoders + mean-field adaptive weighting**, enabling:
> - modeling non-uniform interactions
> - generalization to unseen topologies
> - few-shot adaptability
>
> **3. Algorithm: Scalable Linear Control (LQR) in multi-object systems vs. dynamic programming (DP) / MPC / Path-integral in single-object system**
> Prior works use DP, min–max DP, path-integral control, or MPC for single-object system.
> Our method enables **linear control** in the learned feature spaces that scales to **large and random graph structures**, a capability absent in [1–4].
>
> Inspired by your comments, we have modified our Introduction Section to include the analysis of these four papers in the revised version. Please check the attached revised version in blue color.
>
> [1] Thorpe, Adam J., and Meeko MK Oishi. "Stochastic optimal control via Hilbert space embeddings of distributions." 2021 60th IEEE Conference on Decision and Control (CDC). IEEE, 2021.
>
> [2] Romao, Licio, Ashish R. Hota, and Alessandro Abate. "Distributionally robust optimal and safe control of stochastic systems via kernel conditional mean embedding." 2023 62nd IEEE Conference on Decision and Control (CDC). IEEE, 2023.
>
> [3] Rawlik, Konrad, Marc Toussaint, and Sethu Vijayakumar. "Path integral control by reproducing kernel Hilbert space embedding." arXiv preprint arXiv:1208.2523 (2012).
>
> [4] Bevanda, Petar, et al. "Nonparametric control Koopman operators." arXiv preprint arXiv:2405.07312 (2024).
>
>
> **weakness 3** `The explanation of the benchmark problems is missing key details. The paper considers stochastic control, but only one example has noise (power grid), and how noise comes into the dynamics is not given. For benchmark problems in [5], it seems no noises are in the dynamics, unless the authors adapted the setting, though not mentioned in the paper.`
>
>
> Thanks for the question. To clarify: **we did include noise in the Rope, Soft, and Swim benchmarks.**
>
> **Where and how the noise is added.** These environments are deterministic in the original benchmark, so we introduce stochasticity to evaluate our model under noisy sensing. We add zero-mean Gaussian noise in dynamics with a standard deviation equal to 10% of the empirical standard deviation (see Line 1962-1964).
>
> We will include a clear description of this noise setup in the experimental section to avoid confusion.
>
> [5] Li, Yunzhu, et al.
> "Learning compositional koopman operators for model-based control.
> " arXiv preprint arXiv:1910.08264 (2019).

---

> ### Author Response · Authors · 2025-11-20
> **Response 3**
>
> **question 1**  `Why do you choose a simple LQR for control in the embedding space? How would MPC perform in this space?`
>
> Thank you for this question. Our choice of LQR follows directly from the design of GCE: the embedding space is learned to make the dynamics linear, so simple linear controllers can be applied effectively. LQR is therefore a natural selection, not a limitation of the framework.
>
> **How MPC performs in this space:** Consistent with the theory, when we implemented MPC (20-step receding horizon) using the same learned linear dynamics in the embedding space, it achieved similar performance to LQR in different environments:
>
> _Table a reports the control errors under MPC and LQR_
>
> | Method        | Rope         | Soft         | Swim         |
> |---------------|--------------|--------------|--------------|
> | **Ours (LQR)** | 0.26 ± 0.008 | 0.13 ± 0.05  | 0.41 ± 0.08  |
> | **Ours (MPC)** | 0.25 ± 0.010 | 0.14 ± 0.06  | 0.40 ± 0.11  |
>
>
> Results in Table a match the theory: In linear systems with quadratic cost, LQR and MPC yield the same optimal control behavior, which explains why their performance in the embedding space is similar.
>
> **question 2**  `How does the proposed method compare with [1-4] in weakness 2?`
>
> Please see our response to your  `Weakness 2 `.
>
> ----
> Many thanks again for your time and consideration. Please let us know if we have addressed the concerns and increased your confidence in our work.

---

> > ### Author Response · Authors · 2025-11-24
> >
> > Dear Reviewer 5oTz,
> >
> > We sincerely appreciate the time and effort you have devoted to reviewing our work. Your insights and comments mean a great deal to us. We greatly value this opportunity to engage in a deeper dialogue with you, and we hope that through our responses and further clarifications, you will gain increased confidence in our methodology, results, and the contributions we aim to make.
> >
> > We are truly eager to communicate with you, address any concerns you may have, and provide any additional information that could help strengthen your understanding of our work. Your feedback is invaluable to us, and we are committed to refining and improving our research based on your thoughtful suggestions.
> >
> > Thank you again for your careful evaluation. We look forward to continued constructive exchange with you.
> >
> > Warm regards,
> >
> > Authors from submission # 8143

---

### Official Review · Reviewer_imNB · 2025-10-29

**Soundness:** 3
**Presentation:** 3
**Contribution:** 3
**Rating:** 6
**Confidence:** 2

**Summary:**

This paper introduces Graph Controllable Embeddings (GCE) to model and control stochastic, nonlinear, multi-object systems. The core idea is to use Hilbert space embeddings to represent the system's stochastic dynamics (probability distributions) within a Reproducing Kernel Hilbert Space (RKHS). The central insight is that in this space, the complex dynamics become linear, permitting simple LQR control. To implement this efficiently, GCE integrates Graph Neural Networks (GNNs) with a mean field approximation. This novel approach uses adaptive, non-uniform weights to capture complex inter-object dependencies, breaking the "uniform neighbor" assumption of prior work and achieving provably low sample complexity. Experiments on robotics and large-scale, random-topology power grids demonstrate GCE is scalable, robust to noise, generalizes to unseen topologies, and outperforms baselines in both in-distribution and few-shot tests.

**Strengths:**

* The paper is exceptionally clear and well-written. It provides a strong motivation by precisely identifying the limitations of existing deterministic and non-controllable methods. The proposed framework is derived logically, making the theoretical foundations accessible.

* The work is built on a strong theoretical foundation. It provides a principled framework for stochastic multi-object control by extending Hilbert space embeddings (RKHS) to handle probability distributions of system dynamics, a non-trivial extension of prior deterministic approaches.

* The proposed "Hom + Mean" model is a significant innovation. By introducing adaptive Boltzmann-Gibbs weights to a mean-field approximation, the method effectively captures non-uniform neighbor interactions. This is a well-motivated improvement over prior methods that rely on misspecified uniform weighting.

* The experimental validation is thorough and compelling. The method is tested across diverse environments, including a challenging large-scale, random-topology, and noisy power grid simulation. The results convincingly demonstrate the model's scalability and superior robustness compared to baselines that fail under noise. Ablation studies on sample efficiency further validate the theoretical benefits of the model.

**Weaknesses:**

* The "homogeneity" assumption, which uses a single shared operator for history dynamics, appears to contradict its application to environments with explicitly heterogeneous object types (like 'Soft' and 'Rope'). It is unclear how one operator can model the distinct dynamics of different object types.

* The scalability claims are weakened by the model's design. While the mean-field approximation reduces the history term's complexity, the action term remains dense and scales quadratically with the number of objects, creating a potential bottleneck for large-scale systems.

* The adaptive weighting mechanism seems sensitive to implementation choices. The paper reports that a flexible, neural-network-based potential function was "unstable," and the best performance relied on a specific kernel choice (Gaussian). This suggests potential fragility and limits the generality of the adaptive component.

**Questions:**

1. How is the "homogeneity" assumption (a single shared history operator) reconciled with the clear object-type heterogeneity present in the 'Soft' and 'Rope' environments?

2. The action term's complexity remains quadratic, which appears to be the true scalability bottleneck. Could you elaborate on this design choice and whether a mean-field approximation for actions was also considered?

3. Could you provide more insight into the "unstable" behavior of the neural potential function? Does this suggest a fundamental difficulty in learning such energy functions in RKHS, or was it a more straightforward optimization challenge?

4. The framework's theoretical guarantees hinge on the use of characteristic kernels. How is this property ensured, or at least encouraged, when the kernel features are learned by the GNN encoder rather than being predefined?
5. The linearization is strongly motivated by its use with LQR, which assumes a quadratic cost. How much of the framework's value is retained for tasks with non-quadratic costs, where LQR is inapplicable and other methods (like MPC) must be used?
6. Eq. (6) replaces ψ_h ⊗ ψ_a with concatenation and neglects higher-order interactions. Can you bound or empirically assess the accuracy–complexity trade-off?”

---

> ### Author Response · Authors · 2025-11-20
> **Response 1**
>
> Thank you very much for the positive and encouraging feedback. We truly appreciate your recognition of the clarity of our presentation, the theoretical grounding of our framework, and the significance of our adaptive mean-field design. Your comments on the strength of our experiments and the practical impact of our contributions are highly motivating.
>
> **weakness 1 and question 1** `The "homogeneity" assumption, which uses a single shared operator for history dynamics, appears to contradict its application to environments with explicitly heterogeneous object types (like 'Soft' and 'Rope'). It is unclear how one operator can model the distinct dynamics of different object types.`
>
> `How is the "homogeneity" assumption (a single shared history operator) reconciled with the clear object-type heterogeneity present in the 'Soft' and 'Rope' environments?`
>
> Thanks for the question. We clarify your concerns as follows:
>
> **Objects with different types (captured by the GNN embeddings).** GNNs encode each object’s type and state (e.g., the top mass in Rope, or rigid/soft/actuated/fixed parts in Soft) into distinct embedding features. These embedding features preserve the physical differences among objects.
>
> **Operator is shared due to a same physical rule.** The shared operator $\mathcal{C}$ represents one universal law of interaction.
>
> > An analogy might be helpful: Newton's law of universal gravitation, $F = G * (m_1*m_2)/r^2$, is a single, "homogeneous" law. However, it produces different dynamics for a planet orbiting a star versus a moon orbiting a planet. The difference comes from the inputs to the law (the masses $m_1$, $m_2$, and distance $r$), not from the law itself.
>
> Similarly, our shared operator $\mathcal{C}$ is the single, learned "law." It operates on the embedding features. In Rope environment, since the embedding features for a "top mass" and a "regular mass" are different, applying the same operator $\mathcal{C}$ to them will naturally produce different dynamic evolutions, just as the paper's results show.
>
> **Interactions remain non-uniform (via adaptive Gibbs-Boltzmann weights)** The weights depend on the potential energy functions between embedding features, so different object types influence each other to different degrees.
>
> Inspired by your constructive comments, we have modified our Footnote 2 on Page 5 of our paper to provide a more detailed explanation about the "shared operator" in the revised version. Please check the attached revised version in blue color.
>
> **weakness 2** `The scalability claims are weakened by the model's design. While the mean-field approximation reduces the history term's complexity, the action term remains dense and scales quadratically with the number of objects, creating a potential bottleneck for large-scale systems.`
>
> Thanks for the question. The reviewer is correct that the **worst-case** complexity of the action term is $\mathcal{O}(N^2)$. However, this term is not a practical bottleneck for three reasons:
>
> **Action is Typically Sparse:** The $\mathcal{O}(N^2)$ case only occurs in a fully-actuated all-to-all graph system. In practice, only a small subset of nodes are actuated with actions (as in our tasks), making the actual action matrix sparse. This significantly reduces the practical cost.
>
> **The action optimization is low-dimensional and inexpensive:**
>
> > Low dimension: The action feature dimension $d_a$ is much smaller than the state/history feature dimension $d_h$, making the matrices involved in the optimization small.
>
> > Linear Optimization: The optimization itself is a "simple" and computationally cheap convex optimization problem, e.g., using LQR.
>
>
> **Empirical Verification on Large-Scale Graphs:** Our Power-Grid experiments (100–150 nodes) show that the model remains efficient and achieves strong control performance. This empirically confirms that the action term does not introduce a scalability bottleneck in practice.

---

> ### Author Response · Authors · 2025-11-20
> **Response 2**
>
> **weakness 3** `The adaptive weighting mechanism seems sensitive to implementation choices. The paper reports that a flexible, neural-network-based potential function was "unstable," and the best performance relied on a specific kernel choice (Gaussian). This suggests potential fragility and limits the generality of the adaptive component.`
>
> Thanks for the question. Our result does not indicate fragility of the adaptive weighting mechanism; it indicates that joint-learning a neural potential is unstable, whereas kernel potentials are stable.
>
> 1. **Why the neural (MLP) version is unstable:** The instability comes from joint optimization: the GNN encoder and the MLP potential are jointly trained in a self-supervised setting, causing the Boltzmann weights to oscillate. This is an optimization issue, not a limitation of the adaptive weighting mechanism.
>
> 2. **Why kernel potentials are robust:** Kernel functions have fixed functional forms and very few parameters, so they avoid joint training and produce smooth, stable weights. This makes kernel potentials a reliable choice in our setting.
>
> 3. **Why the Method is General (Not Just Gaussian):** We evaluated multiple different kernels (Gaussian, Laplace, vMF); all were stable. Gaussian performed better, but the framework is compatible with many others (e.g., Matern, Rational Quadratic, Fourier kernel).
>
> Therefore, the adaptive weighting mechanism is not fragile; only the choice of potential parameterization affects optimization stability.
>
> **question 2** `The action term's complexity remains quadratic, which appears to be the true scalability bottleneck. Could you elaborate on this design choice and whether a mean-field approximation for actions was also considered?`
>
> Thanks for your insightful questions. For the first part, please refer to our response to your **weakness 2**.
>
> For the second part, the action cannot apply the mean-field approximation. Mean-field averaging would merge all action features into a single term. This eliminates the ability to assign distinct actions to different nodes, which is essential for controllable embeddings and for solving the per-node optimization.
>
> **queation 3** `Could you provide more insight into the "unstable" behavior of the neural potential function? Does this suggest a fundamental difficulty in learning such energy functions in RKHS, or was it a more straightforward optimization challenge?`
>
> Thanks for the question. For the first question, please refer to our response to your `weakness 3`.
>
>
> For the second question, the answer is "NO"; the difficulty is not from our framework, it is from optimization in joint training (see answer Points 1 and 2 in `weakness 3`).
>
> **queation 4** `The framework's theoretical guarantees hinge on the use of characteristic kernels. How is this property ensured, or at least encouraged, when the kernel features are learned by the GNN encoder rather than being predefined?`
>
> Thanks for the question. Using a learned GNN encoder does not violate the characteristic kernel requirement; in fact, it naturally supports it.
>
> **Learned GNN feature maps induce universal (and therefore characteristic) kernels.** Recent results on neural kernel methods [1-5] show that deep neural networks induce universal and therefore characteristic kernels on compact domains. Therefore, representing the kernel via a learned GNN feature map preserves this property.
>
> **Why learning does not break the property.** The characteristic property relies on the richness of the feature space, not on using a hand-crafted kernel. With an expressive encoder, the learned embeddings are rich enough for the kernel to remain characteristic in practice.
>
> **Empirical validation (KPM vs. Ours):** This behavior is consistent with our experiments. KPM uses polynomial features that are known to be non-characteristic and become unstable (often NaN) in noisy or large-scale settings (Table 4 in our paper). In contrast, GCE with learned GNN features remains stable with significantly lower control error. This empirical phenomenon aligns with the above two points, which confirms neural feature maps inducing characteristic kernels.
>
>
> [1] Shimizu, Eiki, Kenji Fukumizu, and Dino Sejdinovic. "Neural-kernel conditional mean embeddings." Forty-first International Conference on Machine Learning, PMLR, 2024
>
> [2] Yang, Yiming, et al. "Tensor-Var: Efficient Four-Dimensional Variational Data Assimilation." Forty-second International Conference on Machine Learning, PMLR, 2025
>
> [3] Sehanobish, Arijit, et al. "Scalable neural network kernels."  International Conference on Learning Representations. 2024
>
> [4] Kidger, Patrick, and Terry Lyons. "Universal approximation with deep narrow networks." Conference on learning theory. PMLR, 2020.
>
> [5] Chen, Lin, and Sheng Xu. "Deep Neural Tangent Kernel and Laplace Kernel Have the Same RKHS." International Conference on Learning Representations. 2020

---

> ### Author Response · Authors · 2025-11-20
> **Response 3**
>
> **question 5** `The linearization is strongly motivated by its use with LQR, which assumes a quadratic cost. How much of the framework's value is retained for tasks with non-quadratic costs, where LQR is inapplicable and other methods (like MPC) must be used?`
>
> Thanks for the question. We clarify that the value of our framework does not rely on the quadratic-cost assumption or on the use of LQR.
>
> **1. The key benefit of GCE comes from the representation of dynamics, not from assuming a quadratic cost.**
> Once the dynamics are linearized in the embedding space, a wide range of controllers (e.g., LQR or MPC) can operate more reliably and efficiently.
>
>
> **2. For non-quadratic costs, MPC can be applied directly in the linear embedding space.** To demonstrate this, we replaced the quadratic action penalty with an $L^1-$norm for the action as a non-quadratic cost (a non-quadratic cost) and used MPC for control. The controller optimizes over the same learned linear transition model, without requiring LQR.
>
> **3. The framework retains its value under non-quadratic costs and non-LQR controllers.** With the $L^1$ cost and MPC, GCE also achieves better performance and robustness than baselines, as shown in Table a. This demonstrates that the benefits of GCE come from the linearized dynamics rather than the form of the cost function.
>
> _Table a (MPC version): Evaluation under varying noise levels in Power-Grid on random graphs with 100–150 objects._
> *MPC uses a 10-step receding horizon. “NaN” indicates unstable control.*
>
> | Method                 | Noiseless           | 2%                 | 5%                 | 10%                | 20%                |
> |------------------------|---------------------|---------------------|---------------------|---------------------|---------------------|
> | KPM (MPC)              | 0.39 ± 0.034    | 0.55 ± 0.041        | NaN                 | NaN                 | NaN                 |
> | CKO (MPC)              | 0.45 ± 0.046        | 0.51 ± 0.059        | 0.57 ± 0.052        | 0.74 ± 0.071        | 0.96 ± 0.083        |
> | **Ours (Gaussian, MPC)** | **0.20 ± 0.009**     | **0.28 ± 0.024**     | **0.41 ± 0.033**     | **0.70 ± 0.048**     | **0.93 ± 0.057**     |
>
>
> **question 6** `Eq. (6) replaces ψ_h ⊗ ψ_a with concatenation and neglects higher-order interactions. Can you bound or empirically assess the accuracy–complexity trade-off?”`
>
> Thanks for the question. We assess the accuracy–complexity trade-off from three points.
>
> **Complexity savings:** Replacing ψ_h ⊗ ψ_a with concatenation avoids forming a Kronecker-product tensor and significantly reduces the operator's parameters to learn. This reduces both computational and sample complexity (see Table 1 in our paper). Empirically, “Hom+Mean” is ~3× faster than the Tensor model (Table b), consistent with the theoretical reduction in complexity.
>
>
> **Accuracy (Long-Horizon Prediction):** Empirically, ”Tensor“ gains a small short-horizon advantage from its higher parameter count, but its large parameterization causes high variance and unstable long-horizon predictions. This causes prediction errors to accumulate much faster:
>
> _Table b: Comparison of Prediction Accuracy and Inference Time_
>
> | Parameteization | 10-step NRMSE     | 50-step NRMSE     | Inference Time  |
> |-----------------|-------------------|-------------------|------------------|
> | Hom+Mean        | 5.1 ± 0.4         | 8.7 ± 1.9         | 0.42 ± 0.07      |
> | Tensor          | 4.7 ± 0.4         | 10.5 ± 2.6        | 1.25 ± 0.18      |
>
> The small expressiveness gain from ψ_h ⊗ ψ_a is outweighed by worse long-horizon stability and much higher computational cost.
>
>
> **Suitability for control:** More importantly, the tensor form entangles history and action features, which makes sequential action optimization computationally intractable. In contrast, the proposed "Dense" is suitable for control by concatenation.
>
> ---
> Many thanks again for your time and consideration. Please let us know if we have addressed the concerns and increased your confidence in our work.

---

> > ### Comment · Reviewer_imNB · 2025-11-22
> >
> > Thanks for the update. I have a quick follow-up on Q5. If the framework is compatible with MPC anyway, and considering that nonlinear MPC with deep dynamics is already a go-to solution, I'm struggling to see the value proposition of the RKHS linearization here. It seems we are accepting a loss of expressiveness through approximations to achieve linearity. My question is: is this trade-off really worth it if standard MPC can handle the nonlinear dynamics directly without such constraints?
> >
> > For now, I am inclined to keep my score.

---

> > > ### Author Response · Authors · 2025-11-22
> > > **Quick Response to Reviewer imNB**
> > >
> > > Thank you for the thoughtful comment. To clarify, the RKHS-based linearization is not meant to make the dynamics more expressive than a nonlinear deep model. Its main purpose is to impose structure so that the resulting optimization becomes simpler and more tractable.
> > >
> > > In contrast, if we use a deep network to directly model the dynamics, obtaining gradients in optimization would require frequent auto-differentiation through the full model, which can make the optimization much heavier. The linear structure avoids this issue, which is another practical advantage.
> > >
> > > We also observed an interesting phenomenon in our experiments: once the dynamics are linearized in feature space, we can achieve few-shot adaptation to new graphs or topologies by optimizing only a small linear layer (linear dynamics), leading to efficient generalization. This explains why this linear way is effective in such setting.
> > >
> > > Overall, this is indeed a trade-off: we give up some nonlinear expressiveness in exchange for more structured dynamics, cheaper gradients, and significantly simpler optimization, which can be beneficial in our setting.

---

> > > > ### Comment · Reviewer_imNB · 2025-11-22
> > > >
> > > > Cool, I will keep my score.

---

> ### Author Response · Authors · 2025-11-23
> **Follow up the comments from Reviewer imNB**
>
> `If the framework is compatible with MPC anyway, and considering that nonlinear MPC with deep dynamics is already a go-to solution, I'm struggling to see the value proposition of the RKHS linearization here. It seems we are accepting a loss of expressiveness through approximations to achieve linearity. My question is: is this trade-off really worth it if standard MPC can handle the nonlinear dynamics directly without such constraints?`
>
> We would like to reply to your question from another point of view. Before, in our original response to your Q5, we compare MPC and LQR performance under our GCE framework; now, we compare GCE directly with standard MPC with deep dynamics in the following.
>
> By “standard MPC,” we refer specifically to nonlinear MPC. For MPC to perform well, it requires an accurate, or at least a good model of the underlying dynamics, which involves stochastic, multi-object control with non-uniform interactions in our paper. We think it is challenging.
>
> If we rely on learned dynamics, “deep dynamics” here essentially corresponds to neural networks, and more specifically, graph neural networks (GNNs) for our problem. Thus, the core question becomes:
>
> > Can a GNN learn a good model for our problem?
>
> Our answer is **NO** in the sense that a GNN can indeed learn *a* model of the system, it cannot learn a good model that is sufficiently reliable. The reasons are as follows:
>
>
> **1. Consistency in Stochastic Environments** Can a GNN model robustly handle uncertainty? The GCE framework is explicitly designed to model the inherent stochasticity of the system.
>
> **The Problem in GNN:**
> The GNNs, being primarily function approximators, do not guarantee to learn the structural properties for a multi-object system for MPC. To be more specific, they lack the characteristic-kernel structure required to uniquely and consistently represent conditional probability distributions of multi-object dynamics. This probabilistic inconsistency fundamentally limits their reliability under noise.
>
> **The GCE Solution:**
> GCE employs Hilbert space embeddings of conditional distributions. This mathematically rigorous foundation ensures probabilistic consistency and convergence (Theorem 1), meaning the learned embedding captures and converges to the true underlying stochastic dynamics.
>
>
> **2. Computational Tractability**
>
> The complexity of solving nonlinear MPC is often the bottleneck, especially for multi-object systems.
>
> **The Problem in GNN:**
> A purely nonlinear GNN model requires nonlinear MPC to use computationally expensive techniques, such as using nonlinear optimization solvers to iteratively solve non-convex optimization problems or performing repeated local linearization (e.g., auto-differentiation) at every time step to find the optimal action sequence. This is difficult in practice.
>
> **The GCE Solution:**
> The RKHS linearization is a principled trade-off that yields a substantial practical benefit: it converts the dynamics into a linear structure in feature space, allowing the entire convex optimization to be carried out directly in this space. This enables a simple and tractable linear control strategy **without** requiring auto-differentiation or repeated nonlinear solving, making sequential action optimization computationally feasible.
>
>
> **3. Scalability & Generalization** The critical limitation of standard GNNs in the nonlinear MPC is their inefficiency and poor generalization when performing few-shot adaptation to systems with unseen or randomized graph structures.
>
> **The Problem:** The GNN models, especially when scaled up, suffer from high parameter counts and computational complexity. This inherent inefficiency—scaling as quadratically in parameter count, makes them sample inefficient to generalize unseen and random topologies.
>
> **The GCE Solution:** GCE, and specifically the "Hom + Mean" design, introduces an efficient mechanism for few-shot adaptation:
>
> - **Provably low sample complexity:**
>   The mean-field approximation replaces neighbor-specific operators with a single shared operator for the history component, drastically reducing sample requirements and enabling few-shot adaptation.
> - **Adaptive generalization:**
>   Expressiveness is preserved through explicit adaptive Boltzmann–Gibbs weights, allowing the model to remain flexible while still scalable.
>
>
> To sum up, the value of GCE lies in its ability to directly overcome the fundamental limitations of the GNN-based models within MPC, transforming an otherwise intractable, probabilistically inconsistent, and unscalable nonlinear MPC problem into a consistent, tractable, scalable, and well-generalized control framework. Yes, we accept a loss of expressiveness through approximation in order to achieve linearity; given the above three advantages of GCE, this trade-off is clearly worthwhile.
>
> Could you re-evaluate our response once again?

---

> > ### Comment · Reviewer_imNB · 2025-11-24
> >
> > Thanks for their detailed follow-up response. My specific technical queries have been addressed by the clarifications provided.
> > After carefully considering the overall novelty and significance of the proposed method in the broader context of the field, I have decided to maintain my initial evaluation.

---

### Official Review · Reviewer_UzG7 · 2025-10-31

**Soundness:** 4
**Presentation:** 4
**Contribution:** 3
**Rating:** 6
**Confidence:** 3

**Summary:**

This paper introduces Graph Controllable Embeddings (GCE), a general framework for modeling and controlling stochastic multi-object systems with unknown nonlinear dynamics. The core idea is to leverage Hilbert space embeddings to represent the system's stochastic dynamics in a Reproducing Kernel Hilbert Space (RKHS), where the evolution becomes linear and amenable to control. The framework innovatively combines Graph Neural Networks to encode relational structure with a mean-field approximation. This allows for adaptive, non-uniform weighting of inter-object interactions, overcoming a key limitation of prior work. The method is supported by rigorous theoretical guarantees and demonstrates superior performance, especially in few-shot and noisy settings.

**Strengths:**

The paper's contribution is well-supported by its technical components. The proposed framework is built upon the established theory of Kernel Bayes' Rule and provides a principled way to unify several graph-based embedding approaches under a single lens. Its core technical contribution is the introduction of a mean-field approximation to model adaptive, non-uniform interactions, which addresses a specific, documented limitation of prior methods like CKO. The paper includes theoretical proofs for the convergence and sample complexity of the proposed estimators. The claims are further substantiated through a series of experiments on diverse control tasks, including comparisons against relevant baselines.

**Weaknesses:**

- The most efficient and recommended model variant, `Hom+Mean`, relies on a strong homogeneity assumption where all objects share the same underlying dynamics operator $C_{O|H}$. This may limit its applicability to highly heterogeneous multi-agent systems, where different types of agents (e.g., a mix of ground robots and aerial drones) possess fundamentally different transition models. The experiments, while strong, do not feature such a deeply heterogeneous environment to test this boundary.

- The framework is fundamentally limited to modeling interactions as a composition of pair-wise relationships. While using a multi-layer GNN allows information to propagate across multiple hops, it cannot capture true, non-decomposable higher-order interactions (e.g., physical effects governed by the angle between three bodies, which is not reducible to a sum of pairs). This restricts the method's utility in domains where many-body physics or complex group dynamics are dominant. The authors rightly acknowledge this as a limitation for future work.

- The current experiment seems to lack a direct contrast with a powerful nonlinear Model-Based RL baseline, and such a contrast might more powerfully demonstrate the necessity of the design choice of forced linearization.

**Questions:**

- The history is defined as the observation at the previous moment, which is a standard first-order Markov assumption. Is this simple historical representation sufficient for systems that require longer historical dependencies to make accurate predictions (for example, systems with momentum or hysteresis effects)?

---

> ### Author Response · Authors · 2025-11-20
> **Response 1**
>
> Thank you very much for the positive and encouraging feedback. We truly appreciate your recognition of the technical soundness of our framework, the clarity of the presentation, and the strength of our contribution. We are especially grateful that you highlighted the principled integration of RKHS embeddings, the role of Kernel Bayes’ Rule, and the significance of our adaptive mean-field approximation in addressing limitations of prior methods. Your acknowledgement of our theoretical analysis and comprehensive experimental validation is highly motivating. Thank you again for the thoughtful and constructive review.
>
>
> **weakness 1** `The most efficient and recommended model variant Hom+Mean, relies on a strong homogeneity assumption where all objects share the same underlying dynamics operator. This may limit its applicability to highly heterogeneous multi-agent systems, where different types of agents (e.g., a mix of ground robots and aerial drones) possess fundamentally different transition models. The experiments, while strong, do not feature such a deeply heterogeneous environment to test this boundary.`
>
>
> We thank the reviewer for the question.
>
> **The GCE Framework Provides Forms for Full Heterogeneity.** "Hom+Mean" is our most efficient variant, but it is not the only variant our GCE framework offers. For a "highly heterogeneous" system where every agent is unique and the class-based assumption of "Hom+Mean" is too strong, our framework provides the "Dense" form (Eq. 6 in our paper). The "Dense" form (Table 1 in our paper) learns a unique, pair-wise operator $\mathcal{C}_{O^i|H^j}$ for every agent $i$ and $j$. This form is fully heterogeneous by design and makes no homogeneity assumptions.
>
>
>
> **weakness 2** `The framework is fundamentally limited to modeling interactions as a composition of pair-wise relationships. While using a multi-layer GNN allows information to propagate across multiple hops, it cannot capture true, non-decomposable higher-order interactions (e.g., physical effects governed by the angle between three bodies, which is not reducible to a sum of pairs). This restricts the method's utility in domains where many-body physics or complex group dynamics are dominant. The authors rightly acknowledge this as a limitation for future work.`
>
>
>
> Thanks for this insightful point. We fully agree that modeling non-decomposable higher-order interactions is an exciting next step. Importantly, this limitation comes from the specific GNN encoder we use, not from the GCE framework itself. The controllable embedding formulation places no restriction on extending the operator to hyperedges or higher-order message passing.
>
> **Why this is not a practical limitation of our results:** In the domains we evaluate (power-grid control, multi-object physical system and robotics), interactions are predominantly pairwise or well-approximated by pairwise potentials. The pairwise interaction is a standard assumption and is also used in GNN-based simulators, mean-field models, and message-passing physics engines. This makes the current formulation expressive enough for the tasks studied.
>
> **Why this is a natural extension:** The adaptive Boltzmann weighting mechanism is not inherently restricted to pairs. It can be generalized to group-based potentials, enabling GCE to incorporate hypergraph neighborhoods or higher-order energy terms. Replacing the GNN with hypergraph message passing would allow GCE to capture true many-body interactions while preserving controllable embedding.

---

> ### Author Response · Authors · 2025-11-20
> **Response 2**
>
> **weakness 3** `The current experiment seems to lack a direct contrast with a powerful nonlinear Model-Based RL baseline, and such a contrast might more powerfully demonstrate the necessity of the design choice of forced linearization.`
>
>
> Thanks for the question. We did include a strong nonlinear MBRL baseline TD-MPC [1], a widely used model-based RL method. It performs consistently worse than our approach, and we will add these results in the following (see Table a). This comparison also clarifies why forced linearization is necessary.
>
> _Table a. Performance comparison with TD-MPC_
> | Method        | Rope         | Soft         |
> |---------------|--------------|--------------|
> | **Ours** | **0.26 ± 0.08** | **0.13 ± 0.05**  |
> | **TD-MPC** | 0.31 ± 0.10 | 0.15 ± 0.03  |
>
>
> Our GCE framework has three other advantages compared with model-based RL.
>
> **1. Stability and long-horizon reliability:** Nonlinear MBRL models accumulate error quickly and suffer from unstable, nonconvex policy optimization. Our controllable embedding enables a stable optimization and produces more reliable multi-step control.
>
> **2. Goal Flexibility:** Nonlinear MBRL must be re-optimized from scratch when the cost function changes. In contrast, when the cost function changes, GCE does convex optimization, with no need for any training.
>
> **3. Topological Generalization:** Nonlinear MBRL is trained on one fixed topology. GCE, through shared operators and GNNs, naturally generalizes to larger and unseen graphs (verified in few-shot and power-grid experiments).
>
> In summary, this comparison shows the necessity of our "forced linearization" design. A nonlinear MBRL approach is a tool for a single task; GCE is a framework for flexible, general-purpose control.
>
> [1] Hansen, Nicklas A., Hao Su, and Xiaolong Wang. "Temporal Difference Learning for Model Predictive Control." International Conference on Machine Learning. PMLR, 2022.
>
>
>
> **question 1** `The history is defined as the observation at the previous moment, which is a standard first-order Markov assumption. Is this simple historical representation sufficient for systems that require longer historical dependencies to make accurate predictions (for example, systems with momentum or hysteresis effects)?`
>
>
> We thank the reviewer for raising this point. Our environment follows first-order Markov dynamics, where the next observation depends only on the previous step. Therefore, providing a longer history window (e.g., 3-step or 5-step) does not introduce additional information.
>
> To further verify this, we trained models using 1-step, 3-step, and 5-step history windows. As shown in the results below, using more history does not improve performance. The 3-step variant is very close to the 1-step model, while the 5-step variant becomes slightly worse due to the inclusion of redundant information (see Table b). Therefore, the 1-step history is sufficient for our setting.
>
> For systems with momentum or hysteresis effects, we think the introduction of longer history information can lead a better performance.
>
> _Table b. Performance comparison with different history window lengths_
>
> | History length | 10-step NRMSE | 20-step NRMSE | 40-step NRMSE | 50-step NRMSE |
> |------------------|---------------|---------------|---------------|---------------|
> | 1-step | 5.1 ± 0.4 | 6.5 ± 0.7 | 7.4 ± 1.5 | 8.7 ± 1.9 |
> | 3-step | 5.2 ± 0.4 | 6.6 ± 0.7 | 7.5 ± 1.5 | 8.7 ± 1.8 |
> | 5-step | 5.3 ± 0.5 | 6.8 ± 0.8 | 7.8 ± 1.6 | 9.0 ± 2.0 |
>
>
> -------
> We sincerely thank the reviewer once again for their insightful comments and suggestions. We believe that the revisions made in response have significantly improved the quality of our manuscript. We eagerly anticipate any further feedback.

---

### Official Review · Reviewer_8izv · 2025-10-31

**Soundness:** 3
**Presentation:** 3
**Contribution:** 3
**Rating:** 6
**Confidence:** 4

**Summary:**

This paper proposes a framework named Graph Controllable Embeddings (GCE) to tackle the modeling and controlling problems of stochastic nonlinear dynamics in multi-object systems. It utilizes the linear properties of RKHS to transform complex nonlinear stochastic dynamics into linear relationships within RKHS, allowing the application of well-established linear control methods. GCE also integrates graph neural networks (GNNs) to adapt to dynamic interaction patterns and mean field approximation to reduce sample complexity. The evaluation results validate that GCE outperforms competitive methods.

**Strengths:**

1. The idea is novel. It leverages the linear properties of RKHS to map originally complex nonlinear stochastic dynamics into RKHS, converting them into linear relationships. This further enables the adaptation of mature linear control methods, effectively overcoming the traditional challenges of nonlinear stochastic control for multi-body systems.
2. The theoretical analysis is detailed and comprehensive. Rigorous derivations are provided for aspects including the existence and convergence of the GCE framework, the sample complexity of mean field approximation, and the property comparison of different embedding forms (such as Tensor, Dense, and Hom).
3. The experimental results are comprehensive. It covers multiple scenarios including physical systems (Rope), robotics (Soft, Swim), and power grids (Power-Grid). Additionally, tests are conducted on noise robustness, few-shot generalization, and the effects of different components (e.g., kernel function types, feature dimensions).
4. The presentation is good. The paper is easy to understand.

**Weaknesses:**

1. Regarding the theoretical analysis in this paper, although convergence and consistency are mentioned as a key contribution in the Introduction section, multi-step approximate derivations (such as Equation (6) only using first-order approximation and adaptive mean field approximation) may undermine the aforementioned theoretical characteristics, leading to a significant deviation between practical implementation and theoretical guarantees.
2. Regarding scalability and generalization, although the authors claim that the research can scale to larger graphs and more random topologies, this conclusion relies on the strong assumption of the "shared approximation operator" mentioned in Line 264 of the paper. The rationality of this inductive bias remains to be verified, which results in a lack of guarantees for the effectiveness of the method’s scalability.

**Questions:**

No.

---

> ### Author Response · Authors · 2025-11-20
> **Response 1**
>
> Thank you very much for your positive and encouraging feedback. We sincerely appreciate your recognition of the novelty of our idea, the thoroughness of our theoretical analysis, the comprehensiveness of our experiments, and the clarity of our presentation. Your comments are highly motivating and help reinforce the value of our contributions. Thank you again for the thoughtful review.
>
> **weakness 1** `Regarding the theoretical analysis in this paper.`
>
> Thank you for raising this insightful comment. In fact, our practical "Hom + Mean" model (using Equation 6 and Equation 9) is an approximation of the "ideal" tensor-based model (Equation 5) for which the consistency guarantee (Theorem 1) is derived. These approximations are not a flaw, but rather a principled and necessary series of trade-offs to achieve a model that is simultaneously expressive, scalable, and controllable.
>
> **Justification for the "First-Order Approximation" (Eq. 6).**
>
> This approximation, which disentangles the tensor product $\psi^{h,j}_t \otimes \psi^{a,j}_t$ into a sum form, is the key step to enable control.
>
> 1. _Practical purpose (controllable in feature space)_: The original tensor form (see Eq. 5) makes "sequential action optimization intractable"  because the action and history features are entangled together. The disentangled form (Eq.6) makes the dynamics linear with respect to the action features, which is the definition of a "controllable embedding" (as defined in Footnote 1 on Page 1) and is what permits the use of classic linear control algorithms for action optimization.
>
> 2. _Theoretical Justification (Why it sounds):_ This approximation is not ad-hoc. As stated in the paper in line 223, this approach is "inspired by the decomposition of joint distributions in exponential families". The paper's appendix (D.4) further clarifies that this method builds on previous influential studies such as ([1], page 5, Eq. 10). This technique is a principled simplification, common in related fields, that factorizes the conditional distribution by "neglecting higher-order interactions"  in favor of a tractable, controllable model.
>
> [1] Song, Le, et al. "Hilbert space embeddings of conditional distributions with applications to dynamical systems." Proceedings of the 26th annual international conference on machine learning. 2009.
>
> **Justification for the "Mean Field Approximation" (Eq. 9)**
>
> This approximation, which moves from $\mathcal{O}(N^2)$ pair-wise operators to an $\mathcal{O}(N)$ shared operator with adaptive weights, is justified by superior theoretical sample complexity and empirical performance.
>
> 1. _Practical Validation (Empirical Results):_ The concern about "significant deviation" is addressed by the experimental results.
>
>     > Tables 5 and 11 are the key evidence. In the few-shot (low data) regime, our "Hom + Mean" (with both approximations) achieves significantly lower sample complexity than the "Dense" form which only has the first approximation of "Tensor" form (Eq. 5). With the increase of data, the "Hom + Mean" is also consistently better than the "Dense"  form.
>
> This demonstrates that the mean-field approximation is an effective form. "Hom+Mean" introduced by the approximations is outweighed by the (large) reduction in control error in Tables 5 and 11, leading to a model that is practically better for control, not worse.
>
> 2. _Theoretical Guarantee:_ The reviewer's concern about "undermining theoretical characteristics" is directly addressed by the analysis in Appendix D.5 and F.4-F.5.
>
>     > We explicitly indicate why the mean field approximation is theoretically sound and accurate by connecting statistical learning and reproducing kernel Hilbert space (RKHS) (see details in Line 1009-1057). Thus, our "Hom + Mean" is principled not heuristic.
>
>     > We also provide dedicated non-asymptotic sample complexity bounds for approximated models. Theorems a and b (in Appendix F.4 and F.5) show that our "Hom + Mean" error bound for the history term scales with $\mathcal{O}(N)$, which is provably better than the $\mathcal{O}(N^2)$ scaling of the "Dense" form. This theoretical result is verified by the empirical results in Tables 5 and 10.
>
>
>
> **More Results** To further address the concern of "accumulating deviation," we provide new experimental results: Prediction Error, Inference Time, and Control Performance.
>
> Prediction Error: The new table below shows that our model's prediction error (NRMSE) degrades more slowly over long horizons (Table a). This is expected, as our 'Hom+Mean' model has significantly fewer parameters by introducing shared operator for history part (e.g., $\mathcal{O}(N^2)$ vs $\mathcal{O}(1)$). This reduced complexity leads to better generalization and makes the model more stable against the error accumulation that affects the higher-parameter 'Dense' and 'Tensor' forms, directly refuting the concern about accumulating deviation.

---

> > ### Author Response · Authors · 2025-11-20
> > **Response 2 (the results for weakness 1)**
> >
> > _Table a. Prediction Error: "Hom+Mean" has better prediction accuracy than "Dense" and "Tensor" in long-term rollout in Soft environment._
> >
> > | Parameterization | 10-step NRMSE | 20-step NRMSE | 40-step NRMSE | 50-step NRMSE |
> > |-------------|--------------|---------------|--------------|---------------|
> > | **Hom+Mean**    | 5.1 ± 0.4 | **6.5 ± 0.7** | **7.4 ± 1.5** | **8.7 ± 1.9** |
> > | **Dense**     | 4.8 ± 0.6  | 6.8 ± 1.1 | 8.3 ± 1.6  | 9.8 ± 2.3 |
> > | **Tensor**    | **4.7 ± 0.4** | 6.9 ± 0.9 | 8.7 ± 1.8 | 10.5 ± 2.6 |
> >
> >
> >
> > Inference Time: "Hom+Mean" is 1.5x-3x faster (see Table b), confirming its high computational efficiency.
> >
> > _Table b. Computation time comparison for predicting a trajecotry with different embeddings in Soft environment._
> >
> > | Parameterization | Inference Time |
> > |-------------|--------------|
> > | **Hom+Mean**    | **0.42 ± 0.07** |
> > | **Dense**     | 0.84 ± 0.11  |
> > | **Tensor**    | 1.25 ± 0.18 |
> >
> >
> >
> > _Table c. Control performance comparison in Rope environment. Full results are reported in Table 11 in our paper._
> >
> > (1) Control Error
> > | Method       | 1            | 4            | 8            | 16           | 32           |
> > |--------------|--------------|--------------|--------------|--------------|--------------|
> > | **Dense**    | 0.79 ± 0.25  | 0.41 ± 0.11  | 0.36 ± 0.12  | 0.28 ± 0.08  | 0.26 ± 0.10  |
> > | **Hom+Mean** | **0.51 ± 0.12**  | **0.29 ± 0.09**  | **0.26 ± 0.08**  | **0.25 ± 0.08**  | **0.23 ± 0.09**  |
> >
> > (2) Control Cost
> > | Method       | 1             | 4             | 8             | 16            | 32            |
> > |--------------|---------------|---------------|---------------|---------------|---------------|
> > | **Dense**    | 185.3 ± 38.6  | 163.9 ± 42.9  | 161.6 ± 42.9  | 169.4 ± 41.5  | 159.2 ± 39.6  |
> > | **Hom+Mean** | **135.3 ± 31.2**  | **119.3 ± 17.5**  | **122.7 ± 19.2**  | **118.7 ± 20.8**  | **111.7 ± 24.9**  |

---

> ### Author Response · Authors · 2025-11-20
> **Response 3**
>
> **weakness 2** `Regarding scalability and generalization, although the authors claim that the research can scale to larger graphs and more random topologies, this conclusion relies on the strong assumption of the "shared approximation operator" mentioned in Line 264 of the paper. The rationality of this inductive bias remains to be verified, which results in a lack of guarantees for the effectiveness of the method’s scalability.`
>
>
> Thanks for the question. We clarify the role of the shared operator as follows.
>
> **1. Rationality of Shared Operator.** The shared operator is based on a common property of many graph dynamical systems: the local interaction rule between neighboring elements is the same. Therefore, the operator $\mathcal{C}$ represents one universal law of interaction.
>
> > An analogy might be helpful: Newton's law of universal gravitation, $F = G * (m_1*m_2)/r^2$, is a single, "homogeneous" law. However, it produces different dynamics for a planet orbiting a star versus a moon orbiting a planet. The difference comes from the inputs to the law (the masses $m_1$, $m_2$, and distance $r$), not from the law itself.
>
> Similarly, our shared operator $\mathcal{C}$ is the single, learned "law." It operates on the embedding features. In Rope environment, since the embedding features for a "top mass" and a "regular mass" are different, applying the same operator $\mathcal{C}$ to them will naturally produce different dynamic evolutions.
>
> **2. Objects with different types (captured by the GNN embeddings).** GNN encoder maps each object’s type and observation (e.g., the top mass in Rope, or rigid/soft/actuated/fixed parts in Soft) into distinct embedding features. These embedding features preserve the physical differences among objects.
>
>
> **3. Interactions remain non-uniform (via adaptive Gibbs-Boltzmann weights)** The weights depend on potential energy functions between embeddings, so different object types influence one another differently.
>
>
> Finally, this design directly supports scalability. Because $\mathcal{C}$ is shared, it can be applied repeatedly across graphs of arbitrary size or topology. The model, therefore, generalizes naturally to larger systems and random graph structures, a behavior also verified by our empirical results in Power-Grid environment.
>
>
> Inspired by your constructive comments, we have modified our Footnote 2 on Page 5 of our paper to provide a more detailed explanation about the "shared operator" in the revised version. Please check the attached revised version in blue color.
>
>
> -------
> Thank you again for the thoughtful comments. If the revisions adequately address your concerns, we would greatly appreciate your consideration of a higher score.

---

> > ### Author Response · Authors · 2025-11-26
> >
> > Dear Reviewer 8izv,
> >
> > Thank you once again for your time and for reviewing our rebuttal. Please feel free to let us know if any concerns remain. Our team is fully dedicated to thoroughly addressing all of your comments. We hope that the additional results and clarifications we have provided sufficiently resolve the issues raised, and we would greatly appreciate it if you could consider raising your score accordingly.
> >
> > We sincerely appreciate your efforts and are happy to provide any further information if needed.
> >
> > Best regards,
> >
> > The Authors

---

### Meta-Review · Area_Chair_FAEf · 2026-01-07

**Summary:**

1. All reviewers appreciate the significant novelty of the proposed ideas.

2. All reviewers acknowledge the strong theoretical foundation of the proposed method and the nontrivial and comprehensive theoretical guarantees for the proposed method.

3. All reviewers find the experimental results thorough because they involve various domains, diverse environments, and various baselines. The ablation studies also further validate the theoretical benefits of the model.

4. Reviewer 8izv and Reviewer imNB also find the paper exceptionally clear and well written.

**Reviewer Concerns:**

### Concerns addressed by the rebuttal: ###

- Reviewer 8izv's concern on the potentially significant deviation between practical implemnetation and theoretical guarantee caused by the mutli-step approximation devliations.

This concern is addressed by the rebuttal by both theoretical justifications and  empirical evidence.

- Reviewer UzG7 and Reviewer imNB's concern about the homogeneity assumption. The rebuttal clarifies that the Dense form in the paper is fully heterogeneous and doesn't make homogeneity assumptions.

- Reviewer UzG7's concern about limitations to pairwise relations in terms of interactions. The rebuttal clarifies that various domains considered in this paper, power grid, robotics, etc, are predominantly pairwise, and the non-pairwise interactions can be a natural next step.

- Reviewer imNB's concerns on adaptive weighting mechanism, stability, and assumptions are fully addressed by the rebuttal as acknowledged by the reviewer themselves.

- Reviewer 5oTz's concern on reduced complexity not formally proved. The rebuttal clarifed that the computation complexity is actually formally proved in the Appendix.

- Reviewer 5oTz's concern on the literature review on RKHS. The rebuttal adds more literature review and comparison.

- Reviewer 5oTz's concern on missing benchmark details. The rebuttal explains that the paper contains these information and points out the specific line numbers.


### Unsolved concerns: ###

- Reviewers 8izv and imNB's concern on scalability and generality is not fully addressed by the rebuttal. Nevertheless, considering the novelty of the proposed method, I think this could be left for future work.

**Reviewer Scores:**

- Reviewer 8izv (score 6): I don't think this score will change.

- Reviewer UzG7 (score 6): not going to change

- Reviewer imNB (score 6): the reviewer said they will keep the score

- Reviewer 5oTz (score 4): is likely to raise the score to 5 because the major weaknesses found by this reviewer are fully addressed in the rebuttal.

---

### Decision · Program_Chairs · 2026-01-26

Accept (Poster)